# In-Context Learning of a Linear Transformer Block: Benefits of the MLP Component and One-Step GD Initialization

**Ruiqi Zhang**
UC Berkeley
rqzhang@berkeley.edu

**Jingfeng Wu**
UC Berkeley
uuujf@berkeley.edu

**Peter L. Bartlett**
UC Berkeley and Google DeepMind
peter@berkeley.edu

## Abstract

We study the *in-context learning* (ICL) ability of a *Linear Transformer Block* (LTB) that combines a linear attention component and a linear multi-layer perceptron (MLP) component. For ICL of linear regression with a Gaussian prior and a *non-zero mean*, we show that LTB can achieve nearly Bayes optimal ICL risk. In contrast, using only linear attention must incur an irreducible additive approximation error. Furthermore, we establish a correspondence between LTB and one-step gradient descent estimators with learnable initialization (GD-$\beta$), in the sense that every GD-$\beta$ estimator can be implemented by an LTB estimator and every optimal LTB estimator that minimizes the in-class ICL risk is effectively a GD-$\beta$ estimator. Finally, we show that GD-$\beta$ estimators can be efficiently optimized with gradient flow, despite a non-convex training objective. Our results reveal that LTB achieves ICL by implementing GD-$\beta$, and they highlight the role of MLP layers in reducing approximation error.

## 1 Introduction

The recent dramatic progress in natural language processing can be attributed in large part to the development of large language models based on *Transformers* [34], such as BERT [11], LLaMA [32], PaLM [8], and the GPT series [28, 29, 6, 25]. In some pioneering pre-trained large language models, a new learning paradigm known as *in-context learning* (ICL) was observed (see, for example, [6, 7, 24, 20]). ICL refers to the capability of a pre-trained model to solve a new task based on a few in-context demonstrations without updating the model parameters.

A recent line of work quantifies ICL in tractable statistical learning setups such as linear regression with a Gaussian prior (see [14, 3] and references thereafter). Specifically, an ICL model takes a sequence $(\mathbf{x}_1, y_1, ..., \mathbf{x}_M, y_M, \mathbf{x}_{\mathsf{query}})$ as input and outputs a prediction of $y_{\mathsf{query}}$, where $(\mathbf{x}_i, y_i)_{i=1}^M$ and $(\mathbf{x}_{\mathsf{query}}, y_{\mathsf{query}})$ are independent samples from an unknown task-specific distribution (where the task admits a prior distribution). The ICL model is pre-trained by fitting many empirical observations of such sequence-label pairs. Experiments show that Transformers can achieve an ICL risk close to that achieved by Bayes optimal estimators in many statistical learning tasks [14, 3].

For ICL of linear regression with a Gaussian prior, the data is generated as

$$\mathbf{x} \sim \mathcal{N}(\mathbf{0}, \ \mathbf{H}), \quad y \mid \widetilde{\boldsymbol{\beta}}, \mathbf{x} \sim \mathcal{N}(\widetilde{\boldsymbol{\beta}}^\top \mathbf{x}, \ \sigma^2),$$

where $\widetilde{\boldsymbol{\beta}}$ is a task parameter that satisfies a Gaussian prior, $\widetilde{\boldsymbol{\beta}} \sim \mathcal{N}(\mathbf{0}, \ \psi^2 \mathbf{I})$. In this setup, [14, 3] showed in experiments that Transformers can achieve nearly optimal ICL by matching the performance of the Bayes optimal estimator, that is, an optimally tuned ridge regression. Besides, [35] showed by construction that a single *linear self-attention* (LSA) can implement *one-step gradient descent with zero initialization* (GD-0), offering an insight into the ICL mechanism of the Transformer.

38th Conference on Neural Information Processing Systems (NeurIPS 2024).

Later, theoretical works showed that optimal LSA models effectively correspond to GD-0 models [1], trained LSA models converge to GD-0 models [38], and GD-0 models (hence LSA models) can provably achieve nearly Bayes optimal ICL in certain regimes [37].

**Our contributions.** In this paper, we consider ICL in a more general setup, that is, linear regression with a Gaussian prior and a *non-zero mean* (that is, $\mathbb{E}[\widetilde{\boldsymbol{\beta}}] = \boldsymbol{\beta}^* \neq 0$). The non-zero mean in task prior captures a common scenario where tasks share a signal. In this setting, we show that the LSA models considered in prior papers [1, 38, 37] incur an irreducible additive approximation error. Furthermore, we show this approximation error is mitigated by considering a *linear Transformer block* (LTB) that combines a linear multi-layer perceptron (MLP) component and an LSA component. Our results highlight the important role of MLP layers in Transformers in reducing the approximation error, and they suggest that theories about LSA [1, 38, 37] do not fully explain the power of Transformers. Motivated by this understanding, we investigate LTB in depth and obtain the following additional results:

- We show that LTB can implement *one-step gradient descent with learnable initialization*, referred to by us as GD-$\boldsymbol{\beta}$. Additionally, we show that every optimal LTB estimator that minimizes the in-class ICL risk is *effectively* a GD-$\boldsymbol{\beta}$ estimator.

- Moreover, we show that the optimal GD-$\boldsymbol{\beta}$, hence also the optimal LTB, nearly matches the performance of the Bayes optimal estimator for linear regression with a Gaussian prior and a non-zero mean, provided that the signal-to-noise ratio is upper bounded. These two results together suggest that LTB performs nearly optimal ICL by implementing GD-$\boldsymbol{\beta}$.

- Finally, we show that the GD-$\boldsymbol{\beta}$ estimator can be efficiently optimized by gradient descent with an infinitesimal stepsize (that is, gradient flow) under the population ICL risk, despite the non-convexity of the objective.

**Paper organization.** The remaining paper is organized as follows. We conclude this section by introducing a set of notations. We then discuss related papers in Section 2. In Section 3, we set up our ICL problems and define LTB and LSA models mathematically. In Section 4, we show a positive approximation error gap between LSA and LTB, which highlights the importance of the MLP component and motivates our subsequent efforts to study LTB. In Section 5, we connect LSA estimators to GD-$\boldsymbol{\beta}$ estimators that are more interpretable for ICL of linear regression. In Section 6, we study the in-context learning and training of GD-$\boldsymbol{\beta}$ estimators. We conclude our paper and discuss future directions in Section 7.

**Notations.** We use lowercase bold letters to denote vectors and uppercase bold letters to denote matrices and tensors. For a vector $\mathbf{x}$ and a positive semi-definite (PSD) matrix $\boldsymbol{A}$, we write $\|\mathbf{x}\|_{\boldsymbol{A}} := \sqrt{\mathbf{x}^\top \boldsymbol{A} \mathbf{x}}$. We write $\langle \cdot, \cdot \rangle$ for the inner product, which is defined as $\langle \mathbf{x}, \mathbf{y} \rangle := \mathbf{x}^\top \mathbf{y}$ for vectors and $\langle \boldsymbol{A}, \boldsymbol{B} \rangle := \mathrm{tr}(\boldsymbol{A} \boldsymbol{B}^\top)$ for matrices. We write $\boldsymbol{A}[i]$ as the $i$-th row of the matrix $\boldsymbol{A}$, $\boldsymbol{A}_{m,n}$ as the $(m, n)$-th entry, and $\boldsymbol{A}_{-1,-1}$ as the right-bottom entry. We write $\mathbf{0}_n, \mathbf{0}_{m \times n}, \mathbf{I}_n$ for the zero vector, zero matrix and identity matrix, respectively. We denote the Kronecker product as $\otimes$. For two matrices $\boldsymbol{A}$ and $\boldsymbol{B}$, $\boldsymbol{A} \otimes \boldsymbol{B}$ is a linear mapping which operates as $(\boldsymbol{A} \otimes \boldsymbol{B}) \circ \boldsymbol{C} = \boldsymbol{B} \boldsymbol{C} \boldsymbol{A}^\top$. For a positive semi-definite matrix $\boldsymbol{A}$, we write $\boldsymbol{A}^{\frac{1}{2}}$ as its principle square root, which is the unique positive semi-definite matrix $\boldsymbol{B}$ such that $\boldsymbol{B} \boldsymbol{B} = \boldsymbol{B} \boldsymbol{B}^\top = \boldsymbol{A}$. We also write $\boldsymbol{A}^{-\frac{1}{2}} = (\boldsymbol{A}^{\frac{1}{2}})^+$, where $(\cdot)^+$ denotes the Moore Penrose pseudo-inverse. For two sets $A, B$ we write $A + B$ for the Minkowski sum, which is defined as $\{a + b : a \in A, b \in B\}$. We also write $a + B = \{a\} + B$ for an element $a$ and a set $B$. We write $\mathrm{null}\,(\cdot)$ for the null set of a matrix or a tensor.

## 2 Related Works

**Empirical results for ICL in controlled settings.** The work by [14] first considered ICL in controlled statistical learning setups. For noiseless linear regression, [14] showed in experiments that Transformers match the performance of the optimal estimator (that is, *Ordinary Least Square*). Subsequent works by [3, 18] extended their result to noisy linear regression with a Gaussian prior and showed that Transformers can match the performance of the Bayesian optimal estimator (that is, an optimally tuned ridge regression). Besides, [30] showed that the above holds even when Transformers are pretrained on a limited number of linear regression tasks. These papers only considered a Gaussian

prior with a zero mean. In contrast, we consider a Gaussian prior with a non-zero mean. Our setup better captures the common scenarios where the tasks share a signal.

The empirical investigation of ICL in tractable statistical learning setups goes beyond linear regression settings. For more examples, researchers empirically studied the ICL ability of Transformers for decision trees (see [14], they also considered two-layer networks), algorithm selection [4], linear mixture models [26], learning Fourier series [2], discrete boolean functions [5], representation learning [13, 16], and reinforcement learning [17, 19]. Among all these settings, Transformers can either compete with the Bayes optimal estimators or expert-designed strong benchmarks. These works are not directly comparable with our paper.

**Transformer implements gradient descent.** A line of work interpreted the ICL of Transformers by their abilities to implement *gradient descent* (GD) [35, 3, 10, 1, 38, 4, 37]. In experiments, [35, 10] showed that (multi-layer) Transformer outputs are close to (multi-step) GD outputs. When specialized to linear regression tasks, [35] constructed a single *linear self-attention* (LSA) that implements *one-step gradient descent with zero initialization* (GD-0). Subsequently, [1] showed that optimal LSA models effectively correspond to GD-0 models, [38] proved that trained LSA models converge to GD-0 models, [37] showed that GD-0 models (hence LSA models) can provably achieve nearly Bayes optimal ICL in certain regimes and provided a sharp task complexity analysis of the pre-training. These papers focused on LSA models. Instead, we consider a linear Transformer block that also utilizes the MLP layer.

From an approximation theory perspective, [3, 4] showed that Transformers can implement multi-step GD under general losses. In comparison, we consider a limited setting of linear regression and show the LSA models can implement one-step GD with learnable initialization (GD-$\beta$), moreover, every optimal LTB model is effectively an GD-$\beta$ model. Both of our constructions utilize MLP layers in Transformers, highlighting its importance in reducing approximation error. Different from their results, we also show a negative approximation result that reveals the limitation of LSA models.

## 3 Preliminaries

**Model input.** We use $\mathbf{x} \in \mathbb{R}^d$ and $y \in \mathbb{R}$ to denote a feature vector and its label, respectively. Throughout the paper, we assume a fixed number of context examples, denoted by $M > 0$. We denote the context examples by $(\mathbf{X}, \mathbf{y}) \in \mathbb{R}^{M \times d} \times \mathbb{R}^M$, where each row represents a context example, denoted by $(\mathbf{x}_i^\top, y_i)$, $i = 1, \ldots, M$. To formalize an ICL problem, the input of a model is a *token matrix* given by [1, 38]

$$\mathbf{E} := \begin{pmatrix} \mathbf{X}^\top & \mathbf{x} \\ \mathbf{y}^\top & 0 \end{pmatrix} \in \mathbb{R}^{(d+1) \times (M+1)}. \tag{3.1}$$

The output of a model corresponds to a prediction of $y$.

**A Transformer block.** Modern large language models are often made by stacking basic Transformer blocks (see, for example, [34]). A basic Transformer block consists of a *self-attention* layer and a *multi-layer perceptron* (MLP) layer [34], $\mathbf{E} \mapsto \mathsf{MLP}\left[\mathsf{ATTN}\left(\mathbf{E}\right)\right]$. Here, the MLP layer is defined as $\mathsf{MLP}\left(\mathbf{E}\right) := \mathbf{W}_2^\top \mathsf{ReLU}\left(\mathbf{W}_1 \mathbf{E}\right)$, where $\mathsf{ReLU}(\cdot)$ refers to the entrywise *rectified linear unit* (ReLU) activation function, and $\mathbf{W}_1$, $\mathbf{W}_2 \in \mathbb{R}^{d_f \times (d+1)}$ are two weight matrices. The self-attention layer (we focus on the single-head version in this paper) is defined as $\mathsf{ATTN}(\mathbf{E}) := \mathbf{E} + \mathbf{W}_P^\top \mathbf{W}_V \mathbf{E} \mathbf{M} \cdot \mathsf{sfmx}\left((\mathbf{W}_K \mathbf{E})^\top \mathbf{W}_Q \mathbf{E}\right)$, where $\mathsf{sfmx}(\cdot)$ refers to the row-wise softmax operator, $\mathbf{W}_K$, $\mathbf{W}_Q \in \mathbb{R}^{d_k \times (d+1)}$ are the key and query matrices, respectively, $\mathbf{W}_P$, $\mathbf{W}_V \in \mathbb{R}^{d_v \times (d+1)}$ are the projection and value matrices, respectively, and $\mathbf{M}$ is a fixed masking matrix given by

$$\mathbf{M} := \begin{pmatrix} \mathbf{I}_M & 0 \\ 0 & 0 \end{pmatrix} \in \mathbb{R}^{(M+1) \times (M+1)}. \tag{3.2}$$

This mask matrix is included to reflect the asymmetric structure of a prompt since the label of query input $\mathbf{x}$ is not included in the token matrix [1, 21]. In the above formulation, $d_k, d_v, d_f$ are three hyperparameters controlling the key size, value size, and width of the MLP layer, respectively. In the single-head case, it is common to set $d_k = d_v = d + 1$ and $d_f = 4(d + 1)$, where $d + 1$ corresponds to the embedding size (see, for example, [34]). Our formulation of a basic transformer block ignores all bias parameters and some popular techniques (such as layer normalization and dropout) to focus solely on the benefits brought by the model structure.

**A linear Transformer block.** To facilitate theoretical analysis, we ignore the non-linearities in the Transformer block (specifically, sfmx and ReLU) and work with a *linear Transformer block* (LTB) defined as

$$f_{\text{LTB}} : \mathbb{R}^{(d+1)\times(M+1)} \to \mathbb{R}, \quad \mathbf{E} \mapsto \left[ \mathbf{W}_2^\top \mathbf{W}_1 \left( \mathbf{E} + \mathbf{W}_P^\top \mathbf{W}_V \mathbf{E} \mathbf{M} \frac{\mathbf{E}^\top \mathbf{W}_K^\top \mathbf{W}_Q \mathbf{E}}{M} \right) \right]_{-1,-1},$$
(3.3)

where $\mathbf{E}$ is the token matrix given by (3.1), $[\,\cdot\,]_{-1,-1}$ refers to the bottom right entry of a matrix, $\mathbf{M}$ is the fixed masking matrix given by (3.2). Here, the trainable parameters are $\mathbf{W}_P, \mathbf{W}_V \in \mathbb{R}^{d_v \times (d+1)}$, $\mathbf{W}_K, \mathbf{W}_Q \in \mathbb{R}^{d_k \times (d+1)}$, and $\mathbf{W}_1, \mathbf{W}_2 \in \mathbb{R}^{d_f \times (d+1)}$. We use the bottom right entry of the transformed token matrix as the model output to form a prediction of the label $y$. The $1/M$ factor is a normalization factor in linear attention and can be absorbed into trainable parameters. Finally, we denote the hypothesis class formed by LTB models as $\mathcal{F}_{\text{LTB}} := \{ f_{\text{LTB}} : \mathbf{W}_K, \mathbf{W}_Q, \mathbf{W}_V, \mathbf{W}_P, \mathbf{W}_1, \mathbf{W}_2, d_k \geq d, \ d_v \geq d+1, \ d_f \geq 1 \}$, where $f_{\text{LTB}}$ is defined in (3.3).

**A linear self-attention.** We will also consider a *linear self-attention* (LSA) defined as

$$f_{\text{LSA}} : \mathbb{R}^{(d+1)\times(M+1)} \to \mathbb{R}, \quad \mathbf{E} \mapsto \left[ \mathbf{E} + \mathbf{W}_P^\top \mathbf{W}_V \mathbf{E} \mathbf{M} \frac{\mathbf{E}^\top \mathbf{W}_K^\top \mathbf{W}_Q \mathbf{E}}{M} \right]_{-1,-1},$$
(3.4)

where $\mathbf{W}_K, \mathbf{W}_Q, \mathbf{W}_P, \mathbf{W}_V$ are trainable parameters. An LSA model can be viewed as an LTB model without the MLP layer (setting $d_f = d + 1$ and $\mathbf{W}_1 = \mathbf{W}_2 = \mathbf{I}$). We remark that, unlike LTB, the residual connection in LSA plays no role because the bottom right entry of the prompt $\mathbf{E}$ is zero (see (3.1)). A variant of the LSA model has been studied by [1, 38, 37], where $\mathbf{W}_K^\top \mathbf{W}_Q$ and $\mathbf{W}_P^\top \mathbf{W}_V$ are respectively merged into one matrix parameter. Similarly, we denote the hypothesis class formed by LSA models as $\mathcal{F}_{\text{LSA}} := \{ f_{\text{LSA}} : \mathbf{W}_K, \mathbf{W}_Q, \mathbf{W}_V, \mathbf{W}_P, d_k \geq d, d_v \geq 1 \}$, where $f_{\text{LSA}}$ is defined in (3.4).

**Linear regression tasks with a shared signal.** Assume that data and context examples are generated as follows.

**Assumption 3.1** (Distributional conditions)**.** Assume that $(\mathbf{X}, \mathbf{y}, \mathbf{x}, y)$ are generated by:

- First, a task parameter is independently generated by $\widetilde{\boldsymbol{\beta}} \sim \mathcal{N}(\boldsymbol{\beta}^*, \boldsymbol{\Psi})$.

- The feature vectors are independently generated by $\mathbf{x}, \mathbf{x}_1, \dots \mathbf{x}_M \overset{\text{i.i.d.}}{\sim} \mathcal{N}(0, \mathbf{H})$.

- Then, the labels are generated by $y = \langle \widetilde{\boldsymbol{\beta}}, \mathbf{x} \rangle + \varepsilon$, $y_i = \langle \widetilde{\boldsymbol{\beta}}, \mathbf{x}_i \rangle + \varepsilon_i$, $i = 1, \dots, M$, where $\varepsilon$ and $\varepsilon_i$'s are independently generated by $\varepsilon, \varepsilon_1, \dots, \varepsilon_M \overset{\text{i.i.d.}}{\sim} \mathcal{N}(0, \sigma^2)$.

Here, $\sigma^2 \geq 0$, $\mathbf{H} \succeq \mathbf{0}$, $\boldsymbol{\Psi} \succeq \mathbf{0}$, and $\boldsymbol{\beta}^* \in \mathbb{R}^d$ are fixed but unknown quantities that govern the data distribution. We denote $\boldsymbol{\varepsilon} = (\varepsilon_1, ..., \varepsilon_M)^\top$.

We emphasize the importance of the mean of the task parameter $\boldsymbol{\beta}^*$ in Assumption 3.1. A non-zero $\boldsymbol{\beta}^*$ represents a shared signal across tasks, which is arguably common in practice. This assumption is implicitly used in [12] where they assumed task parameters are close to a meta parameter. In comparison, the prior works for ICL of linear regression [1, 38, 37] only considered a special case where $\boldsymbol{\beta}^* = 0$. In this special case, they showed that a single LSA layer can achieve nearly optimal ICL by approximating one-step gradient descent from zero initialization (GD-0). In more general cases where $\boldsymbol{\beta}^*$ is non-zero, we will show in Section 4 that LSA is insufficient to learn the shared signal and must incur an irreducible approximation error compared to LTB models. This sets our results apart from the prior papers.

**ICL risk.** We measure the ICL risk of a model $f$ by the mean squared error,

$$\mathcal{R}(f) := \mathbb{E}(f(\mathbf{E}) - y)^2,$$
(3.5)

where $\mathbf{E}$ is defined in (3.1) and the expectation is over $\mathbf{E}$ (equivalent to over $\mathbf{X}, \mathbf{y}$, and $\mathbf{x}$) and $y$.

## 4 Benefits of the MLP Component

Our first main result separates the approximation abilities of the LTB and LSA models for ICL. Recall that an LSA model can be viewed as a special case of the LTB model with $\mathbf{W}_2 = \mathbf{W}_1 = \mathbf{I}$. So the

best ICL risk achieved by LTB models is no larger than the best ICL risk achieved by LSA models. However, our next theorem shows a strictly positive gap between the best ICL risks achieved by those two model classes. This result highlights the benefits of the MLP layer for reducing approximation error in Transformer.

**Theorem 4.1** (Approximation gap). *Consider the ICL risk defined by* (3.5) *and the two hypothesis classes* $\mathcal{F}_{\mathsf{LTB}}$ *and* $\mathcal{F}_{\mathsf{LSA}}$. *Suppose that Assumption 3.1 holds. Then we have*

- $\inf_{f \in \mathcal{F}_{\mathsf{LTB}}} \mathcal{R}(f)$ *is independent of* $\boldsymbol{\beta}^*$.

- $\inf_{f \in \mathcal{F}_{\mathsf{LSA}}} \mathcal{R}(f)$ *is a function of* $\boldsymbol{\beta}^*$. *Moreover,*

$$\inf_{f \in \mathcal{F}_{\mathsf{LSA}}} \mathcal{R}(f) - \inf_{f \in \mathcal{F}_{\mathsf{LTB}}} \mathcal{R}(f) \geq \max \left\{ \frac{2}{3(M+1)}, \frac{\left(\operatorname{tr}(\mathbf{H}\boldsymbol{\Psi}) + \sigma^2\right)^2}{(M+1)^2 \operatorname{tr}((\mathbf{H}\boldsymbol{\Psi})^2)} \right\} \|\boldsymbol{\beta}^*\|_{\mathbf{H}}^2.$$

The proof of Theorem 4.1 is deferred to Appendix B. Theorem 4.1 reveals a gap in terms of the approximation abilities between the hypothesis set of LTB models and that of LSA models. Specifically, the best ICL performance achieved by LTB models is independent of $\boldsymbol{\beta}^*$, while the best ICL performance achieved by LSA models is sensitive to the norm of $\boldsymbol{\beta}^*$. In particular, when $\|\boldsymbol{\beta}^*\|_{\mathbf{H}}^2 = \Omega(M)$, the best ICL risk of the former is smaller than that of the latter by at least a *constant* additive term. So when $\boldsymbol{\beta}^*$ is large, the hypothesis class formed by LSA models is restricted in its ability to perform effective ICL. We will show in Section 5 that the hypothesis class formed by LTB models can achieve nearly optimal ICL in this case.

We also remark that the $\Theta(1/M)$ factor in the lower bound in Theorem 4.1 is not improvable. This is because LSA models can implement a one-step GD algorithm that is *consistent* for linear regression tasks (that is, the risk converges to the Bayes risk as the number of context examples goes to infinity), with an excess risk bound of $\Theta(1/M)$ [38, 37]. So the approximation error gap is at most $\Theta(1/M)$. Nonetheless, the $\Omega(\|\boldsymbol{\beta}^*\|_{\mathbf{H}}^2/M)$ approximation gap between LSA models and LTB models shown by Theorem 4.1 suggests that $\boldsymbol{\beta}^*$ is not learnable by LSA models during pre-training. In contrast, we will show in Section 6 that LTB models can learn $\boldsymbol{\beta}^*$ during pre-training.

We emphasize that the ability of LTB to learn non-zero mean is a joint effect of an MLP component and a skip connection. Note that LSA also has a skip connection, but its skip connection is inactive. In comparison, the MLP component in LTB activates the skip connection. Therefore, we attribute the ability to learn non-zero mean to the MLP component, which is the only difference between LTB and LSA. Nonetheless, one can attribute the ability to learn non-zero mean to the skip connection: without a skip connection, LTB reduces to LSA with a potential rank constraint on the parameter, which cannot learn non-zero mean as we have proved. The above two explanations take different perspectives to interpret the same phenomenon. Finally, we remark that Theorem 4.1 holds even when $\mathbf{H}$ and $\boldsymbol{\Psi}$ in Assumption 3.1 are not full rank.

**Does scratchpad help?** We have demonstrated that employing a single-layer LSA introduces an additional approximation error in the in-context learning problem for the linear regression task defined in Assumption 3.1. Nonetheless, are there alternative structures, apart from MLPs, that could potentially reduce this approximation error? One plausible strategy is to include a "scratchpad" in the input token. Specifically, we construct the token matrix as follows:

$$\mathbf{E} := \begin{pmatrix} \mathbf{X}^\top & \mathbf{x} \\ \mathbf{1}_M^\top & 1 \\ \mathbf{y}^\top & 0 \end{pmatrix} \in \mathbb{R}^{(d+2) \times (M+1)} \tag{4.1}$$

which we then input into the LSA layer. We discuss the limitation of this scheme in Appendix J. Notably, this method does not successfully recover the GD-$\boldsymbol{\beta}$ estimator defined in Section 5. We leave it as future work to see whether the token matrix with scratchpad could implement other types of estimators that more effectively address the linear regression tasks defined in Assumption 3.1, as well as whether additional structures could help alleviate this approximation error.

**Experiments on GPT2.** Theorem 4.1 shows the importance of the MLP component in LSA models for reducing approximation error. We also empirically validate this result by training a more complex GPT2 model [14] for the ICL tasks specified by Assumption 3.1. In the experiments, we use a GPT2-small model (with or without the MLP component) with 6 layers and 4 heads in each layer. The experiments follow the setting in [14], except that we train the model using a token matrix defined

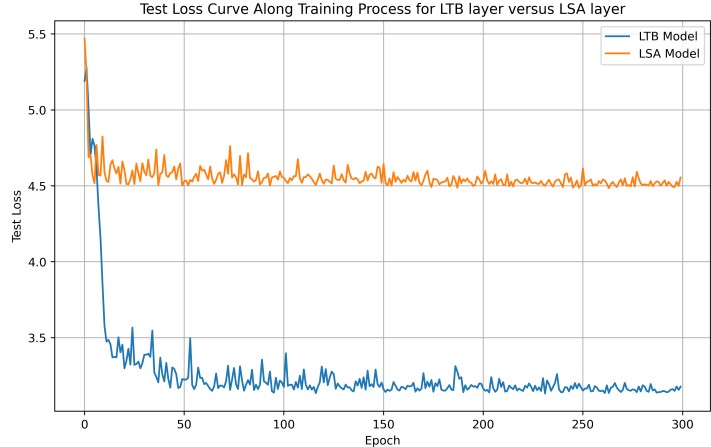

Figure 1: The test loss along the training process for LTB and LSA layer.

in (3.1). We considere two ICL settings, which instantiates Assumption 3.1 with $\boldsymbol{\beta}^* = (0, 0, ..., 0)^\top$ and $\boldsymbol{\beta}^* = (10, 10, ..., 10)^\top$, respectively. We set $d = 20, M = 40, \sigma = 0, \boldsymbol{\Psi} = \mathbf{H} = \mathbf{I}_d$ in both settings. More experimental details are in Appendix I. In each setting, we train and test the model using the same data distribution. The experimental results are presented in Table 1. From Table 1, we observe that both models (with or without the MLP component) achieve a nearly zero loss when the task mean is zero. However, when the task mean is set away from zero, the GPT2 model with MLP component still performs relatively well while the GPT2 model without MLP component incurs a significantly larger loss. These empirical observations are consistent with our Theorem 4.1, indicating the benefits of MLP layers in reducing approximation error for ICL of linear regression with a shared signal.

**Experiments on LSA and LTB.** We trained both the LSA and LTB layers for $\boldsymbol{\beta}^* = (1, 1, ..., 1)^\top$, and found that the trained LTB layer consistently achieved a significantly lower ICL risk than the LSA layer (see Figure 1). In these experiments, we adhered strictly to the previously defined LSA and LTB structures, setting the parameters as follows: $d = 5, M = 5, \sigma = 0$, and $\boldsymbol{\Psi} = \mathbf{H} = \mathbf{I}_d$. At each training step, we sampled $B = 128$ new linear regression tasks.

Table 1: Losses of GPT2 with or without MLP component for linear regression with a shared signal.

| Model | GPT2 | GPT2-noMLP |
|---|---|---|
| $\boldsymbol{\beta}^* = 0 \times \mathbf{1}$ | 0.003 | 0.024 |
| $\boldsymbol{\beta}^* = 10 \times \mathbf{1}$ | 0.013 | 8.871 |

In this part, we have shown that LSA models, the primary focus in previous ICL theory literature (see, e.g., [1, 38, 37] and references therein), are not sufficiently expressive for ICL of linear regression with a shared signal. In what follows, we will show LTB models are sufficient for this ICL problem.

# 5 LTB Implements One-Step GD with Learnable Initialization

To understand the expressive power of the LTB models, we build a connection between $\mathcal{F}_{\text{LTB}}$ and its subset of models which we call *one-step GD with learnable initialization* (GD-$\boldsymbol{\beta}$). We will first introduce GD-$\boldsymbol{\beta}$ models and then show that the best LTB models that minimize the ICL risk effectively belong to GD-$\boldsymbol{\beta}$ models.

**The GD-$\boldsymbol{\beta}$ models.** A GD-$\boldsymbol{\beta}$ model is defined as

$$f_{\text{GD-}\boldsymbol{\beta}} : \mathbb{R}^{(d+1)\times(M+1)} \to \mathbb{R}, \quad \mathbf{E} \mapsto \left\langle \boldsymbol{\beta} - \boldsymbol{\Gamma} \cdot \frac{1}{M} \mathbf{X}^\top (\mathbf{X}\boldsymbol{\beta} - \mathbf{y}), \mathbf{x} \right\rangle, \qquad (5.1)$$

where $\mathbf{E}$ is the token matrix given by (3.1), $\boldsymbol{\Gamma} \in \mathbb{R}^{d \times d}$ and $\boldsymbol{\beta} \in \mathbb{R}^d$ are trainable parameters. Similarly, we define the function class formed by GD-$\boldsymbol{\beta}$ models as $\mathcal{F}_{\text{GD-}\boldsymbol{\beta}} := \{f_{\text{GD-}\boldsymbol{\beta}} : \boldsymbol{\beta} \in \mathbb{R}^d, \boldsymbol{\Gamma} \in \mathbb{R}^{d \times d}\}$.

A GD-$\boldsymbol{\beta}$ model computes a parameter that fits the context examples $(\mathbf{X}, \mathbf{y})$ and uses that parameter to make a linear prediction of label $y$ on feature $\mathbf{x}$. More specifically, the first step is by using one gradient descent step with a *matrix stepsize* $\boldsymbol{\Gamma}$ and an *initialization* $\boldsymbol{\beta}$ on the least square objective formed by context examples $(\mathbf{X}, \mathbf{y})$.

**LTB implements GD-$\boldsymbol{\beta}$.** A GD-$\boldsymbol{\beta}$ model (5.1) is a special case of an LTB model (3.3) by setting

$$\mathbf{W}_2^\top \mathbf{W}_1 = \begin{pmatrix} * & * \\ \boldsymbol{\beta}^\top & 1 \end{pmatrix}, \quad \mathbf{W}_P^\top \mathbf{W}_V = \begin{pmatrix} -\mathbf{I}_d & \mathbf{0}_d \\ \mathbf{0}_d^\top & 1 \end{pmatrix}, \quad \mathbf{W}_K^\top \mathbf{W}_Q = \begin{pmatrix} \boldsymbol{\Gamma} & * \\ \mathbf{0}_d^\top & * \end{pmatrix},$$

where $*$ denotes the entries that do not affect the model output (hence can be set to anything). Note that $\boldsymbol{\Gamma}$ and $\boldsymbol{\beta}$ are *free* provided that $d_v \geq d + 1$, $d_k \geq d$ and $d_f \geq 1$ (as required in $\mathcal{F}_{\mathsf{LTB}}$). In sum, we have proved the following lemma showing that the set of GD-$\boldsymbol{\beta}$ models belongs to the set of LTB models.

**Lemma 5.1.** *We have $\mathcal{F}_{\mathsf{GD}\text{-}\boldsymbol{\beta}} \subseteq \mathcal{F}_{\mathsf{LTB}}$. Therefore, $\inf_{f \in \mathcal{F}_{\mathsf{GD}\text{-}\boldsymbol{\beta}}} \mathcal{R}(f) \geq \inf_{f \in \mathcal{F}_{\mathsf{LTB}}} \mathcal{R}(f)$.*

**Optimal GD-$\boldsymbol{\beta}$ models for ICL.** We now consider $\mathcal{F}_{\mathsf{GD}\text{-}\boldsymbol{\beta}}$ and its optimal ICL risk. We have the following theorem that computes the globally minimal ICL risk over $\mathcal{F}_{\mathsf{GD}\text{-}\boldsymbol{\beta}}$ and specifies the sufficient and necessary conditions for a global minimizer. The proof is deferred to Appendix C.

**Theorem 5.2** (Optimal GD-$\boldsymbol{\beta}$ models). *Consider the ICL risk defined by (3.5). Suppose that Assumption 3.1 holds. Then we have*

- *The minimal ICL risk of $\mathcal{F}_{\mathsf{GD}\text{-}\boldsymbol{\beta}}$ is*

$$\inf_{f \in \mathcal{F}_{\mathsf{GD}\text{-}\boldsymbol{\beta}}} \mathcal{R}(f) = \sigma^2 + \operatorname{tr}\left( \mathbf{H}^{\frac{1}{2}} \boldsymbol{\Psi} \mathbf{H}^{\frac{1}{2}} \left( \mathbf{I} - \mathbf{H}^{\frac{1}{2}} \boldsymbol{\Psi} \mathbf{H}^{\frac{1}{2}} \boldsymbol{\Omega}^{-1} \right) \right), \tag{5.2}$$

*where $\boldsymbol{\Omega} := [(M+1)\mathbf{H}^{\frac{1}{2}} \boldsymbol{\Psi} \mathbf{H}^{\frac{1}{2}} + (\operatorname{tr}(\mathbf{H}\boldsymbol{\Psi}) + \sigma^2)\mathbf{I}_d]/M$.*

- *The global optimal parameters for $f_{\mathsf{GD}\text{-}\boldsymbol{\beta}}$ that attain the minimum (5.2) take the following form:*

$$\boldsymbol{\beta} = \boldsymbol{\beta}^* + \mathsf{null}(\mathbf{H}), \quad \boldsymbol{\Gamma} = \boldsymbol{\Gamma}^* + \mathsf{null}(\mathbf{H}^{\otimes 2}), \quad \text{where} \quad \boldsymbol{\Gamma}^* := \boldsymbol{\Psi} \mathbf{H}^{\frac{1}{2}} \boldsymbol{\Omega}^{-1} \mathbf{H}^{-\frac{1}{2}} \tag{5.3}$$

*and $\boldsymbol{\beta}^*$ is the mean of the task parameter in Assumption 3.1 Here, $\mathsf{null}(\mathbf{H}) := \{ \mathbf{z} \in \mathbb{R}^d : \mathbf{H}\mathbf{z} = \mathbf{0} \}$ is the null space of $\mathbf{H}$, and $\mathsf{null}(\mathbf{H}^{\otimes 2}) = \{ \mathbf{Z} \in \mathbb{R}^{d \times d} : \mathbf{H}\mathbf{Z}\mathbf{H} = \mathbf{0} \}$ is the null space of $\mathbf{H}^{\otimes 2}$. In particular, when $\mathbf{H}$ is positive definite, the global optimal parameter is unique, $(\boldsymbol{\beta}, \boldsymbol{\Gamma}) = (\boldsymbol{\beta}^*, \boldsymbol{\Gamma}^*)$.*

- *Under Assumption 3.1, the global optimal $f_{\mathsf{GD}\text{-}\boldsymbol{\beta}}$ that attains the minimum (5.2) is unique as a function of $\mathbf{E}$ (given by (3.1)) and takes the following form:*

$$f^*(\mathbf{E}) = \left\langle \boldsymbol{\beta}^* - \frac{1}{M} \boldsymbol{\Gamma}^* \mathbf{X}^\top (\mathbf{X}\boldsymbol{\beta}^* - \mathbf{y}), \mathbf{x} \right\rangle. \tag{5.4}$$

Theorem 5.2 characterizes the optimal GD-$\boldsymbol{\beta}$ models for ICL. In the above theorem, $\boldsymbol{\Gamma}^* \to \mathbf{H}^{-1}$ as the context length $M$ goes to infinity (assuming that $\mathbf{H}$ is positive definite). In this case, the optimal GD-$\boldsymbol{\beta}$ function (5.2) implements one Newton step from initialization $\boldsymbol{\beta}^*$. With a finite context length $M$, (5.2) implements one regularized Newton step from initialization $\boldsymbol{\beta}^*$.

**Optimal LTB models for ICL.** Lemma 5.1 shows that the best ICL risk achieved by an GD-$\boldsymbol{\beta}$ model is no smaller than the best ICL risk achieved by an LTB model. Surprisingly, our next theorem shows that the best ICL risk achieved by a GD-$\boldsymbol{\beta}$ is *equal* to that achieved by an LTB model. Therefore, the hypothesis set $\mathcal{F}_{\mathsf{GD}\text{-}\boldsymbol{\beta}}$ is diverse enough to match the approximation ability of the larger hypothesis set $\mathcal{F}_{\mathsf{LTB}}$ for ICL.

**Theorem 5.3** (Optimal LTB models). *Consider the ICL risk defined by (3.5). Suppose that Assumption 3.1 holds and that $\mathsf{rank}(\mathbf{H}^{\frac{1}{2}} \boldsymbol{\Psi}^{\frac{1}{2}}) \geq 2$. Then we have*

- *The minimal ICL risk of $\mathcal{F}_{\mathsf{LTB}}$ and of $\mathcal{F}_{\mathsf{GD}\text{-}\boldsymbol{\beta}}$ are equal,*

$$\inf_{f \in \mathcal{F}_{\mathsf{LTB}}} \mathcal{R}(f) = \inf_{f \in \mathcal{F}_{\mathsf{GD}\text{-}\boldsymbol{\beta}}} \mathcal{R}(f) = \text{RHS of (5.2)}$$

- *Rewrite an LTB model as $f_{\mathsf{LTB}}$ in (3.3) with parameters ($*$ denotes parameters that do not affect the output)*

$$\mathbf{W}_2^\top \mathbf{W}_1 = \begin{pmatrix} * & * \\ \boldsymbol{\gamma}^\top & * \end{pmatrix}, \quad \mathbf{W}_K^\top \mathbf{W}_Q = \begin{pmatrix} \mathbf{V}_{11} & * \\ \mathbf{v}_{12}^\top & * \end{pmatrix}, \quad \mathbf{W}_2^\top \mathbf{W}_1 \mathbf{W}_P^\top \mathbf{W}_V = \begin{pmatrix} * & * \\ \mathbf{v}_{21}^\top & v_{-1} \end{pmatrix}.$$

  *Then the sufficient and necessary conditions for $f_{\mathsf{LTB}} \in \arg\min_{f \in \mathcal{F}_{\mathsf{LTB}}} \mathcal{R}(f)$ are*

$$v_{-1} \neq 0, \quad v_{-1}\mathbf{v}_{12} \in \mathsf{null}\,(\mathbf{H}), \quad \mathbf{v}_{21} \in -v_{-1}\boldsymbol{\beta}^* + \mathsf{null}\,(\mathbf{H}), \quad \boldsymbol{\gamma} \in \boldsymbol{\beta}^* + \mathsf{null}\,(\mathbf{H}),$$
$$v_{-1}\mathbf{V}_{11} \in \boldsymbol{\Gamma}^{*\top} - v_{-1}\boldsymbol{\beta}^*\mathbf{v}_{12}^\top + \mathsf{null}\,\left(\mathbf{H}^{\otimes 2}\right),$$

  *where $\boldsymbol{\beta}^*$ is defined in Assumption 3.1 and $\boldsymbol{\Gamma}^*$ is defined in (5.3). In particular, when $\mathbf{H}$ is positive definite, the globally optimal parameter represented by $(\mathbf{V}_{11}, \mathbf{v}_{12}, \mathbf{v}_{21}, v_{-1}, \boldsymbol{\gamma})$ is unique up to a rescaling of $v_{-1}$.*

- *Under Assumption 3.1, the globally optimal LTB model (that is, a function in $\arg\min_{f \in \mathcal{F}_{\mathsf{LTB}}}$) is unique as a function of $\mathbf{E}$ (given by (3.1)) and takes the form of (5.4) almost surely.*

Lemma 5.1 and Theorem 5.3 together show that $\mathcal{F}_{\mathsf{GD}\text{-}\boldsymbol{\beta}}$ is a representative subset of $\mathcal{F}_{\mathsf{LTB}}$ that does not incur additional approximation error. In addition, every optimal LTB model is effectively an optimal GD-$\boldsymbol{\beta}$ model when restricted to all possible token matrices. Note that the optimal model parameters for LTB or GD-$\boldsymbol{\beta}$ are not unique because of redundant parameterization. But the optimal LTB and GD-$\boldsymbol{\beta}$ models are unique as a function of all possible token matrices. The above holds even when $\mathbf{H}$ and $\boldsymbol{\Psi}$ are potentially rank deficient.

**Comparison with prior works.** A line of papers considers LSA models for ICL under Assumption 3.1 with $\boldsymbol{\beta}^* = 0$ [35, 1, 38, 37]. They show that LSA models can (effectively) implement all possible GD-0 models that specialize GD-$\boldsymbol{\beta}$ models by fixing $\boldsymbol{\beta} = \mathbf{0}$. In addition, they show that every optimal LSA model is (effectively) a GD-0 model for ICL under Assumption 3.1 with $\boldsymbol{\beta}^* = \mathbf{0}$ (see, for example, Theorem 1 in [1]). In comparison, we consider a harder ICL problem that allows a large shared signal in tasks (that is, a large $\boldsymbol{\beta}^*$ in Assumption 3.1). In this setting, our Theorem 4.1 shows that $\mathcal{F}_{\mathsf{LSA}}$ (hence its subset formed by GD-0 models), as a subet of $\mathcal{F}_{\mathsf{LTB}}$, incurs an additional approximation error propotional to $\|\boldsymbol{\beta}^*\|_{\mathbf{H}}^2$ compared with $\mathcal{F}_{\mathsf{LTB}}$. In contrast, $\mathcal{F}_{\mathsf{GD}\text{-}\boldsymbol{\beta}}$, as a subset of $\mathcal{F}_{\mathsf{LTB}}$, does not incur additional approximation error according to our Theorem 5.3. Thus the LSA and GD-0 models considered by [35, 1, 38, 37] are not capable of learning the shared signal $\boldsymbol{\beta}^*$, while an LTB model can learn $\boldsymbol{\beta}^*$ through implementing GD-$\boldsymbol{\beta}$ and encoding $\boldsymbol{\beta}^*$ in the initialization parameter.

## 6 Training and In-Context Learning of GD-beta

We have shown that $\mathcal{F}_{\mathsf{GD}\text{-}\boldsymbol{\beta}}$ is a representative subset of $\mathcal{F}_{\mathsf{LTB}}$ that effectively contains every optimal LTB model. We now examine the ICL and training of GD-$\boldsymbol{\beta}$ models.

**Nearly optimal ICL with GD-$\boldsymbol{\beta}$.** We will compare the best ICL risk achieved by GD-$\boldsymbol{\beta}$ with the best ICL risk achieved by any estimator. The following lemma is an extension of Proposition 5.1 and Corollary 5.2 in [37] (which is based on [33]) that characterizes the Bayes optimal ICL risk among all estimators.

**Lemma 6.1** (Bayes optimal ICL). *Given a task-specific dataset $(\mathbf{X}, \mathbf{y}, \mathbf{x}, y)$ sampled according to Assumption 3.1, let $g(\mathbf{X}, \mathbf{y}, \mathbf{x})$ be an arbitrary estimator for $y$ and measure the average linear regression risk by $\mathcal{L}(g; \mathbf{X}) := \mathbb{E}[(g(\mathbf{X}, \mathbf{y}, \mathbf{x}) - y)^2 \mid \mathbf{X}]$. It is clear that $\mathbb{E}\mathcal{L}(g; \mathbf{X}) = \mathcal{R}(g)$. Then,*

- *The optimal estimator that minimizes the average linear regression risk $\mathcal{L}(\cdot; \mathbf{X})$ is $g^*(\mathbf{X}, \mathbf{y}, \mathbf{x}) = \mathbf{x}^\top \boldsymbol{\beta}^* + \mathbf{x}^\top \boldsymbol{\Psi}^{\frac{1}{2}} \left(\boldsymbol{\Psi}^{\frac{1}{2}} \mathbf{X}^\top \mathbf{X} \boldsymbol{\Psi}^{\frac{1}{2}} + \sigma^2 \mathbf{I}_d\right)^{-1} \boldsymbol{\Psi}^{\frac{1}{2}} \mathbf{X}^\top (\mathbf{y} - \mathbf{X}\boldsymbol{\beta}^*).$*

- *Assume the signal-to-noise ratio is upper bounded, that is, $\mathrm{tr}\,(\mathbf{H}\boldsymbol{\Psi}) \lesssim \sigma^2$, then with probability at least $1 - \exp\left(-\Omega\,(M)\right)$ over the randomness of $\mathbf{X}$, it holds that $\mathcal{L}\,(g^*; \mathbf{X}) - \sigma^2 \simeq \sum_{i=1}^d \min\{\bar{\phi}, \phi_i\}$, where $\bar{\phi} \asymp \frac{\sigma^2}{M}$, and $(\phi_i)_{i \geq 1}$ are the eigenvalues of $\boldsymbol{\Psi}^{\frac{1}{2}} \mathbf{H} \boldsymbol{\Psi}^{\frac{1}{2}}$.*

The proof is deferred to Appendix F. Lemma 6.1 shows that the Bayes optimal estimator is a ridge regression estimator centered at $\boldsymbol{\beta}^*$. This is consistent with [37] where the Bayes optimal estimator

is a ridge regression estimator as they assumed $\boldsymbol{\beta}^* = 0$. The following corollary of Theorem 5.2 computes the rate of the ICL risk achieved by the optimal GD-$\boldsymbol{\beta}$ model.

**Corollary 6.2.** *Under the setup of Theorem 5.2, additionaly assume the signal-to-noise ratio is upper bounded, that is,* $\operatorname{tr}(\boldsymbol{\Psi}\mathbf{H}) \lesssim \sigma^2$. *Then we have* $\inf_{f \in \mathcal{F}_{\mathsf{GD-\beta}}} \mathcal{R}(f) - \sigma^2 \simeq \sum_{i=1}^{d} \min\{\bar{\phi}, \phi_i\}$, *where* $(\phi_i)_{i\geq 0}$ *are the eigenvalues of* $\boldsymbol{\Psi}^{\frac{1}{2}}\mathbf{H}\boldsymbol{\Psi}^{\frac{1}{2}}$ *and* $\bar{\phi} := [\operatorname{tr}(\boldsymbol{\Psi}^{\frac{1}{2}}\mathbf{H}\boldsymbol{\Psi}^{\frac{1}{2}}) + \sigma^2]/M \simeq \sigma^2/M$.

The optimal (expected) ICL risk achieved by $\mathcal{F}_{\mathsf{GD-\beta}}$ in Corollary 6.2 matches the (high probability) Bayes optimal ICL risk in Lemma 6.1 ignoring constant factors, provided that the signal-to-noise ratio is upper bounded. Therefore $\mathcal{F}_{\mathsf{GD-\beta}}$ achieves nearly Bayes optimal ICL risk. As a consequence, the larger hypothesis set $\mathcal{F}_{\mathsf{LTB}}$ also achieves nearly Bayes optimal ICL of linear regression under Assumption 3.1.

For the simplicity of discussion, we assume a fixed context length during pretraining and inference. Our discussions can be extended to allow a different context length during pretraining and inference using techniques in [37]. However, this is not the main focus of this work.

**Optimization of** GD-$\boldsymbol{\beta}$ **with infinite tasks.** We have shown that $\mathcal{F}_{\mathsf{GD-\beta}}$ is a representative subset of $\mathcal{F}_{\mathsf{LTB}}$ that covers the optimal LTB models and achieves nearly optimal ICL risk. We now consider the optimization in the parameter space specified by $\mathcal{F}_{\mathsf{GD-\beta}}$. For simplicity, we follow [38] and consider gradient descent with an infinitesimal stepsize on the ICL objective with an infinite number of tasks. That is, we consider the optimization of gradient flow on the population ICL risk under the parameterization of GD-$\boldsymbol{\beta}$,

$$\frac{\mathrm{d}\boldsymbol{\beta}(t)}{\mathrm{d}t} = -\frac{\partial}{\partial\boldsymbol{\beta}}\mathcal{R}(f_{\mathsf{GD-\beta}}), \quad \frac{\mathrm{d}\boldsymbol{\Gamma}(t)}{\mathrm{d}t} = -\frac{\partial}{\partial\boldsymbol{\Gamma}}\mathcal{R}(f_{\mathsf{GD-\beta}}), \tag{6.1}$$

where $\mathcal{R}$ is defined by (3.5) and $f_{\mathsf{GD-\beta}}$ is defined by (5.1).

The following theorem guarantees the global convergence of gradient flow. We introduce some notation to accommodate cases when $\mathbf{H}$ is rank deficient. Let $\mathcal{P}_{\mathcal{S}}$ be the orthogonal projection operator onto a subspace $\mathcal{S}$. Let $\mathcal{H} = \mathsf{Im}(\mathbf{H})$ be the image space of matrix $\mathbf{H}$ (viewing $\mathbf{H}$ as a linear map) and $\mathcal{H}^\perp := \mathsf{null}(\mathbf{H})$ be its orthogonal complement. Similarly, let $\mathcal{Z} := \mathsf{Im}(\mathbf{H}^{\otimes 2}) = \{\mathbf{HZH}, \boldsymbol{Z} \in \mathbb{R}^{d \times d}\}$ be the image space of the operator $\mathbf{H}^{\otimes 2}$ and $\mathcal{Z}^\perp$ be its orthogonal complement. Then we have the following theorem.

**Theorem 6.3.** *Consider the gradient flow defined by* (6.1) *with initialization* $\boldsymbol{\beta}(0), \boldsymbol{\Gamma}(0)$. *We have,*

$$\mathcal{P}_{\mathcal{H}}(\boldsymbol{\beta}(t)) \to \mathcal{P}_{\mathcal{H}}(\boldsymbol{\beta}^*), \quad \mathcal{P}_{\mathcal{H}^\perp}(\boldsymbol{\beta}(t)) = \mathcal{P}_{\mathcal{H}^\perp}(\boldsymbol{\beta}(0)),$$
$$\mathcal{P}_{\mathcal{Z}}(\boldsymbol{\Gamma}(t)) \to \mathcal{P}_{\mathcal{Z}}(\boldsymbol{\Gamma}^*), \quad \mathcal{P}_{\mathcal{Z}^\perp}(\boldsymbol{\Gamma}(t)) = \mathcal{P}_{\mathcal{Z}^\perp}(\boldsymbol{\Gamma}(0))$$

*as* $t \to \infty$. *In particular, if* $\mathbf{H}$ *is positive definite, the gradient flow converges to the unique global minimizer of ICL risk over* GD-$\boldsymbol{\beta}$ *class, that is,* $\boldsymbol{\beta}(t) \to \boldsymbol{\beta}^*$ *and* $\boldsymbol{\Gamma}(t) \to \boldsymbol{\Gamma}^*$ *as* $t \to \infty$.

The proof, as well as the convergence rate, is deferred to Appendix G. We remark that (6.1) is a complex dynamical system on a non-convex potential function of $\boldsymbol{\beta}$ and $\boldsymbol{\Gamma}$. We briefly discuss our proof techniques assuming that $\mathbf{H}$ is full rank. The rank-deficient cases can be handled in the same way by applying appropriate project operators. To conquer the non-convex optimization issue, we observe that for every fixed $\boldsymbol{\Gamma}$, the potential as a function of $\boldsymbol{\beta}$ is smooth and strongly convex with a uniformly bounded condition number. This observation allows us to establish a uniform convergence for $\boldsymbol{\beta}$. When $\boldsymbol{\beta}$ is sufficiently close to $\boldsymbol{\beta}^*$, the potential as a function of $\boldsymbol{\Gamma}$ is approximately convex, allowing us to track the convergence of $\boldsymbol{\Gamma}$.

Theorem 6.3 shows that optimization of GD-$\boldsymbol{\beta}$ can be done efficiently by gradient flow without suffering from non-convexity. However, as we have shown in previous sections, LTB utilizes a more complex parameterization than GD-$\boldsymbol{\beta}$. So Theorem 6.3 does not imply optimization of LTB is easy. We leave it as future work to study the optimization and statistical complexity for directly learning LTB models.

## 7 Concluding Remarks

In this paper, we study the in-context learning of linear regression with a shared signal represented by a Gaussian prior with a non-zero mean. We show that although the linear self-attention layer

discussed in prior works is consistent for this more complex task, its risk has an inevitable gap compared to that of the linear Transformer block (LTB), which is a linear self-attention layer followed by a linear multi-layer perception (MLP) layer. Next, we show that the effectiveness of the LTB arises because it can implement the one-step gradient descent estimator with learnable initialization (GD-$\beta$). Moreover, all global minimizers in the LTB class are equivalent to the unique global minimizer in the GD-$\beta$ class, which can achieve nearly Bayes optimal in-context learning risk. Finally, we consider training on in-context examples and prove global convergence over the GD-$\beta$ class of gradient flow on the population loss. Several future directions are worth discussing.

**Optimization and statistical complexity.** This paper provides an approximation theory of LTB and an optimization theory of GD-$\beta$. However, the statistical complexity of learning LTB or GD-$\beta$ is not considered. The work by [37] provided techniques for analyzing the statistical task complexity for pre-training GD-0. An interesting direction is to extend their method to study the statistical complexity of learning GD-$\beta$. However, their method crucially relies on the convexity of the risk induced GD-0, while we have shown that the risk induced by GD-$\beta$ is non-convex. New ideas for dealing with non-convexity are needed here.

**From LTB to Transformer block.** We focus on LTB in this work, which simplifies a vanilla Transformer block by removing the non-linearities from the softmax self-attention and the ReLU activation in the MLP layers. This simplification allows us to obtain precise theoretical results for LTB (such as its connection to GD-$\beta$). On the other hand, non-linearities are arguably necessary for Transformers to work well in practice. An important next step is to further consider the theoretical benefits of non-linearities based on our current results.

**Roles of MLP layers.** The work by [15] (and references thereafter) empirically found that MLP layers operate as key-value memories that store human-interpretable patterns in some pre-trained Transformers. Their work motivated a method for locating and editing information stored in language models by modifying their MLP layers (see, e.g., [9, 23]). Our work proves that the MLP component enables LTB to learn the shared signal in linear regression tasks, which cannot be done by a single LSA component. We leave it as future work to theoretically clarify the information stored in the MLP component.

## Acknowledgments

We gratefully acknowledge the support of the NSF and the Simons Foundation for the Collaboration on the Theoretical Foundations of Deep Learning through awards DMS-2031883 and #814639, and of the NSF through grant DMS-2023505.

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

# A  Notation and Variable Transformation

In order to simplify the proof, let's first describe a variable transformation scheme for our model and data coming from Assumption 3.1.

**Definition A.1** (Variable Transformation). We fix $M$ as the length of the contexts. Recall that $\mathbf{X}, \mathbf{x}$ are, respectively, the features in the context and the query input, while $\widetilde{\boldsymbol{\beta}}$ and $\boldsymbol{\beta}^*$ are the true task parameter for the inference prompt and its expectation, respectively. Following the Assumption 3.1, we have $\widetilde{\boldsymbol{\beta}} \sim \mathcal{N}\left(\boldsymbol{\beta}^*, \boldsymbol{\Psi}\right)$. From the definition for multivariate Gaussian distribution, we know there exists a random vector $\widetilde{\boldsymbol{\theta}}$ such that

$$\widetilde{\boldsymbol{\beta}} = \boldsymbol{\beta}^* + \boldsymbol{\Psi}^{\frac{1}{2}}\widetilde{\boldsymbol{\theta}}, \tag{A.1}$$

where

$$\widetilde{\boldsymbol{\theta}} \sim \mathcal{N}\left(0, \mathbf{I}_d\right). \tag{A.2}$$

Recall the noise vector is defined as $\boldsymbol{\varepsilon} = \mathbf{y} - \mathbf{X}\boldsymbol{\beta}$ and is generated from $\boldsymbol{\varepsilon} \sim \mathcal{N}\left(\mathbf{0}, \sigma^2 \cdot \mathbf{I}_M\right)$.

**Rank deficient case**   Note that, even when $\boldsymbol{\Psi}$ is rank-deficient, the variable transformation in (A.1) and (A.2) still hold. This can be seen from the definition of the multivariate Gaussian distribution (Def 20.11 and the discussion below in [31]): A vector $\widetilde{\boldsymbol{\beta}} \in \mathbb{R}^d$ with mean $\boldsymbol{\beta}^*$ and covariance matrix $\boldsymbol{\Psi}$ has a multivariate normal distribution, if it has the same distribution as $\boldsymbol{A}\widetilde{\boldsymbol{\theta}} + \boldsymbol{\beta}^*$ where $\boldsymbol{A}$ is any $d \times m$ matrix satisfying $\boldsymbol{A}\boldsymbol{A}^\top = \boldsymbol{\Psi}$ and $\widetilde{\boldsymbol{\theta}} \sim \mathcal{N}\left(\mathbf{0}_m, \mathbf{I}_m\right)$. Here, we can recover the variable transformation above if we take $m = d$ and $\boldsymbol{A} = \boldsymbol{\Psi}^{\frac{1}{2}}$.

**Notation**   Before we delve into the detailed proof, let's repeat the notation part with some additional notations. We use lowercase bold letters to note vectors and uppercase bold letters to denote matrices and tensors. For a vector $\mathbf{x}$ and a positive semi-definite (PSD) matrix $\boldsymbol{A}$, we denote $\|x\|_{\boldsymbol{A}} := \sqrt{\mathbf{x}^\top \boldsymbol{A}\mathbf{x}}$. We denote $\langle \cdot, \cdot \rangle$ as the inner product, which is defined as $\langle \mathbf{x}, \mathbf{y} \rangle := \mathbf{x}^\top \mathbf{y}$ for vectors and $\langle \boldsymbol{A}, \boldsymbol{B} \rangle := \mathrm{tr}\left(\boldsymbol{A}\boldsymbol{B}^\top\right)$ for matrices. For a matrix $\boldsymbol{A}$, we denote $\|\boldsymbol{A}\|_{op}, \|\boldsymbol{A}\|_F$ as the operator norm and the Frobenius norm, respectively. We denote $\boldsymbol{A}[i]$ as the $i$-th row of the matrix $\boldsymbol{A}$, $\boldsymbol{A}_{m,n}$ as the $(m, n)$-th entry, and $\boldsymbol{A}_{-1,-1}$ as the right-bottom entry. We denote $\mathbf{0}_n, \mathbf{0}_{m \times n}, \mathbf{I}_n$ as the zero vector, zero matrix and identity matrix, respectively.

For a positive semi-definite matrix $\boldsymbol{A}$, we denote $\boldsymbol{A}^{\frac{1}{2}}$ for the principle square root of $\boldsymbol{A}$, which is defined as the unique real matrix $\boldsymbol{B}$ such that $\boldsymbol{B}$ is positive semi-definite and $\boldsymbol{B}\boldsymbol{B} = \boldsymbol{B}\boldsymbol{B}^\top = \boldsymbol{A}$. For positive definite $\boldsymbol{A}$, its principle square root is also positive definite. We denote $\boldsymbol{A}^+$ as the Moore-Penrose pseudo-inverse for any matrix $\boldsymbol{A}$. We also denote $\boldsymbol{A}^{-\frac{1}{2}} = \left(\boldsymbol{A}^{\frac{1}{2}}\right)^+$. We denote $\otimes$ as the Kronecker product. For compatible matrices of proper size $\boldsymbol{A}, \boldsymbol{B}, \boldsymbol{C}, \boldsymbol{B}^\top \otimes \boldsymbol{A}$ is a linear mapping which is defined by $\left(\boldsymbol{B}^\top \otimes \boldsymbol{A}\right) \circ \boldsymbol{C} = \boldsymbol{A}\boldsymbol{C}\boldsymbol{B}$. We denote $\boldsymbol{\Lambda} := \boldsymbol{\Psi}^{\frac{1}{2}}\mathbf{H}\boldsymbol{\Psi}^{\frac{1}{2}}$ and $\phi_1 \geq \phi_2 \geq ... \geq \phi_d \geq 0$ are its ordered eigenvalues. We also denote $\lambda_1 \geq \lambda_2 \geq ... \geq \lambda_d \geq 0$ as the ordered eigenvalues of $\mathbf{H}$, and $\lambda_{-1} > 0$ as its minimal positive eigenvalue. Finally, we denote another important matrix

$$\boldsymbol{\Omega} := \frac{M+1}{M}\mathbf{H}^{\frac{1}{2}}\boldsymbol{\Psi}\mathbf{H}^{\frac{1}{2}} + \frac{\mathrm{tr}\left(\mathbf{H}\boldsymbol{\Psi}\right) + \sigma^2}{M} \cdot \mathbf{I}_d. \tag{A.3}$$

Note that, under this definition and our assumption on $\mathbf{H}$ and $\boldsymbol{\Psi}$, we have $\boldsymbol{\Omega}$ is invertible.

# B Proof of Theorem 4.1

*Proof.* The fact that $\inf_{f \in \mathcal{F}_{\mathsf{LTB}}} \mathcal{R}(f)$ does not depend on the vector $\boldsymbol{\beta}^*$ is subsumed in the Theorem 5.3, so we do not prove it here. We are going to prove the inequality in the theorem. First, from Theorem 5.3 we know that,

$$\inf_{f \in \mathcal{F}_{\mathsf{LTB}}} \mathcal{R}(f) = \sigma^2 + \mathrm{tr}\left(\mathbf{H}\boldsymbol{\Psi}\right) - \mathrm{tr}\left(\left(\mathbf{H}^{\frac{1}{2}}\boldsymbol{\Psi}\mathbf{H}^{\frac{1}{2}}\right)^2 \boldsymbol{\Omega}^{-1}\right), \tag{B.1}$$

where $\boldsymbol{\Omega}$ is defined in (A.3). Then, it suffices to lower bound $\inf_{f \in \mathcal{F}_{\mathsf{LSA}}} \mathcal{R}(f)$. So let's take an arbitrary $f \in \mathcal{F}_{\mathsf{LSA}}$, which is denoted as

$$f(\mathbf{E}) = \left[\mathbf{E} + \mathbf{W}_P^\top \mathbf{W}_V \mathbf{E}\mathbf{M}\frac{\mathbf{E}^\top \mathbf{W}_K^\top \mathbf{W}_Q \mathbf{E}}{M}\right]_{-1,-1},$$

where $\mathbf{E}$ is the token matrix defined in (3.1). Since the prediction is the right-bottom entry of the output matrix, we know only the last row of the product $\mathbf{W}_P^\top \mathbf{W}_V$ attends the prediction. Similarly, only the last column of the $\mathbf{E}$ on the far right in the above equation atttends the prediction. Since the last column of $\mathbf{E}$ is $\left(\mathbf{x}^\top \ 0\right)^\top$, we know that only the first $d$ rows of the product $\mathbf{W}_K^\top \mathbf{W}_Q$ enter the calculation (since other parts are multipled by zero). Therefore, we denote

$$\mathbf{W}_P^\top \mathbf{W}_V = \begin{pmatrix} * & * \\ \mathbf{u}_{21}^\top & u_{-1} \end{pmatrix}, \quad \mathbf{W}_K^\top \mathbf{W}_Q = \begin{pmatrix} \mathbf{U}_{11} & * \\ \mathbf{u}_{12}^\top & * \end{pmatrix},$$

where $\mathbf{U}_{11} \in \mathbb{R}^{d \times d}, \mathbf{u}_{12}, \mathbf{u}_{21} \in \mathbb{R}^{d \times 1}, u_{-1} \in \mathbb{R}$, and $*$ denotes entries that do not enter the final prediction. The model prediction can be written as

$$f(\mathbf{E}) = \begin{pmatrix} \mathbf{u}_{21}^\top & u_{-1} \end{pmatrix} \cdot \frac{\mathbf{E}\mathbf{M}_M \mathbf{E}^\top}{M} \cdot \begin{pmatrix} \mathbf{U}_{11} \\ \mathbf{u}_{12}^\top \end{pmatrix} \cdot \mathbf{x}$$

$$= \left[\mathbf{u}_{21}^\top \cdot \frac{1}{M}\mathbf{X}^\top \mathbf{X} \cdot \mathbf{U}_{11} + \mathbf{u}_{21}^\top \cdot \frac{1}{M}\mathbf{X}^\top \mathbf{y} \cdot \mathbf{u}_{12}^\top + u_{-1} \cdot \frac{1}{M}\mathbf{y}^\top \mathbf{X} \cdot \mathbf{U}_{11} + u_{-1} \cdot \frac{1}{M}\mathbf{y}^\top \mathbf{y} \cdot \mathbf{u}_{12}^\top\right] \cdot \mathbf{x}.$$

**Step 1: simplify the risk function.** We use $\widetilde{\boldsymbol{\beta}}$ to denote the task parameter. From the Assumption 3.1 and Definition A.1, we have

$$\mathbf{y} = \mathbf{X}\widetilde{\boldsymbol{\beta}} + \boldsymbol{\varepsilon}, \quad y = \left\langle \widetilde{\boldsymbol{\beta}}, \mathbf{x} \right\rangle + \varepsilon, \quad \widetilde{\boldsymbol{\beta}} \sim \mathcal{N}\left(\boldsymbol{\beta}^*, \boldsymbol{\Psi}\right), \quad \widetilde{\boldsymbol{\beta}} = \boldsymbol{\beta}^* + \boldsymbol{\Psi}^{\frac{1}{2}}\widetilde{\boldsymbol{\theta}};$$

and

$$\mathbf{X}[i], \mathbf{x} \overset{\text{i.i.d.}}{\sim} \mathcal{N}\left(\mathbf{0}, \mathbf{H}\right), \quad \boldsymbol{\varepsilon}[i], \varepsilon \overset{\text{i.i.d.}}{\sim} \mathcal{N}\left(0, \sigma^2\right), \quad \widetilde{\boldsymbol{\theta}} \sim \mathcal{N}\left(\mathbf{0}, \mathbf{I}_d\right).$$

Then the model output can be written as

$$f(\mathbf{E}) = \left[\mathbf{u}_{21}^\top \cdot \frac{1}{M}\mathbf{X}^\top \mathbf{X} \cdot \mathbf{U}_{11} + \mathbf{u}_{21}^\top \cdot \frac{1}{M}\mathbf{X}^\top \mathbf{y} \cdot \mathbf{u}_{12}^\top + u_{-1} \cdot \frac{1}{M}\mathbf{y}^\top \mathbf{X} \cdot \mathbf{U}_{11} + u_{-1} \cdot \frac{1}{M}\mathbf{y}^\top \mathbf{y} \cdot \mathbf{u}_{12}^\top\right] \cdot \mathbf{x}$$

$$= \left[\left(\mathbf{u}_{21} + u_{-1}\widetilde{\boldsymbol{\beta}}\right)^\top \cdot \frac{1}{M}\mathbf{X}^\top \mathbf{X} \cdot \left(\mathbf{U}_{11} + \widetilde{\boldsymbol{\beta}}\mathbf{u}_{12}^\top\right)\right] \cdot \mathbf{x}$$

$$+ \left[\mathbf{u}_{21}^\top \cdot \frac{1}{M}\mathbf{X}^\top \boldsymbol{\varepsilon} \cdot \mathbf{u}_{12}^\top + u_{-1} \cdot \frac{1}{M}\boldsymbol{\varepsilon}^\top \mathbf{X} \cdot \mathbf{U}_{11} + u_{-1} \cdot \frac{2}{M}\boldsymbol{\varepsilon}^\top \mathbf{X}\widetilde{\boldsymbol{\beta}}\mathbf{u}_{12}^\top + \frac{1}{M}\boldsymbol{\varepsilon}^\top \boldsymbol{\varepsilon} \cdot u_{-1}\mathbf{u}_{12}^\top\right] \cdot \mathbf{x}.$$

To simplify the presentation, we denote

$$\boldsymbol{z}_1^\top = \left(\mathbf{u}_{21} + u_{-1}\widetilde{\boldsymbol{\beta}}\right)^\top \cdot \frac{1}{M}\mathbf{X}^\top \mathbf{X} \cdot \left(\mathbf{U}_{11} + \widetilde{\boldsymbol{\beta}}\mathbf{u}_{12}^\top\right),$$

$$\boldsymbol{z}_2^\top = \mathbf{u}_{21}^\top \cdot \frac{1}{M}\mathbf{X}^\top \boldsymbol{\varepsilon} \cdot \mathbf{u}_{12}^\top + u_{-1} \cdot \frac{1}{M}\boldsymbol{\varepsilon}^\top \mathbf{X} \cdot \mathbf{U}_{11} + u_{-1} \cdot \frac{2}{M}\boldsymbol{\varepsilon}^\top \mathbf{X}\widetilde{\boldsymbol{\beta}}\mathbf{u}_{12}^\top$$

$$\boldsymbol{z}_3^\top = \frac{1}{M}\boldsymbol{\varepsilon}^\top \boldsymbol{\varepsilon} \cdot u_{-1}\mathbf{u}_{12}^\top.$$

Since $\mathbf{x}, \mathbf{X}, \varepsilon, \widetilde{\boldsymbol{\beta}}$ are independent, we have

$$
\begin{aligned}
\mathcal{R}\left(f\right) &= \mathbb{E}\left(f(\mathbf{E}) - \left\langle\widetilde{\boldsymbol{\beta}}, \mathbf{x}\right\rangle - \varepsilon\right)^2 \\
&= \sigma^2 + \mathbb{E}\left(f(\mathbf{E}) - \left\langle\widetilde{\boldsymbol{\beta}}, \mathbf{x}\right\rangle\right)^2 \quad (\varepsilon \text{ is independent from other variables and zero-mean}) \\
&= \mathbb{E}\left[\left\langle\boldsymbol{z}_1 + \boldsymbol{z}_2 + \boldsymbol{z}_3 - \widetilde{\boldsymbol{\beta}}, \mathbf{x}\right\rangle^2\right] + \sigma^2 \\
&= \left\langle\mathbf{H}, \mathbb{E}\left(\boldsymbol{z}_1 + \boldsymbol{z}_2 + \boldsymbol{z}_3 - \widetilde{\boldsymbol{\beta}}\right)\left(\boldsymbol{z}_1 + \boldsymbol{z}_2 + \boldsymbol{z}_3 - \widetilde{\boldsymbol{\beta}}\right)^\top\right\rangle + \sigma^2.
\end{aligned}
$$

Note that $\boldsymbol{z}_1$ does not contain $\varepsilon$, $\boldsymbol{z}_2$ is a linear form of $\varepsilon$, and $\boldsymbol{z}_3$ is a quadratic form of $\varepsilon$. Using $\varepsilon \sim \mathcal{N}(\mathbf{0}, \sigma^2 \mathbf{I}_d)$, we have $\mathbb{E}[(\boldsymbol{z}_1 - \widetilde{\boldsymbol{\beta}}) \cdot \boldsymbol{z}_2^\top] = \mathbf{0}$ and $\mathbb{E}[\boldsymbol{z}_2 \boldsymbol{z}_3^\top] = \mathbf{0}$. Therefore, we have

$$
\mathcal{R}\left(f\right) - \sigma^2 = \underbrace{\left\langle\mathbf{H}, \mathbb{E}\left(\boldsymbol{z}_1 - \widetilde{\boldsymbol{\beta}}\right)\left(\boldsymbol{z}_1 - \widetilde{\boldsymbol{\beta}}\right)^\top\right\rangle}_{S_1} + \underbrace{\left\langle\mathbf{H}, \mathbb{E}\boldsymbol{z}_2\boldsymbol{z}_2^\top\right\rangle}_{S_2} + \underbrace{\left\langle\mathbf{H}, \mathbb{E}\boldsymbol{z}_3\boldsymbol{z}_3^\top\right\rangle}_{S_3}
$$

$$
+ \underbrace{2\left\langle\mathbf{H}, \mathbb{E}\left(\boldsymbol{z}_1 - \widetilde{\boldsymbol{\beta}}\right)\boldsymbol{z}_3^\top\right\rangle}_{S_4}. \tag{B.2}
$$

**Step 2: compute $S_1$.** By Lemma H.4 and that $\mathbf{X}$ is independent of all other random variables, we have

$$
\begin{aligned}
&\left\langle\mathbf{H}, \mathbb{E}\boldsymbol{z}_1\boldsymbol{z}_1^\top\right\rangle \\
=&\mathbb{E}\mathrm{tr}\left[\left(\mathbf{U}_{11} + \widetilde{\boldsymbol{\beta}}\mathbf{u}_{12}^\top\right)^\top \cdot \frac{1}{M}\mathbf{X}^\top\mathbf{X} \cdot \left(\mathbf{u}_{21} + u_{-1}\widetilde{\boldsymbol{\beta}}\right)\left(\mathbf{u}_{21} + u_{-1}\widetilde{\boldsymbol{\beta}}\right)^\top \cdot \frac{1}{M}\mathbf{X}^\top\mathbf{X} \cdot \left(\mathbf{U}_{11} + \widetilde{\boldsymbol{\beta}}\mathbf{u}_{12}^\top\right)\mathbf{H}\right] \\
=&\frac{M+1}{M}\mathbb{E}_{\widetilde{\boldsymbol{\beta}}}\mathrm{tr}\left[\left(\mathbf{U}_{11} + \widetilde{\boldsymbol{\beta}}\mathbf{u}_{12}^\top\right)^\top \cdot \mathbf{H} \cdot \left(\mathbf{u}_{21} + u_{-1}\widetilde{\boldsymbol{\beta}}\right)\left(\mathbf{u}_{21} + u_{-1}\widetilde{\boldsymbol{\beta}}\right)^\top \cdot \mathbf{H} \cdot \left(\mathbf{U}_{11} + \widetilde{\boldsymbol{\beta}}\mathbf{u}_{12}^\top\right)\mathbf{H}\right] \\
&+\frac{1}{M}\mathbb{E}_{\widetilde{\boldsymbol{\beta}}}\mathrm{tr}\left[\mathrm{tr}\left(\left(\mathbf{u}_{21} + u_{-1}\widetilde{\boldsymbol{\beta}}\right)\left(\mathbf{u}_{21} + u_{-1}\widetilde{\boldsymbol{\beta}}\right)^\top\mathbf{H}\right)\left(\mathbf{U}_{11} + \widetilde{\boldsymbol{\beta}}\mathbf{u}_{12}^\top\right)^\top\mathbf{H}\left(\mathbf{U}_{11} + \widetilde{\boldsymbol{\beta}}\mathbf{u}_{12}^\top\right)\mathbf{H}\right].
\end{aligned}
$$

We denote

$$
\boldsymbol{b} := \mathbf{u}_{21} + u_{-1}\boldsymbol{\beta}^* \in \mathbb{R}^d, \quad \boldsymbol{A} := \mathbf{U}_{11} + \boldsymbol{\beta}^*\mathbf{u}_{12}^\top \in \mathbb{R}^{d\times d}, \tag{B.3}
$$

then applying $\widetilde{\boldsymbol{\beta}} = \boldsymbol{\beta}^* + \boldsymbol{\Psi}^{\frac{1}{2}}\widetilde{\boldsymbol{\theta}}$, we get

$$
\begin{aligned}
&\left\langle\mathbf{H}, \mathbb{E}\boldsymbol{z}_1\boldsymbol{z}_1^\top\right\rangle \\
=&\frac{M+1}{M}\mathbb{E}_{\widetilde{\boldsymbol{\theta}}\sim\mathcal{N}(\mathbf{0},\mathbf{I}_d)}\mathrm{tr}\left[\left(\boldsymbol{A} + \boldsymbol{\Psi}^{\frac{1}{2}}\widetilde{\boldsymbol{\theta}}\mathbf{u}_{12}^\top\right)^\top \cdot \mathbf{H} \cdot \left(\boldsymbol{b} + u_{-1}\boldsymbol{\Psi}^{\frac{1}{2}}\widetilde{\boldsymbol{\theta}}\right)\left(\boldsymbol{b} + u_{-1}\boldsymbol{\Psi}^{\frac{1}{2}}\widetilde{\boldsymbol{\theta}}\right)^\top \cdot \mathbf{H} \cdot \left(\boldsymbol{A} + \boldsymbol{\Psi}^{\frac{1}{2}}\widetilde{\boldsymbol{\theta}}\mathbf{u}_{12}^\top\right)\mathbf{H}\right] \\
&+\frac{1}{M}\mathbb{E}_{\widetilde{\boldsymbol{\theta}}\sim\mathcal{N}(\mathbf{0},\mathbf{I}_d)}\mathrm{tr}\left(\left(\boldsymbol{b} + u_{-1}\boldsymbol{\Psi}^{\frac{1}{2}}\widetilde{\boldsymbol{\theta}}\right)\left(\boldsymbol{b} + u_{-1}\boldsymbol{\Psi}^{\frac{1}{2}}\widetilde{\boldsymbol{\theta}}\right)^\top\mathbf{H}\right)\mathrm{tr}\left[\left(\boldsymbol{A} + \boldsymbol{\Psi}^{\frac{1}{2}}\widetilde{\boldsymbol{\theta}}\mathbf{u}_{12}^\top\right)^\top\mathbf{H}\left(\boldsymbol{A} + \boldsymbol{\Psi}^{\frac{1}{2}}\widetilde{\boldsymbol{\theta}}\mathbf{u}_{12}^\top\right)\mathbf{H}\right].
\end{aligned}
$$

Note that the first and third moments of $\widetilde{\boldsymbol{\theta}}$ are zero. So the above equation only involves the zeroth, second, and fourth moments of $\widetilde{\boldsymbol{\theta}}$. Therefore we have

$$
\left\langle\mathbf{H}, \mathbb{E}\boldsymbol{z}_1\boldsymbol{z}_1^\top\right\rangle = \underbrace{\frac{M+1}{M}\mathrm{tr}\left(\boldsymbol{A}^\top\mathbf{H}\boldsymbol{b}\boldsymbol{b}^\top\mathbf{H}\boldsymbol{A}\mathbf{H}\right) + \frac{1}{M}\mathrm{tr}\left(\boldsymbol{b}\boldsymbol{b}^\top\mathbf{H}\right)\mathrm{tr}\left(\boldsymbol{A}^\top\mathbf{H}\boldsymbol{A}\mathbf{H}\right)}_{\text{The leading term}} + T_2 + T_4,
$$

$$\tag{B.4}$$

where $T_2$ and $T_4$ are the second order and the fourth order term, respectively. More concretely, the second order term is

$$
T_2 = \frac{M+1}{M}\mathbb{E}\mathrm{tr}\left\{u_{-1}\boldsymbol{A}^\top\mathbf{H}\boldsymbol{b}\widetilde{\boldsymbol{\theta}}^\top\boldsymbol{\Psi}^{\frac{1}{2}}\mathbf{H}\boldsymbol{\Psi}^{\frac{1}{2}}\widetilde{\boldsymbol{\theta}}\mathbf{u}_{12}^\top\mathbf{H} + u_{-1}\mathbf{u}_{12}\widetilde{\boldsymbol{\theta}}^\top\boldsymbol{\Psi}^{\frac{1}{2}}\mathbf{H}\boldsymbol{\Psi}^{\frac{1}{2}}\widetilde{\boldsymbol{\theta}}\boldsymbol{b}^\top\mathbf{H}\boldsymbol{A}\mathbf{H}\right.
$$

$$+ u_{-1}\mathbf{u}_{12}\widetilde{\boldsymbol{\theta}}^\top \boldsymbol{\Psi}^{\frac{1}{2}}\mathbf{H}b\widetilde{\boldsymbol{\theta}}^\top \boldsymbol{\Psi}^{\frac{1}{2}}\mathbf{H}\boldsymbol{A}\mathbf{H} + u_{-1}\boldsymbol{A}^\top\mathbf{H}\boldsymbol{\Psi}^{\frac{1}{2}}\widetilde{\boldsymbol{\theta}}b^\top\mathbf{H}\boldsymbol{\Psi}^{\frac{1}{2}}\widetilde{\boldsymbol{\theta}}\mathbf{u}_{12}^\top\mathbf{H}$$

$$+ \mathbf{u}_{12}\widetilde{\boldsymbol{\theta}}^\top\boldsymbol{\Psi}^{\frac{1}{2}}\mathbf{H}bb^\top\mathbf{H}\boldsymbol{\Psi}^{\frac{1}{2}}\widetilde{\boldsymbol{\theta}}\mathbf{u}_{12}^\top\mathbf{H} + u_{-1}^2\boldsymbol{A}^\top\mathbf{H}\boldsymbol{\Psi}^{\frac{1}{2}}\widetilde{\boldsymbol{\theta}}\widetilde{\boldsymbol{\theta}}^\top\boldsymbol{\Psi}^{\frac{1}{2}}\mathbf{H}\boldsymbol{A}\mathbf{H}\Big\}$$

$$+ \frac{1}{M}\mathbb{E}\Big\{u_{-1}^2\widetilde{\boldsymbol{\theta}}^\top\boldsymbol{\Psi}^{\frac{1}{2}}\mathbf{H}\boldsymbol{\Psi}^{\frac{1}{2}}\widetilde{\boldsymbol{\theta}}\cdot\mathrm{tr}\left(\boldsymbol{A}^\top\mathbf{H}\boldsymbol{A}\mathbf{H}\right) + b^\top\mathbf{H}b\cdot\mathrm{tr}\left(\mathbf{u}_{12}\widetilde{\boldsymbol{\theta}}^\top\boldsymbol{\Psi}^{\frac{1}{2}}\mathbf{H}\boldsymbol{\Psi}^{\frac{1}{2}}\widetilde{\boldsymbol{\theta}}\mathbf{u}_{12}^\top\mathbf{H}\right)$$

$$+ 2u_{-1}b^\top\mathbf{H}\boldsymbol{\Psi}^{\frac{1}{2}}\widetilde{\boldsymbol{\theta}}\cdot\mathrm{tr}\left(\mathbf{u}_{12}\widetilde{\boldsymbol{\theta}}^\top\boldsymbol{\Psi}^{\frac{1}{2}}\mathbf{H}\boldsymbol{A}\mathbf{H}\right) + 2u_{-1}b^\top\mathbf{H}\boldsymbol{\Psi}^{\frac{1}{2}}\widetilde{\boldsymbol{\theta}}\cdot\mathrm{tr}\left(\boldsymbol{A}^\top\mathbf{H}\boldsymbol{\Psi}^{\frac{1}{2}}\widetilde{\boldsymbol{\theta}}\mathbf{u}_{12}^\top\mathbf{H}\right)\Big\}$$

$$= \frac{M+1}{M}\Big\{2u_{-1}\mathrm{tr}(\boldsymbol{\Psi}^{\frac{1}{2}}\mathbf{H}\boldsymbol{\Psi}^{\frac{1}{2}})\cdot\mathbf{u}_{12}^\top\mathbf{H}\boldsymbol{A}^\top\mathbf{H}b + 2u_{-1}b^\top\mathbf{H}\boldsymbol{\Psi}\mathbf{H}\boldsymbol{A}\mathbf{H}\mathbf{u}_{12} + b^\top\mathbf{H}\boldsymbol{\Psi}\mathbf{H}b\cdot\mathbf{u}_{12}^\top\mathbf{H}\mathbf{u}_{12}$$

$$+ u_{-1}^2\mathrm{tr}\left(\boldsymbol{A}^\top\mathbf{H}\boldsymbol{\Psi}\mathbf{H}\boldsymbol{A}\mathbf{H}\right)\Big\} + \frac{1}{M}\Big\{u_{-1}^2\mathrm{tr}\left(\boldsymbol{\Psi}^{\frac{1}{2}}\mathbf{H}\boldsymbol{\Psi}^{\frac{1}{2}}\right)\cdot\mathrm{tr}\left(\boldsymbol{A}^\top\mathbf{H}\boldsymbol{A}\mathbf{H}\right)$$

$$+ \mathrm{tr}\left(\boldsymbol{\Psi}^{\frac{1}{2}}\mathbf{H}\boldsymbol{\Psi}^{\frac{1}{2}}\right)\cdot b^\top\mathbf{H}b\cdot\mathbf{u}_{12}^\top\mathbf{H}\mathbf{u}_{12} + 4u_{-1}\cdot b^\top\mathbf{H}\boldsymbol{\Psi}^{\frac{1}{2}}\mathbf{H}\boldsymbol{A}\mathbf{H}\mathbf{u}_{12}\Big\}$$

$$= \frac{2(M+1)}{M}u_{-1}b^\top\left[\mathrm{tr}(\mathbf{H}\boldsymbol{\Psi})\mathbf{H} + \mathbf{H}\boldsymbol{\Psi}\mathbf{H}\right]\boldsymbol{A}\mathbf{H}\mathbf{u}_{12} + b^\top\left(\frac{M+1}{M}\mathbf{H}\boldsymbol{\Psi}\mathbf{H} + \frac{1}{M}\mathrm{tr}(\mathbf{H}\boldsymbol{\Psi})\mathbf{H}\right)b\cdot\mathbf{u}_{12}^\top\mathbf{H}\mathbf{u}_{12}$$

$$+ u_{-1}^2\mathrm{tr}\left(\boldsymbol{A}^\top\left(\frac{M+1}{M}\mathbf{H}\boldsymbol{\Psi}\mathbf{H} + \frac{1}{M}\mathrm{tr}(\mathbf{H}\boldsymbol{\Psi})\mathbf{H}\right)\boldsymbol{A}\mathbf{H}\right) + \frac{4}{M}u_{-1}\cdot b^\top\mathbf{H}\boldsymbol{\Psi}\mathbf{H}\boldsymbol{A}\mathbf{H}\mathbf{u}_{12}. \quad \text{(B.5)}$$

The fourth order term is

$$T_4 = \frac{M+1}{M}\mathbb{E}\mathrm{tr}\Big\{u_{-1}^2\mathbf{u}_{12}\widetilde{\boldsymbol{\theta}}^\top\boldsymbol{\Psi}^{\frac{1}{2}}\mathbf{H}\boldsymbol{\Psi}^{\frac{1}{2}}\widetilde{\boldsymbol{\theta}}\widetilde{\boldsymbol{\theta}}^\top\boldsymbol{\Psi}^{\frac{1}{2}}\mathbf{H}\boldsymbol{\Psi}^{\frac{1}{2}}\widetilde{\boldsymbol{\theta}}\mathbf{u}_{12}^\top\mathbf{H}\Big\}$$

$$+ \frac{1}{M}\mathbb{E}\mathrm{tr}\Big\{u_{-1}^2\widetilde{\boldsymbol{\theta}}^\top\boldsymbol{\Psi}^{\frac{1}{2}}\mathbf{H}\boldsymbol{\Psi}^{\frac{1}{2}}\widetilde{\boldsymbol{\theta}}\cdot\mathrm{tr}\left(\mathbf{u}_{12}\widetilde{\boldsymbol{\theta}}^\top\boldsymbol{\Psi}^{\frac{1}{2}}\mathbf{H}\boldsymbol{\Psi}^{\frac{1}{2}}\widetilde{\boldsymbol{\theta}}\mathbf{u}_{12}^\top\mathbf{H}\right)\Big\}$$

$$= \frac{M+2}{M}u_{-1}^2\mathbf{u}_{12}^\top\mathbf{H}\mathbf{u}_{12}\cdot\mathbb{E}\left(\widetilde{\boldsymbol{\theta}}^\top\boldsymbol{\Psi}^{\frac{1}{2}}\mathbf{H}\boldsymbol{\Psi}^{\frac{1}{2}}\widetilde{\boldsymbol{\theta}}\right)^2$$

$$= \frac{M+2}{M}u_{-1}^2\cdot\left(2\mathrm{tr}\left(\mathbf{H}\boldsymbol{\Psi}\mathbf{H}\boldsymbol{\Psi}\right) + \mathrm{tr}\left(\mathbf{H}\boldsymbol{\Psi}\right)^2\right)\cdot\mathbf{u}_{12}^\top\mathbf{H}\mathbf{u}_{12}. \quad \text{(B.6)}$$

For the cross term in $S_1$, we have

$$\left\langle\mathbf{H}, \mathbb{E}\boldsymbol{z}_1\widetilde{\boldsymbol{\beta}}^\top\right\rangle$$

$$= \mathbb{E}\left\{\left(\mathbf{u}_{21} + u_{-1}\widetilde{\boldsymbol{\beta}}\right)^\top\cdot\frac{1}{M}\mathbf{X}^\top\mathbf{X}\cdot\left(\mathbf{U}_{11} + \widetilde{\boldsymbol{\beta}}\mathbf{u}_{12}^\top\right)\mathbf{H}\widetilde{\boldsymbol{\beta}}\right\} \quad \text{(B.7)}$$

$$= \mathbb{E}\left\{\left(\mathbf{u}_{21} + u_{-1}\widetilde{\boldsymbol{\beta}}\right)^\top\mathbf{H}\left(\mathbf{U}_{11} + \widetilde{\boldsymbol{\beta}}\mathbf{u}_{12}^\top\right)\mathbf{H}\widetilde{\boldsymbol{\beta}}\right\} \quad \text{(From the distribution of } \mathbf{X}\text{)}$$

$$= \mathbb{E}\left\{\left(b + u_{-1}\boldsymbol{\Psi}^{\frac{1}{2}}\widetilde{\boldsymbol{\theta}}\right)^\top\mathbf{H}\left(\boldsymbol{A} + \boldsymbol{\Psi}^{\frac{1}{2}}\widetilde{\boldsymbol{\theta}}\mathbf{u}_{12}^\top\right)\mathbf{H}\left(\boldsymbol{\beta}^* + \boldsymbol{\Psi}^{\frac{1}{2}}\widetilde{\boldsymbol{\theta}}\right)\right\} \quad \text{(By (B.3))}$$

$$= b^\top\mathbf{H}\boldsymbol{A}\mathbf{H}\boldsymbol{\beta}^* + \mathbb{E}\left\{u_{-1}\widetilde{\boldsymbol{\theta}}^\top\boldsymbol{\Psi}^{\frac{1}{2}}\mathbf{H}\boldsymbol{\Psi}^{\frac{1}{2}}\widetilde{\boldsymbol{\theta}}\mathbf{u}_{12}^\top\mathbf{H}\boldsymbol{\beta}^* + u_{-1}\widetilde{\boldsymbol{\theta}}^\top\boldsymbol{\Psi}^{\frac{1}{2}}\mathbf{H}\boldsymbol{A}\mathbf{H}\boldsymbol{\Psi}^{\frac{1}{2}}\widetilde{\boldsymbol{\theta}} + b^\top\mathbf{H}\boldsymbol{\Psi}^{\frac{1}{2}}\widetilde{\boldsymbol{\theta}}\mathbf{u}_{12}^\top\mathbf{H}\boldsymbol{\Psi}^{\frac{1}{2}}\widetilde{\boldsymbol{\theta}}\right\}$$

$$= b^\top\mathbf{H}\boldsymbol{A}\mathbf{H}\boldsymbol{\beta}^* + u_{-1}\mathrm{tr}\left(\mathbf{H}\boldsymbol{\Psi}\right)\cdot\mathbf{u}_{12}^\top\mathbf{H}\boldsymbol{\beta}^* + u_{-1}\mathrm{tr}\left(\mathbf{H}\boldsymbol{A}\mathbf{H}\boldsymbol{\Psi}\right) + \mathbf{u}_{12}^\top\mathbf{H}\boldsymbol{\Psi}\mathbf{H}b, \quad \text{(B.8)}$$

where the last row comes from the fact that $\widetilde{\boldsymbol{\theta}}\sim\mathcal{N}\left(\mathbf{0}, \mathbf{I}_d\right)$. Moreover, we have

$$\left\langle\mathbf{H}, \mathbb{E}\widetilde{\boldsymbol{\beta}}\widetilde{\boldsymbol{\beta}}^\top\right\rangle = \boldsymbol{\beta}^{*\top}\mathbf{H}\boldsymbol{\beta}^* + \mathrm{tr}\left(\mathbf{H}\boldsymbol{\Psi}\right). \quad \text{(B.9)}$$

Combining (B.4), (B.5), (B.6), (B.8), (B.9), we have

$$S_1 = \frac{M+1}{M}\mathrm{tr}\left(\boldsymbol{A}^\top\mathbf{H}bb^\top\mathbf{H}\boldsymbol{A}\mathbf{H}\right) + \frac{1}{M}\mathrm{tr}\left(bb^\top\mathbf{H}\right)\mathrm{tr}\left(\boldsymbol{A}^\top\mathbf{H}\boldsymbol{A}\mathbf{H}\right)$$

$$+ \frac{M+2}{M}u_{-1}^2\cdot\left(2\mathrm{tr}\left(\mathbf{H}\boldsymbol{\Psi}\mathbf{H}\boldsymbol{\Psi}\right) + \mathrm{tr}\left(\mathbf{H}\boldsymbol{\Psi}\right)^2\right)\cdot\mathbf{u}_{12}^\top\mathbf{H}\mathbf{u}_{12}$$

$$+ \frac{2(M+1)}{M} u_{-1} \boldsymbol{b}^\top \left[\mathrm{tr}(\mathbf{H}\boldsymbol{\Psi})\mathbf{H} + \mathbf{H}\boldsymbol{\Psi}\mathbf{H}\right] \boldsymbol{A}\mathbf{H}\mathbf{u}_{12}$$

$$+ u_{-1}^2 \mathrm{tr}\left(\boldsymbol{A}^\top \left(\frac{M+1}{M}\mathbf{H}\boldsymbol{\Psi}\mathbf{H} + \frac{1}{M}\mathrm{tr}(\mathbf{H}\boldsymbol{\Psi})\mathbf{H}\right)\boldsymbol{A}\mathbf{H}\right) + \frac{4}{M}u_{-1} \cdot \boldsymbol{b}^\top \mathbf{H}\boldsymbol{\Psi}\mathbf{H}\boldsymbol{A}\mathbf{H}\mathbf{u}_{12}$$

$$- 2\bigg[\boldsymbol{b}^\top \mathbf{H}\boldsymbol{A}\mathbf{H}\boldsymbol{\beta}^* + u_{-1}\mathrm{tr}\left(\mathbf{H}\boldsymbol{\Psi}\right)\cdot\mathbf{u}_{12}^\top \mathbf{H}\boldsymbol{\beta}^* + u_{-1}\mathrm{tr}\left(\mathbf{H}\boldsymbol{A}\mathbf{H}\boldsymbol{\Psi}\right)$$

$$+ \mathbf{u}_{12}^\top \mathbf{H}\boldsymbol{\Psi}\mathbf{H}\boldsymbol{b}\bigg] + \boldsymbol{\beta}^{*\top}\mathbf{H}\boldsymbol{\beta}^* + \mathrm{tr}\left(\mathbf{H}\boldsymbol{\Psi}\right) + \boldsymbol{b}^\top\left(\frac{M+1}{M}\mathbf{H}\boldsymbol{\Psi}\mathbf{H} + \frac{1}{M}\mathrm{tr}(\mathbf{H}\boldsymbol{\Psi})\mathbf{H}\right)\boldsymbol{b}\cdot\mathbf{u}_{12}^\top\mathbf{H}\mathbf{u}_{12}.$$
$$(\text{B.10})$$

**Step 3: other terms.** Let us compute $S_2, S_3$ and $S_4$. Using definitions, we rewrite $\boldsymbol{z}_2$ as

$$\boldsymbol{z}_2^\top = \mathbf{u}_{21}^\top \cdot \frac{1}{M}\mathbf{X}^\top\boldsymbol{\varepsilon}\cdot\mathbf{u}_{12}^\top + u_{-1}\cdot\frac{1}{M}\boldsymbol{\varepsilon}^\top\mathbf{X}\cdot\mathbf{U}_{11} + u_{-1}\cdot\frac{2}{M}\boldsymbol{\varepsilon}^\top\mathbf{X}\widetilde{\boldsymbol{\beta}}\mathbf{u}_{12}^\top$$

$$= \boldsymbol{b}^\top\cdot\frac{1}{M}\mathbf{X}^\top\boldsymbol{\varepsilon}\cdot\mathbf{u}_{12}^\top + u_{-1}\cdot\frac{1}{M}\boldsymbol{\varepsilon}^\top\mathbf{X}\cdot\boldsymbol{A} + u_{-1}\cdot\frac{2}{M}\boldsymbol{\varepsilon}^\top\mathbf{X}\boldsymbol{\Psi}^{\frac{1}{2}}\widetilde{\boldsymbol{\theta}}\mathbf{u}_{12}^\top. \qquad (\text{B.11})$$

Since $\boldsymbol{\varepsilon}\sim\mathcal{N}\left(\mathbf{0},\sigma^2\mathbf{I}_M\right), \widetilde{\boldsymbol{\theta}}\sim\mathcal{N}\left(\mathbf{0},\mathbf{I}_d\right)$ and they are independent, all terms in $S_2$ vanish except if the term contains even orders of $\widetilde{\boldsymbol{\theta}}$ or $\boldsymbol{\varepsilon}$. So we have

$$S_2 = \left\langle\mathbf{H},\mathbb{E}\boldsymbol{z}_2\boldsymbol{z}_2^\top\right\rangle = \frac{1}{M^2}\mathbb{E}\bigg\{\boldsymbol{b}^\top\mathbf{X}^\top\boldsymbol{\varepsilon}\cdot\mathbf{u}_{12}^\top\mathbf{H}\mathbf{u}_{12}\cdot\boldsymbol{\varepsilon}^\top\mathbf{X}\cdot\boldsymbol{b} + u_{-1}^2\boldsymbol{\varepsilon}^\top\mathbf{X}\boldsymbol{A}\mathbf{H}\boldsymbol{A}^\top\mathbf{X}^\top\boldsymbol{\varepsilon}$$

$$+ 2u_{-1}\cdot\boldsymbol{\varepsilon}^\top\mathbf{X}\boldsymbol{A}\mathbf{H}\mathbf{u}_{12}\cdot\boldsymbol{\varepsilon}^\top\mathbf{X}\boldsymbol{b} + 4u_{-1}^2\cdot\boldsymbol{\varepsilon}^\top\mathbf{X}\boldsymbol{\Psi}^{\frac{1}{2}}\widetilde{\boldsymbol{\theta}}\cdot\mathbf{u}_{12}^\top\mathbf{H}\mathbf{u}_{12}\cdot\widetilde{\boldsymbol{\theta}}^\top\boldsymbol{\Psi}^{\frac{1}{2}}\mathbf{X}^\top\boldsymbol{\varepsilon}\bigg\}$$

$$= \frac{\sigma^2}{M}\bigg\{\boldsymbol{b}^\top\mathbf{H}\boldsymbol{b}\cdot\mathbf{u}_{12}^\top\mathbf{H}\mathbf{u}_{12} + u_{-1}^2\mathrm{tr}\left(\mathbf{H}\boldsymbol{A}\mathbf{H}\boldsymbol{A}^\top\right) + 2u_{-1}\mathbf{u}_{12}^\top\mathbf{H}\boldsymbol{A}^\top\mathbf{H}\boldsymbol{b} + 4u_{-1}^2\mathrm{tr}\left(\mathbf{H}\boldsymbol{\Psi}\right)\mathbf{u}_{12}^\top\mathbf{H}\mathbf{u}_{12}\bigg\}$$
$$(\text{B.12})$$

For the other two terms, we have

$$S_3 = \frac{1}{M^2}u_{-1}^2\mathbb{E}\left\{\boldsymbol{\varepsilon}^\top\boldsymbol{\varepsilon}\mathbf{u}_{12}^\top\mathbf{H}\mathbf{u}_{12}\boldsymbol{\varepsilon}^\top\boldsymbol{\varepsilon}\right\} = \frac{\sigma^4(M+2)}{M}u_{-1}^2\mathbf{u}_{12}^\top\mathbf{H}\mathbf{u}_{12} \qquad (\text{B.13})$$

and

$$S_4 = 2\left\langle\mathbf{H},\mathbb{E}\left(\boldsymbol{z}_1-\widetilde{\boldsymbol{\beta}}\right)\boldsymbol{z}_3^\top\right\rangle$$

$$= 2\mathbb{E}\bigg\{\left[\left(\mathbf{u}_{21}+u_{-1}\widetilde{\boldsymbol{\beta}}\right)^\top\cdot\frac{1}{M}\mathbf{X}^\top\mathbf{X}\cdot\left(\mathbf{U}_{11}+\widetilde{\boldsymbol{\beta}}\mathbf{u}_{12}^\top\right)-\widetilde{\boldsymbol{\beta}}^\top\right]\mathbf{H}\cdot\frac{1}{M}\boldsymbol{\varepsilon}^\top\boldsymbol{\varepsilon}\cdot u_{-1}\mathbf{u}_{12}^\top\bigg\}$$

$$= 2\sigma^2 u_{-1}\mathbb{E}\bigg\{\left[\left(\boldsymbol{b}+u_{-1}\boldsymbol{\Psi}^{\frac{1}{2}}\widetilde{\boldsymbol{\theta}}\right)^\top\cdot\mathbf{H}\cdot\left(\boldsymbol{A}+\boldsymbol{\Psi}^{\frac{1}{2}}\widetilde{\boldsymbol{\theta}}\mathbf{u}_{12}^\top\right)-\widetilde{\boldsymbol{\beta}}^{*\top}-\widetilde{\boldsymbol{\theta}}^\top\boldsymbol{\Psi}^{\frac{1}{2}}\right]\mathbf{H}\mathbf{u}_{12}\bigg\}$$

$$= 2\sigma^2 u_{-1}\left[\boldsymbol{b}^\top\mathbf{H}\boldsymbol{A}-\widetilde{\boldsymbol{\beta}}^{*\top}\right]\mathbf{H}\mathbf{u}_{12} + 2\sigma^2 u_{-1}^2\mathrm{tr}\left(\mathbf{H}\boldsymbol{\Psi}\right)\cdot\mathbf{u}_{12}^\top\mathbf{H}\mathbf{u}_{12}. \qquad (\text{B.14})$$

**Step 4: combine all parts.** Combining the four parts (B.10), (B.12), (B.13) and (B.14), we have

$$\mathcal{R}\left(f\right)-\sigma^2 = S_1 + S_2 + S_3 + S_4$$

We remark that in the (B.10), (B.12), (B.13) and (B.14), the parameters are $\boldsymbol{A},\boldsymbol{b},u_{-1}$ and $\mathbf{u}_{12}$, instead of the original parameter $\mathbf{U}_{11},\mathbf{u}_{12},\mathbf{u}_{21},u_{-1}$. From the definitions of $\boldsymbol{A}$ and $\boldsymbol{b}$, we know there is a bijective map between $(\mathbf{U}_{11},\mathbf{u}_{12},\mathbf{u}_{21},u_{-1})$ and $(\boldsymbol{A},\boldsymbol{b},\mathbf{u}_{12},u_{-1})$, so these two parameterizations are equivalent when computing the minimal risk achieved by a model in a hypothesis class. From the expression of $S_1, S_2, s_3, S_4$, we can rewrite the risk as

$$\mathcal{R}\left(f\right)-\sigma^2$$

$$= \boldsymbol{b}^\top \left( \frac{M+1}{M} \mathbf{H}\boldsymbol{\Psi}\mathbf{H} + \frac{1}{M}\mathrm{tr}(\mathbf{H}\boldsymbol{\Psi})\mathbf{H} \right) \boldsymbol{b} \cdot \mathbf{u}_{12}^\top \mathbf{H}\mathbf{u}_{12} + \frac{\sigma^2}{M} \cdot \boldsymbol{b}^\top \mathbf{H}\boldsymbol{b} \cdot \mathbf{u}_{12}^\top \mathbf{H}\mathbf{u}_{12}$$

$$+ u_{-1}^2 \mathrm{tr}\left( \boldsymbol{A}^\top \left( \frac{M+1}{M}\mathbf{H}\boldsymbol{\Psi}\mathbf{H} + \frac{1}{M}\mathrm{tr}(\mathbf{H}\boldsymbol{\Psi})\mathbf{H} \right) \boldsymbol{A}\mathbf{H} \right) + \frac{\sigma^2}{M} \cdot \mathrm{tr}\left( \mathbf{H}\boldsymbol{A}\mathbf{H}\boldsymbol{A}^\top \right)$$

$$\underbrace{+ 2u_{-1}\boldsymbol{b}^\top \left[ \frac{1}{M}\mathrm{tr}(\mathbf{H}\boldsymbol{\Psi})\mathbf{H} + \frac{M+1}{M}\mathbf{H}\boldsymbol{\Psi}\mathbf{H} + \frac{\sigma^2}{M}\cdot\mathbf{H} \right] \boldsymbol{A}\mathbf{H}\mathbf{u}_{12} - 2u_{-1}\mathrm{tr}\left( \mathbf{H}\boldsymbol{A}\mathbf{H}\boldsymbol{\Psi} \right) - 2\mathbf{u}_{12}^\top\mathbf{H}\boldsymbol{\Psi}\mathbf{H}\boldsymbol{b}}_{\text{I}}$$

$$+ \underbrace{\frac{1}{M} \left[ \boldsymbol{b}^\top \mathbf{H}\boldsymbol{b} \cdot \mathrm{tr}\left( \boldsymbol{A}^\top \mathbf{H}\boldsymbol{A}\mathbf{H} \right) + 4u_{-1} \cdot \boldsymbol{b}^\top \mathbf{H}\boldsymbol{\Psi}\mathbf{H}\boldsymbol{A}\mathbf{H}\mathbf{u}_{12} + 4u_{-1}^2 \cdot \mathrm{tr}\left( \mathbf{H}\boldsymbol{\Psi}\mathbf{H}\boldsymbol{\Psi} \right) \cdot \mathbf{u}_{12}^\top \mathbf{H}\mathbf{u}_{12} \right]}_{\text{II}}$$

$$+ \underbrace{\boldsymbol{b}^\top \left( \frac{M+1}{M}\mathbf{H}\boldsymbol{A}\mathbf{H}\boldsymbol{A}^\top\mathbf{H} \right) \boldsymbol{b} - 2\boldsymbol{b}^\top\mathbf{H}\boldsymbol{A}\mathbf{H}\boldsymbol{\beta}^* + 2u_{-1}\mathrm{tr}(\mathbf{H}\boldsymbol{\Psi}) \cdot \boldsymbol{b}^\top \mathbf{H}\boldsymbol{A}\mathbf{H}\mathbf{u}_{12} + 2\sigma^2\boldsymbol{b}^\top\mathbf{H}\boldsymbol{A}\mathbf{H}\left( u_{-1}\mathbf{u}_{12} \right)}_{\text{III}}$$

$$+ \boldsymbol{\beta}^{*\top}\mathbf{H}\boldsymbol{\beta}^* - 2\left( \mathrm{tr}(\mathbf{H}\boldsymbol{\Psi}) + \sigma^2 \right) \cdot \boldsymbol{\beta}^{*\top}\mathbf{H}\left( u_{-1}\mathbf{u}_{12} \right) + \mathrm{tr}(\mathbf{H}\boldsymbol{\Psi})$$

$$\underbrace{+ u_{-1}^2 \cdot \left[ \left( 2\mathrm{tr}(\mathbf{H}\boldsymbol{\Psi}\mathbf{H}\boldsymbol{\Psi}) + \frac{M+2}{M}\mathrm{tr}(\mathbf{H}\boldsymbol{\Psi})^2 \right) \cdot \mathbf{u}_{12}^\top\mathbf{H}\mathbf{u}_{12} + \left( 2 + \frac{4}{M} \right)\sigma^2\mathrm{tr}(\mathbf{H}\boldsymbol{\Psi}) + \frac{M+2}{M}\sigma^4 \right] \cdot \mathbf{u}_{12}^\top\mathbf{H}\mathbf{u}_{12}}_{\text{IV}}$$

$$= \mathrm{I} + \mathrm{II} + \mathrm{III} + \mathrm{IV}, \tag{B.15}$$

where I, II, III, IV are defined as above.

**Step 5: lower bound the risk function.** Let's first define a new matrix, which by definition is invertible:

$$\boldsymbol{\Omega} := \frac{M+1}{M}\mathbf{H}^{\frac{1}{2}}\boldsymbol{\Psi}\mathbf{H}^{\frac{1}{2}} + \frac{\mathrm{tr}(\mathbf{H}\boldsymbol{\Psi}) + \sigma^2}{M}\mathbf{I}_d. \tag{B.16}$$

So we can write I as

$$\mathrm{I} = \boldsymbol{b}^\top \mathbf{H}^{\frac{1}{2}}\boldsymbol{\Omega}\mathbf{H}^{\frac{1}{2}}\boldsymbol{b} \cdot \mathbf{u}_{12}^\top\mathbf{H}\mathbf{u}_{12} + u_{-1}^2\mathrm{tr}\left( \boldsymbol{A}^\top\mathbf{H}^{\frac{1}{2}}\boldsymbol{\Omega}\mathbf{H}^{\frac{1}{2}}\boldsymbol{A}\mathbf{H} \right) + 2u_{-1}\boldsymbol{b}^\top\mathbf{H}^{\frac{1}{2}}\boldsymbol{\Omega}\mathbf{H}^{\frac{1}{2}}\boldsymbol{A}\mathbf{H}\mathbf{u}_{12}$$

$$- 2u_{-1}\mathrm{tr}(\mathbf{H}\boldsymbol{A}\mathbf{H}\boldsymbol{\Psi}) - 2\mathbf{u}_{12}^\top\mathbf{H}\boldsymbol{\Psi}\mathbf{H}\boldsymbol{b}$$

$$= \mathrm{tr}\left[ \left( \mathbf{H}^{\frac{1}{2}}\mathbf{u}_{12}\boldsymbol{b}^\top\mathbf{H}^{\frac{1}{2}} + u_{-1}\mathbf{H}^{\frac{1}{2}}\boldsymbol{A}^\top\mathbf{H}^{\frac{1}{2}} - \mathbf{H}^{\frac{1}{2}}\boldsymbol{\Psi}\mathbf{H}^{\frac{1}{2}}\boldsymbol{\Omega}^{-1} \right)\boldsymbol{\Omega} \right.$$

$$\left. \cdot \left( \mathbf{H}^{\frac{1}{2}}\mathbf{u}_{12}\boldsymbol{b}^\top\mathbf{H}^{\frac{1}{2}} + u_{-1}\mathbf{H}^{\frac{1}{2}}\boldsymbol{A}^\top\mathbf{H}^{\frac{1}{2}} - \mathbf{H}^{\frac{1}{2}}\boldsymbol{\Psi}\mathbf{H}^{\frac{1}{2}}\boldsymbol{\Omega}^{-1} \right)^\top \right] - \mathrm{tr}\left( \mathbf{H}^{\frac{1}{2}}\boldsymbol{\Psi}\mathbf{H}^{\frac{1}{2}}\boldsymbol{\Omega}^{-1}\mathbf{H}^{\frac{1}{2}}\boldsymbol{\Psi}\mathbf{H}^{\frac{1}{2}} \right)$$

$$\geq -\mathrm{tr}\left( \mathbf{H}^{\frac{1}{2}}\boldsymbol{\Psi}\mathbf{H}^{\frac{1}{2}}\boldsymbol{\Omega}^{-1}\mathbf{H}^{\frac{1}{2}}\boldsymbol{\Psi}\mathbf{H}^{\frac{1}{2}} \right) \tag{B.17}$$

$$= -\mathrm{tr}\left( \left( \mathbf{H}^{\frac{1}{2}}\boldsymbol{\Psi}\mathbf{H}^{\frac{1}{2}} \right)^2 \boldsymbol{\Omega}^{-1} \right),$$

where the last line comes from the fact that $\boldsymbol{\Omega}$ and $\mathbf{H}^{\frac{1}{2}}\boldsymbol{\Psi}\mathbf{H}^{\frac{1}{2}}$ commute.

Next, we claim II $\geq 0$. To see this, it suffices to notice that

$$4u_{-1} \cdot \boldsymbol{b}^\top\mathbf{H}\boldsymbol{\Psi}\mathbf{H}\boldsymbol{A}\mathbf{H}\mathbf{u}_{12} \geq -4 \left\| \boldsymbol{b}^\top\mathbf{H}^{\frac{1}{2}} \right\|_F \cdot \left\| u_{-1}\mathbf{H}^{\frac{1}{2}}\boldsymbol{\Psi}\mathbf{H}^{\frac{1}{2}} \right\|_F \cdot \left\| \mathbf{H}^{\frac{1}{2}}\boldsymbol{A}\mathbf{H}^{\frac{1}{2}} \right\|_F \cdot \left\| \mathbf{H}^{\frac{1}{2}}\mathbf{u}_{12} \right\|_F$$

$$\geq - \left\| \boldsymbol{b}^\top\mathbf{H}^{\frac{1}{2}} \right\|_F^2 \left\| \mathbf{H}^{\frac{1}{2}}\boldsymbol{A}\mathbf{H}^{\frac{1}{2}} \right\|_F^2 - 4 \left\| u_{-1}\mathbf{H}^{\frac{1}{2}}\boldsymbol{\Psi}\mathbf{H}^{\frac{1}{2}} \right\|_F^2 \left\| \mathbf{H}^{\frac{1}{2}}\mathbf{u}_{12} \right\|_F^2$$

$$\geq -\boldsymbol{b}^\top\mathbf{H}\boldsymbol{b} \cdot \mathrm{tr}\left( \boldsymbol{A}^\top\mathbf{H}\boldsymbol{A}\mathbf{H} \right) - 4u_{-1}^2 \cdot \mathrm{tr}(\mathbf{H}\boldsymbol{\Psi}\mathbf{H}\boldsymbol{\Psi}) \cdot \mathbf{u}_{12}^\top\mathbf{H}\mathbf{u}_{12}, \tag{B.18}$$

where the last line comes from the fact that $\|\boldsymbol{A}\|_F^2 = \mathrm{tr}\left( \boldsymbol{A}\boldsymbol{A}^\top \right)$ for any matrix $\boldsymbol{A}$.

Then, let's consider III. Notice that actually, III can be viewed as a quadratic function of $\boldsymbol{A}^\top \mathbf{H} \boldsymbol{b}$, which can be easily minimized. More concretely, we have

$$\text{III} + \frac{M}{M+1}\left(\boldsymbol{\beta}^* - \left(\text{tr}\left(\mathbf{H}\boldsymbol{\Psi}\right) + \sigma^2\right)u_{-1}\mathbf{u}_{12}\right)^\top \mathbf{H}\left(\boldsymbol{\beta}^* - \left(\text{tr}\left(\mathbf{H}\boldsymbol{\Psi}\right) + \sigma^2\right)u_{-1}\mathbf{u}_{12}\right)$$

$$= \left[\boldsymbol{A}^\top \mathbf{H}\boldsymbol{b} - \frac{M}{M+1}\left(\boldsymbol{\beta}^* - \left(\text{tr}\left(\mathbf{H}\boldsymbol{\Psi}\right) + \sigma^2\right)u_{-1}\mathbf{u}_{12}\right)\right]^\top \left(\frac{M+1}{M}\mathbf{H}\right)$$

$$\cdot \left[\boldsymbol{A}^\top \mathbf{H}\boldsymbol{b} - \frac{M}{M+1}\left(\boldsymbol{\beta}^* - \left(\text{tr}\left(\mathbf{H}\boldsymbol{\Psi}\right) + \sigma^2\right)u_{-1}\mathbf{u}_{12}\right)\right]$$

$$\geq 0. \tag{B.19}$$

Therefore, one has

$$\text{III} \geq -\frac{M}{M+1}\left(\boldsymbol{\beta}^* - \left(\text{tr}\left(\mathbf{H}\boldsymbol{\Psi}\right) + \sigma^2\right)u_{-1}\mathbf{u}_{12}\right)^\top \mathbf{H}\left(\boldsymbol{\beta}^* - \left(\text{tr}\left(\mathbf{H}\boldsymbol{\Psi}\right) + \sigma^2\right)u_{-1}\mathbf{u}_{12}\right).$$

Combining the three parts above, one has

$$\mathcal{R}\left(f\right) - \sigma^2$$

$$\geq \text{IV} - \text{tr}\left(\left(\mathbf{H}^{\frac{1}{2}}\boldsymbol{\Psi}\mathbf{H}^{\frac{1}{2}}\right)^2 \boldsymbol{\Omega}^{-1}\right)$$

$$- \frac{M}{M+1}\left(\boldsymbol{\beta}^* - \left(\text{tr}\left(\mathbf{H}\boldsymbol{\Psi}\right) + \sigma^2\right)u_{-1}\mathbf{u}_{12}\right)^\top \mathbf{H}\left(\boldsymbol{\beta}^* - \left(\text{tr}\left(\mathbf{H}\boldsymbol{\Psi}\right) + \sigma^2\right)u_{-1}\mathbf{u}_{12}\right)$$

$$= \underbrace{\text{tr}\left(\mathbf{H}\boldsymbol{\Psi}\right) - \text{tr}\left(\left(\mathbf{H}^{\frac{1}{2}}\boldsymbol{\Psi}\mathbf{H}^{\frac{1}{2}}\right)^2 \boldsymbol{\Omega}^{-1}\right)}_{\inf_{f \in \mathcal{F}_{\text{LTB}}} \mathcal{R}(f) - \sigma^2} + \frac{1}{M+1}\boldsymbol{\beta}^{*\top}\mathbf{H}\boldsymbol{\beta}^* - \frac{2\left(\text{tr}\left(\mathbf{H}\boldsymbol{\Psi}\right) + \sigma^2\right)}{M+1}\boldsymbol{\beta}^{*\top}\mathbf{H}\left(u_{-1}\mathbf{u}_{12}\right)$$

$$+ \left[2\text{tr}\left(\mathbf{H}\boldsymbol{\Psi}\mathbf{H}\boldsymbol{\Psi}\right) + \frac{3M+2}{M(M+1)}\left(\text{tr}\left(\mathbf{H}\boldsymbol{\Psi}\right) + \sigma^2\right)^2\right]\left(u_{-1}\mathbf{u}_{12}\right)^\top \mathbf{H}\left(u_{-1}\mathbf{u}_{12}\right).$$

Therefore, one has

$$\mathcal{R}\left(f\right) - \inf_{f \in \mathcal{F}_{\text{LTB}}} \mathcal{R}(f) \geq \frac{1}{M+1}\boldsymbol{\beta}^{*\top}\mathbf{H}\boldsymbol{\beta}^* - \frac{2\left(\text{tr}\left(\mathbf{H}\boldsymbol{\Psi}\right) + \sigma^2\right)}{M+1}\boldsymbol{\beta}^{*\top}\mathbf{H}\left(u_{-1}\mathbf{u}_{12}\right)$$

$$+ \left[2\text{tr}\left(\mathbf{H}\boldsymbol{\Psi}\mathbf{H}\boldsymbol{\Psi}\right) + \frac{3M+2}{M(M+1)}\left(\text{tr}\left(\mathbf{H}\boldsymbol{\Psi}\right) + \sigma^2\right)^2\right]\left(u_{-1}\mathbf{u}_{12}\right)^\top \mathbf{H}\left(u_{-1}\mathbf{u}_{12}\right).$$

The right hand side in the above inequality is a quadratic function of $u_{-1}\mathbf{u}_{12}$, so we can take its global minimizer:

$$u_{-1}\mathbf{u}_{12} = \frac{\frac{\left(\text{tr}(\mathbf{H}\boldsymbol{\Psi}) + \sigma^2\right)}{M+1}}{2\text{tr}\left(\mathbf{H}\boldsymbol{\Psi}\mathbf{H}\boldsymbol{\Psi}\right) + \frac{3M+2}{M(M+1)}\left(\text{tr}\left(\mathbf{H}\boldsymbol{\Psi}\right) + \sigma^2\right)^2}\boldsymbol{\beta}^*$$

to lower bound the risk gap as

$$\mathcal{R}\left(f\right) - \inf_{f \in \mathcal{F}_{\text{LTB}}} \mathcal{R}(f) \geq \left[\frac{1}{M+1} - \frac{\frac{\left(\text{tr}(\mathbf{H}\boldsymbol{\Psi}) + \sigma^2\right)^2}{(M+1)^2}}{2\text{tr}\left(\mathbf{H}\boldsymbol{\Psi}\mathbf{H}\boldsymbol{\Psi}\right) + \frac{3M+2}{M(M+1)}\left(\text{tr}\left(\mathbf{H}\boldsymbol{\Psi}\right) + \sigma^2\right)^2}\right]\|\boldsymbol{\beta}^*\|_{\mathbf{H}}^2.$$

Finally, to simplify the results, we notice that on the one hand,

$$\frac{\frac{\left(\text{tr}(\mathbf{H}\boldsymbol{\Psi}) + \sigma^2\right)^2}{(M+1)^2}}{2\text{tr}\left(\mathbf{H}\boldsymbol{\Psi}\mathbf{H}\boldsymbol{\Psi}\right) + \frac{3M+2}{M(M+1)}\left(\text{tr}\left(\mathbf{H}\boldsymbol{\Psi}\right) + \sigma^2\right)^2} \leq \frac{\left(\text{tr}\left(\mathbf{H}\boldsymbol{\Psi}\right) + \sigma^2\right)^2}{2(M+1)^2\text{tr}\left(\mathbf{H}\boldsymbol{\Psi}\mathbf{H}\boldsymbol{\Psi}\right)}.$$

On the other hand, one has

$$\frac{\frac{\left(\operatorname{tr}(\mathbf{H}\boldsymbol{\Psi})+\sigma^2\right)^2}{(M+1)^2}}{2\operatorname{tr}\left(\mathbf{H}\boldsymbol{\Psi}\mathbf{H}\boldsymbol{\Psi}\right)+\frac{3M+2}{M(M+1)}\left(\operatorname{tr}\left(\mathbf{H}\boldsymbol{\Psi}\right)+\sigma^2\right)^2} \leq \frac{M}{(M+1)(3M+2)} \leq \frac{1}{3(M+1)}.$$

Therefore, we have

$$\mathcal{R}\left(f\right) - \inf_{f \in \mathcal{F}_{\mathsf{LTB}}} \mathcal{R}(f) \geq \max\left\{\frac{2}{3(M+1)}, \frac{1}{M+1} - \frac{\left(\operatorname{tr}\left(\mathbf{H}\boldsymbol{\Psi}\right)+\sigma^2\right)^2}{2(M+1)^2\operatorname{tr}\left(\mathbf{H}\boldsymbol{\Psi}\mathbf{H}\boldsymbol{\Psi}\right)}\right\} \cdot \|\boldsymbol{\beta}^*\|_{\mathbf{H}}^2.$$

When $\frac{1}{M+1} - \frac{\left(\operatorname{tr}(\mathbf{H}\boldsymbol{\Psi})+\sigma^2\right)^2}{2(M+1)^2\operatorname{tr}(\mathbf{H}\boldsymbol{\Psi}\mathbf{H}\boldsymbol{\Psi})} \geq \frac{2}{3(M+1)}$, we have $\frac{1}{M+1} \geq \frac{3\left(\operatorname{tr}(\mathbf{H}\boldsymbol{\Psi})+\sigma^2\right)^2}{2(M+1)^2\operatorname{tr}(\mathbf{H}\boldsymbol{\Psi}\mathbf{H}\boldsymbol{\Psi})}$, which implies

$$\mathcal{R}\left(f\right) - \inf_{f \in \mathcal{F}_{\mathsf{LTB}}} \mathcal{R}(f) \geq \left[\frac{1}{M+1} - \frac{\left(\operatorname{tr}\left(\mathbf{H}\boldsymbol{\Psi}\right)+\sigma^2\right)^2}{2(M+1)^2\operatorname{tr}\left(\mathbf{H}\boldsymbol{\Psi}\mathbf{H}\boldsymbol{\Psi}\right)}\right] \cdot \|\boldsymbol{\beta}^*\|_{\mathbf{H}}^2 \geq \frac{\left(\operatorname{tr}\left(\mathbf{H}\boldsymbol{\Psi}\right)+\sigma^2\right)^2}{(M+1)^2\operatorname{tr}\left(\mathbf{H}\boldsymbol{\Psi}\mathbf{H}\boldsymbol{\Psi}\right)} \cdot \|\boldsymbol{\beta}^*\|_{\mathbf{H}}^2.$$

Therefore, we finally have

$$\mathcal{R}\left(f\right) - \inf_{f \in \mathcal{F}_{\mathsf{LTB}}} \mathcal{R}(f) \geq \max\left\{\frac{2}{3(M+1)}, \frac{\left(\operatorname{tr}\left(\mathbf{H}\boldsymbol{\Psi}\right)+\sigma^2\right)^2}{(M+1)^2\operatorname{tr}\left(\mathbf{H}\boldsymbol{\Psi}\mathbf{H}\boldsymbol{\Psi}\right)}\right\} \cdot \|\boldsymbol{\beta}^*\|_{\mathbf{H}}^2.$$

Note that this holds for an arbitrary $f \in \mathcal{F}_{\mathsf{LSA}}$, so the proof finishes by taking infimum on the left hand side. □

## C   Proof of Theorem 5.2

*Proof.* Recall that from Assumption 3.1,

$$\mathbf{x} \sim \mathcal{N}(0, \mathbf{H}), \quad y = \mathbf{x}^\top \widetilde{\boldsymbol{\beta}} + \varepsilon, \quad \mathbf{X}[i] \sim \mathcal{N}(0, \mathbf{H}), \quad \mathbf{y} = \mathbf{X}\widetilde{\boldsymbol{\beta}} + \boldsymbol{\varepsilon},$$

where

$$\widetilde{\boldsymbol{\beta}} \sim \mathcal{N}\left(\boldsymbol{\beta}^*, \boldsymbol{\Psi}\right), \quad \varepsilon \sim \mathcal{N}\left(0, \sigma^2\right), \quad \boldsymbol{\varepsilon} \sim \mathcal{N}\left(\mathbf{0}, \sigma^2 \cdot \mathbf{I}_M\right).$$

**Step 1: compute the risk function.**   From the independence of $\varepsilon, \boldsymbol{\varepsilon}$ with other random variables, we have

$$
\begin{aligned}
\mathcal{R}_M\left(\boldsymbol{\beta}, \boldsymbol{\Gamma}\right) &= \mathbb{E}\left(\mathbf{x}^\top\left(\widetilde{\boldsymbol{\beta}} - \boldsymbol{\beta}\right) + \varepsilon + \mathbf{x}^\top \boldsymbol{\Gamma} \cdot \frac{\mathbf{X}^\top \mathbf{X}}{M}\left(\boldsymbol{\beta} - \widetilde{\boldsymbol{\beta}}\right) - \mathbf{x}^\top \boldsymbol{\Gamma} \cdot \frac{\mathbf{X}^\top \boldsymbol{\varepsilon}}{M}\right)^2 \\
&= \mathbb{E}\left(\mathbf{x}^\top\left(\mathbf{I}_d - \boldsymbol{\Gamma} \cdot \frac{\mathbf{X}^\top \mathbf{X}}{M}\right) \cdot \left(\widetilde{\boldsymbol{\beta}} - \boldsymbol{\beta}\right)\right)^2 + \frac{1}{M^2}\mathbb{E}\left(\mathbf{x}^\top \boldsymbol{\Gamma} \mathbf{X}^\top \boldsymbol{\varepsilon}\right)^2 + \sigma^2 \\
&= \left\langle \mathbf{H}, \mathbb{E}\left(\mathbf{I}_d - \boldsymbol{\Gamma} \cdot \frac{\mathbf{X}^\top \mathbf{X}}{M}\right)^{\otimes 2} \circ \mathbb{E}\left(\widetilde{\boldsymbol{\beta}} - \boldsymbol{\beta}\right)^{\otimes 2}\right\rangle + \frac{1}{M^2}\mathbb{E}\left(\mathbf{x}^\top \boldsymbol{\Gamma} \mathbf{X}^\top \boldsymbol{\varepsilon}\right)^2 + \sigma^2,
\end{aligned}
$$

$$\text{(C.1)}$$

where the last line comes from the independence between $\widetilde{\boldsymbol{\beta}}$ and $\mathbf{X}, \mathbf{x}$, as well as the fact that $\mathbf{x} \sim \mathcal{N}(\mathbf{0}, \mathbf{H})$ and the property of tensor product $\circ$. Note that, $\boldsymbol{\beta}$ ad $\boldsymbol{\Gamma}$ are learnable parameters here, and we should set them apart from the task vector $\widetilde{\boldsymbol{\beta}}$ or the prior mean vector $\widetilde{\boldsymbol{\beta}}^*$. Recall that $\widetilde{\boldsymbol{\beta}} \sim \mathcal{N}\left(\boldsymbol{\beta}^*, \boldsymbol{\Psi}\right)$, we have $\mathbb{E}\left(\widetilde{\boldsymbol{\beta}} - \boldsymbol{\beta}\right)^{\otimes 2} = (\boldsymbol{\beta} - \boldsymbol{\beta}^*)^{\otimes 2} + \boldsymbol{\Psi}$. Moreover, using the following

$$\mathbf{x} \sim \mathcal{N}\left(\mathbf{0}, \mathbf{H}\right), \quad \mathbf{X}[i] \sim \mathcal{N}\left(\mathbf{0}, \mathbf{H}\right), \quad \boldsymbol{\varepsilon} \sim \mathcal{N}\left(\mathbf{0}, \sigma^2 \cdot \mathbf{I}_M\right),$$

we have that

$$\mathbb{E}\left(\mathbf{x}^\top \boldsymbol{\Gamma} \mathbf{X}^\top \boldsymbol{\varepsilon}\right)^2 = \sigma^2 \mathbb{E}\mathrm{tr}\left(\mathbf{X}\boldsymbol{\Gamma}^\top \mathbf{x}\mathbf{x}^\top \boldsymbol{\Gamma} \mathbf{X}^\top\right) = M\sigma^2 \cdot \left\langle \mathbf{H}\boldsymbol{\Gamma}\mathbf{H}, \boldsymbol{\Gamma}^\top \right\rangle.$$

Bridging the two terms above into (C.1), we get

$$
\begin{aligned}
\mathcal{R}_M\left(\boldsymbol{\beta}, \boldsymbol{\Gamma}\right) &= \left\langle \mathbf{H}, \mathbb{E}\left(\mathbf{I}_d - \boldsymbol{\Gamma} \cdot \frac{\mathbf{X}^\top \mathbf{X}}{M}\right)^{\otimes 2} \circ (\boldsymbol{\beta} - \boldsymbol{\beta}^*)^{\otimes 2}\right\rangle \\
&\quad + \left\langle \mathbf{H}, \mathbb{E}\left(\mathbf{I}_d - \boldsymbol{\Gamma} \cdot \frac{\mathbf{X}^\top \mathbf{X}}{M}\right)^{\otimes 2} \circ \boldsymbol{\Psi} + \frac{\sigma^2}{M}\boldsymbol{\Gamma}\mathbf{H}\boldsymbol{\Gamma}^\top\right\rangle + \sigma^2 \\
&= \underbrace{\left\langle \mathbb{E}\left(\mathbf{I}_d - \boldsymbol{\Gamma} \cdot \frac{\mathbf{X}^\top \mathbf{X}}{M}\right)^{\otimes 2} \circ \mathbf{H}, (\boldsymbol{\beta} - \boldsymbol{\beta}^*)^{\otimes 2}\right\rangle}_{V_1} \\
&\quad + \underbrace{\left\langle \mathbf{H}, \mathbb{E}\left(\left(\mathbf{I}_d - \boldsymbol{\Gamma} \cdot \frac{\mathbf{X}^\top \mathbf{X}}{M}\right)^\top\right)^{\otimes 2} \circ \boldsymbol{\Psi} + \frac{\sigma^2}{M}\boldsymbol{\Gamma}\mathbf{H}\boldsymbol{\Gamma}^\top\right\rangle}_{V_2} + \sigma^2 \quad\quad \text{(C.2)}
\end{aligned}
$$

First, let's compute $V_1$. From the definition above, we have

$$V_1 = (\boldsymbol{\beta} - \boldsymbol{\beta}^*)^\top \mathbf{H}_{\boldsymbol{\Gamma}}\left(\boldsymbol{\beta} - \boldsymbol{\beta}^*\right),$$

where

$$\mathbf{H}_{\boldsymbol{\Gamma}} := \mathbb{E}\left(\left(\mathbf{I}_d - \boldsymbol{\Gamma} \cdot \frac{\mathbf{X}^\top \mathbf{X}}{M}\right)^\top\right)^{\otimes 2} \circ \mathbf{H} \quad\quad\quad\quad \text{(C.3)}$$

$$= \mathbb{E}\left(\mathbf{I}_d - \mathbf{\Gamma} \cdot \frac{\mathbf{X}^\top \mathbf{X}}{M}\right)^\top \mathbf{H}\left(\mathbf{I}_d - \mathbf{\Gamma} \cdot \frac{\mathbf{X}^\top \mathbf{X}}{M}\right) \qquad \text{(the definition of the tensor product)}$$

$$= \mathbf{H} - \mathbf{H}\left(\mathbf{\Gamma} + \mathbf{\Gamma}^\top\right)\mathbf{H} + \frac{\text{tr}\left(\mathbf{H}\mathbf{\Gamma}^\top \mathbf{H}\mathbf{\Gamma}\right)}{M}\mathbf{H} + \frac{M+1}{M}\mathbf{H}\mathbf{\Gamma}^\top \mathbf{H}\mathbf{\Gamma}\mathbf{H}$$
$$\qquad\qquad (\mathbf{X}[i] \sim \mathcal{N}(\mathbf{0}, \mathbf{H}) \text{ and Lemma H.4})$$

$$= (\mathbf{I}_d - \mathbf{\Gamma}\mathbf{H})^\top \mathbf{H}(\mathbf{I}_d - \mathbf{\Gamma}\mathbf{H}) + \frac{\text{tr}\left(\mathbf{H}\mathbf{\Gamma}^\top \mathbf{H}\mathbf{\Gamma}\right)}{M}\mathbf{H} + \frac{1}{M}\mathbf{H}\mathbf{\Gamma}^\top \mathbf{H}\mathbf{\Gamma}\mathbf{H} \succeq \mathbf{0}. \qquad (C.4)$$

Then, let's compute $V_2$. We have

$$\mathbb{E}\left(\mathbf{I}_d - \mathbf{\Gamma} \cdot \frac{\mathbf{X}^\top \mathbf{X}}{M}\right)^{\otimes 2} \circ \mathbf{\Psi} + \frac{\sigma^2}{M}\mathbf{\Gamma}\mathbf{H}\mathbf{\Gamma}^\top = \mathbb{E}\left(\mathbf{I}_d - \mathbf{\Gamma} \cdot \frac{\mathbf{X}^\top \mathbf{X}}{M}\right)\mathbf{\Psi}\left(\mathbf{I}_d - \mathbf{\Gamma} \cdot \frac{\mathbf{X}^\top \mathbf{X}}{M}\right)^\top + \frac{\sigma^2}{M}\mathbf{\Gamma}\mathbf{H}\mathbf{\Gamma}^\top$$

$$= \mathbf{\Psi} - \mathbf{\Gamma}\mathbf{H}\mathbf{\Psi} - \mathbf{\Psi}\mathbf{H}\mathbf{\Gamma}^\top + \mathbf{\Gamma}\left(\frac{\text{tr}(\mathbf{H}\mathbf{\Psi}) + \sigma^2}{M} \cdot \mathbf{H} + \frac{M+1}{M}\mathbf{H}\mathbf{\Psi}\mathbf{H}\right)\mathbf{\Gamma}^\top.$$
$$\qquad\qquad (\mathbf{X}[i] \overset{\text{i.i.d.}}{\sim} \mathcal{N}(\mathbf{0}, \mathbf{H}) \text{ and the Lemma H.4})$$

From the definition of $\mathbf{\Omega}$ in (A.3), we know that it is invertible. Moreover, it holds that

$$\mathbb{E}\left(\mathbf{I}_d - \mathbf{\Gamma} \cdot \frac{\mathbf{X}^\top \mathbf{X}}{M}\right)^{\otimes 2} \circ \mathbf{\Psi} + \frac{\sigma^2}{M}\mathbf{\Gamma}\mathbf{H}\mathbf{\Gamma}^\top = \mathbf{\Psi} - \mathbf{\Gamma}\mathbf{H}\mathbf{\Psi} - \mathbf{\Psi}\mathbf{H}\mathbf{\Gamma}^\top + \mathbf{\Gamma}\mathbf{H}^{\frac{1}{2}}\mathbf{\Omega}\mathbf{H}^{\frac{1}{2}}\mathbf{\Gamma}^\top$$

$$= \left(\mathbf{\Gamma}\mathbf{H}^{\frac{1}{2}} - \mathbf{\Psi}\mathbf{H}^{\frac{1}{2}}\mathbf{\Omega}^{-1}\right)\mathbf{\Omega}\left(\mathbf{\Gamma}\mathbf{H}^{\frac{1}{2}} - \mathbf{\Psi}\mathbf{H}^{\frac{1}{2}}\mathbf{\Omega}^{-1}\right)^\top + \mathbf{\Psi} - \mathbf{\Psi}\mathbf{H}^{\frac{1}{2}}\mathbf{\Omega}^{-1}\mathbf{H}^{\frac{1}{2}}\mathbf{\Psi}.$$

Therefore, we have

$$V_2 = \text{tr}\left[\left(\mathbf{H}^{\frac{1}{2}}\mathbf{\Gamma}\mathbf{H}^{\frac{1}{2}} - \mathbf{H}^{\frac{1}{2}}\mathbf{\Psi}\mathbf{H}^{\frac{1}{2}}\mathbf{\Omega}^{-1}\right)\mathbf{\Omega}\left(\mathbf{H}^{\frac{1}{2}}\mathbf{\Gamma}\mathbf{H}^{\frac{1}{2}} - \mathbf{H}^{\frac{1}{2}}\mathbf{\Psi}\mathbf{H}^{\frac{1}{2}}\mathbf{\Omega}^{-1}\right)^\top\right]$$
$$+ \text{tr}(\mathbf{H}\mathbf{\Psi}) - \text{tr}\left(\mathbf{H}^{\frac{1}{2}}\mathbf{\Psi}\mathbf{H}^{\frac{1}{2}}\mathbf{\Omega}^{-1}\mathbf{H}^{\frac{1}{2}}\mathbf{\Psi}\mathbf{H}^{\frac{1}{2}}\right)$$

**Step 2: solve the global minimum.** Now we have

$$\mathcal{R}(\boldsymbol{\beta}, \mathbf{\Gamma}) - \sigma^2$$

$$= (\boldsymbol{\beta} - \boldsymbol{\beta}^*)^\top \left[(\mathbf{I}_d - \mathbf{\Gamma}\mathbf{H})^\top \mathbf{H}(\mathbf{I}_d - \mathbf{\Gamma}\mathbf{H}) + \frac{\text{tr}\left(\mathbf{H}\mathbf{\Gamma}^\top \mathbf{H}\mathbf{\Gamma}\right)}{M}\mathbf{H} + \frac{1}{M}\mathbf{H}\mathbf{\Gamma}^\top \mathbf{H}\mathbf{\Gamma}\mathbf{H}\right](\boldsymbol{\beta} - \boldsymbol{\beta}^*)$$

$$+ \text{tr}\left[\left(\mathbf{H}^{\frac{1}{2}}\mathbf{\Gamma}\mathbf{H}^{\frac{1}{2}} - \mathbf{H}^{\frac{1}{2}}\mathbf{\Psi}\mathbf{H}^{\frac{1}{2}}\mathbf{\Omega}^{-1}\right)\mathbf{\Omega}\left(\mathbf{H}^{\frac{1}{2}}\mathbf{\Gamma}\mathbf{H}^{\frac{1}{2}} - \mathbf{H}^{\frac{1}{2}}\mathbf{\Psi}\mathbf{H}^{\frac{1}{2}}\mathbf{\Omega}^{-1}\right)^\top\right]$$

$$+ \text{tr}(\mathbf{H}\mathbf{\Psi}) - \text{tr}\left(\mathbf{H}^{\frac{1}{2}}\mathbf{\Psi}\mathbf{H}^{\frac{1}{2}}\mathbf{\Omega}^{-1}\mathbf{H}^{\frac{1}{2}}\mathbf{\Psi}\mathbf{H}^{\frac{1}{2}}\right) \qquad (C.5)$$

$$\geq \text{tr}(\mathbf{H}\mathbf{\Psi}) - \text{tr}\left(\mathbf{H}^{\frac{1}{2}}\mathbf{\Psi}\mathbf{H}^{\frac{1}{2}}\mathbf{\Omega}^{-1}\mathbf{H}^{\frac{1}{2}}\mathbf{\Psi}\mathbf{H}^{\frac{1}{2}}\right)$$

$$= \text{tr}(\mathbf{H}\mathbf{\Psi}) - \text{tr}\left(\left(\mathbf{H}^{\frac{1}{2}}\mathbf{\Psi}\mathbf{H}^{\frac{1}{2}}\right)^2 \mathbf{\Omega}^{-1}\right),$$

where the last line comes from the fact that $\mathbf{\Omega}$ and $\mathbf{H}^{\frac{1}{2}}\mathbf{\Psi}\mathbf{H}^{\frac{1}{2}}$ commute. Taking infimum over all $\boldsymbol{\beta}, \mathbf{\Gamma}$ in the left hand side, we have

$$\inf_{f \in \mathcal{F}_{\text{GD-}\boldsymbol{\beta}}} \mathcal{R}(f) - \sigma^2 = \inf_{\boldsymbol{\beta}, \mathbf{\Gamma}} \mathcal{R}(\boldsymbol{\beta}, \mathbf{\Gamma}) - \sigma^2 \geq \text{tr}(\mathbf{H}\mathbf{\Psi}) - \text{tr}\left(\left(\mathbf{H}^{\frac{1}{2}}\mathbf{\Psi}\mathbf{H}^{\frac{1}{2}}\right)^2 \mathbf{\Omega}^{-1}\right).$$

On the other hand, we take

$$\boldsymbol{\beta} = \boldsymbol{\beta}^*, \quad \boldsymbol{\Gamma} = \boldsymbol{\Gamma}^* := \boldsymbol{\Psi} \mathbf{H}^{\frac{1}{2}} \boldsymbol{\Omega}^{-1} \mathbf{H}^{-\frac{1}{2}}, \tag{C.6}$$

where $\mathbf{H}^{\frac{1}{2}}$ is the principle square root of $\mathbf{H}$ and $\mathbf{H}^{-\frac{1}{2}}$ is its Moore-Penrose pseudo inverse (see notation part in Section A). Then, we have

$$\begin{aligned}
&\mathcal{R}\left(\boldsymbol{\beta}^*, \boldsymbol{\Gamma}^*\right) - \sigma^2 \\
&= \operatorname{tr}\left[\left(\mathbf{H}^{\frac{1}{2}} \boldsymbol{\Gamma}^* \mathbf{H}^{\frac{1}{2}} - \mathbf{H}^{\frac{1}{2}} \boldsymbol{\Psi} \mathbf{H}^{\frac{1}{2}} \boldsymbol{\Omega}^{-1}\right) \boldsymbol{\Omega} \left(\mathbf{H}^{\frac{1}{2}} \boldsymbol{\Gamma}^* \mathbf{H}^{\frac{1}{2}} - \mathbf{H}^{\frac{1}{2}} \boldsymbol{\Psi} \mathbf{H}^{\frac{1}{2}} \boldsymbol{\Omega}^{-1}\right)^{\top}\right] \\
&\quad + \operatorname{tr}\left(\mathbf{H}\boldsymbol{\Psi}\right) - \operatorname{tr}\left(\mathbf{H}^{\frac{1}{2}} \boldsymbol{\Psi} \mathbf{H}^{\frac{1}{2}} \boldsymbol{\Omega}^{-1} \mathbf{H}^{\frac{1}{2}} \boldsymbol{\Psi} \mathbf{H}^{\frac{1}{2}}\right).
\end{aligned}$$

Since

$$\begin{aligned}
\mathbf{H}^{\frac{1}{2}} \boldsymbol{\Gamma}^* \mathbf{H}^{\frac{1}{2}} - \mathbf{H}^{\frac{1}{2}} - \mathbf{H}^{\frac{1}{2}} \boldsymbol{\Psi} \mathbf{H}^{\frac{1}{2}} \boldsymbol{\Omega}^{-1} &= \mathbf{H}^{\frac{1}{2}} \boldsymbol{\Psi} \mathbf{H}^{\frac{1}{2}} \boldsymbol{\Omega}^{-1} \mathbf{H}^{-\frac{1}{2}} \mathbf{H}^{\frac{1}{2}} - \mathbf{H}^{\frac{1}{2}} \boldsymbol{\Psi} \mathbf{H}^{\frac{1}{2}} \boldsymbol{\Omega}^{-1} \\
&= \boldsymbol{\Omega}^{-1} \mathbf{H}^{\frac{1}{2}} \boldsymbol{\Psi} \mathbf{H}^{\frac{1}{2}} - \boldsymbol{\Omega}^{-1} \mathbf{H}^{\frac{1}{2}} \boldsymbol{\Psi} \mathbf{H}^{\frac{1}{2}} \mathbf{H}^{-\frac{1}{2}} \mathbf{H}^{\frac{1}{2}} \\
&\qquad\qquad (\boldsymbol{\Omega} \text{ and } \mathbf{H}^{\frac{1}{2}} \boldsymbol{\Psi} \mathbf{H}^{\frac{1}{2}} \text{ commute}) \\
&= \boldsymbol{\Omega}^{-1} \mathbf{H}^{\frac{1}{2}} \boldsymbol{\Psi} \mathbf{H}^{\frac{1}{2}} - \boldsymbol{\Omega}^{-1} \mathbf{H}^{\frac{1}{2}} \boldsymbol{\Psi} \mathbf{H}^{\frac{1}{2}} = \mathbf{0}_{d \times d}.. \\
&\qquad\qquad (\mathbf{H}^{\frac{1}{2}} \mathbf{H}^{-\frac{1}{2}} \mathbf{H}^{\frac{1}{2}} = \mathbf{H}^{\frac{1}{2}})
\end{aligned}$$

Therefore, we have

$$\mathcal{R}\left(\boldsymbol{\beta}^*, \boldsymbol{\Gamma}^*\right) - \sigma^2 = \operatorname{tr}\left(\mathbf{H}\boldsymbol{\Psi}\right) - \operatorname{tr}\left(\left(\mathbf{H}^{\frac{1}{2}} \boldsymbol{\Psi} \mathbf{H}^{\frac{1}{2}}\right)^2 \boldsymbol{\Omega}^{-1}\right) \geq \inf_{f \in \mathcal{F}_{\text{GD-}\boldsymbol{\beta}}} \mathcal{R}(f) - \sigma^2 = \inf_{\boldsymbol{\beta}, \boldsymbol{\Gamma}} \mathcal{R}\left(\boldsymbol{\beta}, \boldsymbol{\Gamma}\right) - \sigma^2.$$

Combining both directions, we conclude that

$$\inf_{f \in \mathcal{F}_{\text{GD-}\boldsymbol{\beta}}} \mathcal{R}(f) - \sigma^2 = \inf_{\boldsymbol{\beta}, \boldsymbol{\Gamma}} \mathcal{R}\left(\boldsymbol{\beta}, \boldsymbol{\Gamma}\right) - \sigma^2 = \operatorname{tr}\left(\mathbf{H}\boldsymbol{\Psi}\right) - \operatorname{tr}\left(\left(\mathbf{H}^{\frac{1}{2}} \boldsymbol{\Psi} \mathbf{H}^{\frac{1}{2}}\right)^2 \boldsymbol{\Omega}^{-1}\right). \tag{C.7}$$

**Step 3: identify all global minimizers**   Above we show that $\mathcal{R}\left(\boldsymbol{\beta}^*, \boldsymbol{\Gamma}^*\right)$ achieves the global minimum of the ICL risk over GD-$\boldsymbol{\beta}$ class. Now we will figure out the sufficient and necessary condition to achieve the global minimal risk. From the proof above, we know that for any $\boldsymbol{\beta} \in \mathbb{R}^d, \boldsymbol{\Gamma} \in \mathbb{R}^{d \times d}$, it holds that

$$\mathcal{R}\left(\boldsymbol{\beta}, \boldsymbol{\Gamma}\right) = \inf_{f \in \mathcal{F}_{\text{GD-}\boldsymbol{\beta}}} \mathcal{R}(f) \iff \begin{cases} V_1 = \left(\boldsymbol{\beta} - \boldsymbol{\beta}^*\right)^{\top} \mathbf{H}_{\boldsymbol{\Gamma}} \left(\boldsymbol{\beta} - \boldsymbol{\beta}^*\right) = 0 \\ \operatorname{tr}\left[\left(\mathbf{H}^{\frac{1}{2}} \boldsymbol{\Gamma} \mathbf{H}^{\frac{1}{2}} - \mathbf{H}^{\frac{1}{2}} \boldsymbol{\Psi} \mathbf{H}^{\frac{1}{2}} \boldsymbol{\Omega}^{-1}\right) \boldsymbol{\Omega} \left(\mathbf{H}^{\frac{1}{2}} \boldsymbol{\Gamma} \mathbf{H}^{\frac{1}{2}} - \mathbf{H}^{\frac{1}{2}} \boldsymbol{\Psi} \mathbf{H}^{\frac{1}{2}} \boldsymbol{\Omega}^{-1}\right)^{\top}\right] = 0 \end{cases}$$

Since $\boldsymbol{\Omega}$ is invertible, the second equation is equivalent to

$$\mathbf{H}^{\frac{1}{2}} \boldsymbol{\Gamma} \mathbf{H}^{\frac{1}{2}} - \mathbf{H}^{\frac{1}{2}} \boldsymbol{\Psi} \mathbf{H}^{\frac{1}{2}} \boldsymbol{\Omega}^{-1} = \mathbf{0}_{d \times d},$$

which is a linear system of $\boldsymbol{\Gamma}$. Since $\boldsymbol{\Gamma}^*$ is proved to be one solution, all solutions of this equation is

$$\boldsymbol{\Gamma} = \boldsymbol{\Gamma}^* + \left\{\mathbf{Z} \in \mathbb{R}^{d \times d} : \mathbf{H}^{\frac{1}{2}} \mathbf{Z} \mathbf{H}^{\frac{1}{2}} = \mathbf{0}_{d \times d}\right\} = \boldsymbol{\Gamma}^* + \operatorname{Im}\left(\mathbf{H}^{\otimes 2}\right),$$

where the last equation comes from Lemma H.6. Here, $+$ denotes Minkowski sum. This is defined as $A + B = \{a + b, a \in A, b \in B\}$ for two sets $A, B$ and $a + B = \{a\} + B$ for an entry $a$ and a set $B$. Under this condition, we know that

$$\mathbf{H}_{\boldsymbol{\Gamma}} = \mathbf{H} - \mathbf{H}\left(\boldsymbol{\Gamma}^* + \boldsymbol{\Gamma}_M^{*\top}\right)\mathbf{H} + \frac{\operatorname{tr}\left(\mathbf{H}\boldsymbol{\Gamma}_M^{*\top}\mathbf{H}\boldsymbol{\Gamma}^*\right)}{M}\mathbf{H} + \frac{M+1}{M}\mathbf{H}\boldsymbol{\Gamma}_M^{*\top}\mathbf{H}\boldsymbol{\Gamma}^*\mathbf{H}.$$

Consider the following inequality:

$$\mathbf{I}_d - \mathbf{H}^{\frac{1}{2}}\left(\boldsymbol{\Gamma}^* + \boldsymbol{\Gamma}^{*\top}\right)\mathbf{H}^{\frac{1}{2}} + \frac{\operatorname{tr}\left(\mathbf{H}\boldsymbol{\Gamma}_M^{*\top}\mathbf{H}\boldsymbol{\Gamma}^*\right)}{M}\mathbf{I}_d + \frac{M+1}{M}\mathbf{H}^{\frac{1}{2}}\boldsymbol{\Gamma}_M^{*\top}\mathbf{H}\boldsymbol{\Gamma}^*\mathbf{H}^{\frac{1}{2}}$$

$$\succeq \mathbf{I}_d - \mathbf{H}^{\frac{1}{2}}\left(\mathbf{\Gamma}^* + \mathbf{\Gamma}^{*\top}\right)\mathbf{H}^{\frac{1}{2}} + \frac{M+1}{M}\mathbf{H}^{\frac{1}{2}}\mathbf{\Gamma}_M^{*\top}\mathbf{H}\mathbf{\Gamma}^*\mathbf{H}^{\frac{1}{2}}$$

$$= \left(\sqrt{\frac{M}{M+1}}\mathbf{I}_d - \sqrt{\frac{M+1}{M}}\mathbf{H}^{\frac{1}{2}}\mathbf{\Gamma}^*\mathbf{H}^{\frac{1}{2}}\right)\cdot\left(\sqrt{\frac{M}{M+1}}\mathbf{I}_d - \sqrt{\frac{M+1}{M}}\mathbf{H}^{\frac{1}{2}}\mathbf{\Gamma}^*\mathbf{H}^{\frac{1}{2}}\right)^{\top} + \frac{M}{M+1}\mathbf{I}_d$$

$$\succ \mathbf{0}_{d\times d}.$$

Therefore, we have

$$(\boldsymbol{\beta} - \boldsymbol{\beta}^*)^{\top}\mathbf{H}_{\mathbf{\Gamma}}(\boldsymbol{\beta} - \boldsymbol{\beta}^*) = 0$$

$$\Longleftrightarrow \left[(\boldsymbol{\beta} - \boldsymbol{\beta}^*)^{\top}\mathbf{H}^{\frac{1}{2}}\right]\left(\mathbf{I}_d - \mathbf{H}^{\frac{1}{2}}\left(\mathbf{\Gamma}^* + \mathbf{\Gamma}^{*\top}\right)\mathbf{H}^{\frac{1}{2}} + \frac{\text{tr}\left(\mathbf{H}\mathbf{\Gamma}_M^{*\top}\mathbf{H}\mathbf{\Gamma}^*\right)}{M}\mathbf{I}_d\right.$$

$$\left. + \frac{M+1}{M}\mathbf{H}^{\frac{1}{2}}\mathbf{\Gamma}_M^{*\top}\mathbf{H}\mathbf{\Gamma}^*\mathbf{H}^{\frac{1}{2}}\right)\left[\mathbf{H}^{\frac{1}{2}}(\boldsymbol{\beta} - \boldsymbol{\beta}^*)\right] = 0$$

$$\Longleftrightarrow \mathbf{H}^{\frac{1}{2}}(\boldsymbol{\beta} - \boldsymbol{\beta}^*) = \mathbf{0}_d \Longleftrightarrow \boldsymbol{\beta} = \boldsymbol{\beta}^* + \text{null}\left(\mathbf{H}^{\frac{1}{2}}\right) \Longleftrightarrow \boldsymbol{\beta} = \boldsymbol{\beta}^* + \text{null}\left(\mathbf{H}\right),$$

where $\text{null}\left(\cdot\right)$ denotes the null space of a matrix and $+$ denotes the Minkowski sum. The last equivalence comes from Lemma H.6. Therefore, we conclude that the $f_{\boldsymbol{\beta},\mathbf{\Gamma}}$ achieves the minimal ICL risk in GD-$\boldsymbol{\beta}$ class if and only if

$$\boldsymbol{\beta} = \boldsymbol{\beta}^* + \text{null}\left(\mathbf{H}\right), \quad \mathbf{\Gamma} = \mathbf{\Gamma}^* + \text{Im}\left(\mathbf{H}^{\otimes 2}\right).$$

Specially, if $\mathbf{H}$ is positive definite, there is unique global minimizer in GD-$\boldsymbol{\beta}$ class with parameters

$$\boldsymbol{\beta} = \boldsymbol{\beta}^*, \quad \mathbf{\Gamma} = \mathbf{\Gamma}^*.$$

**Step 4: equivalence in the hypothesis class** Finally, we show when $\mathbf{H}$ is rank-deficient, any global minimizer actually corresponds to one single function in $\mathcal{F}_{\text{GD-}\boldsymbol{\beta}}$ almost surely. For an arbitrary global minimizer, we assume $\boldsymbol{\beta} = \boldsymbol{\beta}^* + \mathbf{h}, \mathbf{\Gamma} = \mathbf{\Gamma}^* + \mathbf{Z}$, where $\mathbf{h} \in \text{null}\left(\mathbf{H}\right), \mathbf{Z} \in \text{Im}\left(\mathbf{H}^{\otimes 2}\right)$. Suppose we have a prompt $\mathbf{X}, \mathbf{y}, \mathbf{x}, y$ which follows the Assumption 3.1 and the token matrix is formed by (3.1), we have

$$f_{\boldsymbol{\beta},\mathbf{\Gamma}}(\mathbf{E}) = \left\langle \boldsymbol{\beta} - \frac{\mathbf{\Gamma}}{M}\mathbf{X}^{\top}(\mathbf{X}\boldsymbol{\beta} - \mathbf{y}), \mathbf{x}\right\rangle$$

$$= \left\langle \boldsymbol{\beta}^* - \frac{\mathbf{\Gamma}^*}{M}\mathbf{X}^{\top}(\mathbf{X}\boldsymbol{\beta}^* - \mathbf{y}), \mathbf{x}\right\rangle$$

$$+ \mathbf{h}^{\top}\mathbf{x} - \frac{1}{M}\mathbf{x}^{\top}\mathbf{Z}\mathbf{X}^{\top}\mathbf{X}\boldsymbol{\beta}^* - \frac{1}{M}\mathbf{x}^{\top}\mathbf{Z}\mathbf{X}^{\top}\mathbf{X}\mathbf{h} - \frac{1}{M}\mathbf{x}^{\top}\mathbf{\Gamma}^*\mathbf{X}^{\top}\mathbf{X}\mathbf{h} + \mathbf{x}^{\top}\mathbf{Z}\mathbf{X}^{\top}\mathbf{y}.$$

Notice that $\mathbf{X}[i], \mathbf{x} \overset{\text{i.i.d.}}{\sim} \mathcal{N}\left(\mathbf{0}_d, \mathbf{H}\right)$, we have

$$\mathbb{E}\left(\mathbf{h}^{\top}\mathbf{x}\right)^2 = \mathbf{h}^{\top}\mathbf{H}\mathbf{h} = 0$$

since $\mathbf{h} \in \text{null}\left(\mathbf{H}\right)$. Therefore, we know $\mathbf{h}^{\top}\mathbf{x} = 0$ almost surely, and similarly, $\mathbf{X}\mathbf{h} = \mathbf{0}_d$ almost surely. Finally, we have

$$\mathbb{E}\left[\left(\frac{1}{M}\mathbf{x}^{\top}\mathbf{Z}\mathbf{X}^{\top}\right)\cdot\left(\frac{1}{M}\mathbf{x}^{\top}\mathbf{Z}\mathbf{X}^{\top}\right)^{\top}\right] = \frac{1}{M}\mathbb{E}\text{tr}\left[\mathbf{H}\mathbf{Z}\mathbf{H}\right] = 0,$$

which imples $\mathbf{x}^{\top}\mathbf{Z}\mathbf{X}^{\top} = \mathbf{0}_d$ almost surely. Therefore, we conclude

$$f_{\boldsymbol{\beta},\mathbf{\Gamma}}(\mathbf{E}) = f_{\boldsymbol{\beta}^*,\mathbf{\Gamma}^*}(\mathbf{E}) \quad \text{almost surely.}$$

$$\square$$

# D Proof of Theorem 5.3

*Proof.* Recall the definition of Linear Transformer Block class:

$$f_{\mathsf{LTB}} : \mathbb{R}^{(d+1)\times(M+1)} \to \mathbb{R}$$

$$\mathbf{E} \mapsto \left[ \mathbf{W}_2^\top \mathbf{W}_1 \left( \mathbf{E} + \mathbf{W}_P^\top \mathbf{W}_V \mathbf{E} M \frac{\mathbf{E}^\top \mathbf{W}_K^\top \mathbf{W}_Q \mathbf{E}}{M} \right) \right]_{-1,-1},$$

where $\mathbf{E}$ is the token matrix defined by (3.1). Let's take an arbitrary function $f$ in the LTB class with trainable matrices

$$\mathbf{W}_K, \ \mathbf{W}_Q \in \mathbb{R}^{d_k\times(d+1)}, \quad \mathbf{W}_P, \ \mathbf{W}_V \in \mathbb{R}^{d_v\times(d+1)}, \quad \mathbf{W}_1, \mathbf{W}_2 \in \mathbb{R}^{d_f\times(d+1)}.$$

Similar to the proof of the Theorem 4.1, we know that only the last row of $\mathbf{W}_2^\top \mathbf{W}_1$ and the first $d$-columns of $\mathbf{W}_K^\top \mathbf{W}_Q$ attend the prediction. Therefore, we denote

$$\mathbf{W}_2^\top \mathbf{W}_1 = \begin{pmatrix} * & * \\ \boldsymbol{\gamma}^\top & * \end{pmatrix}, \quad \mathbf{W}_2^\top \mathbf{W}_1 \mathbf{W}_P^\top \mathbf{W}_V = \begin{pmatrix} * & * \\ \mathbf{v}_{21}^\top & v_{-1} \end{pmatrix}, \quad \mathbf{W}_K^\top \mathbf{W}_Q = \begin{pmatrix} \mathbf{V}_{11} & * \\ \mathbf{v}_{12}^\top & * \end{pmatrix},$$

$$\text{(D.1)}$$

where $\boldsymbol{\gamma}, \mathbf{v}_{12}, \mathbf{v}_{21}, \in \mathbb{R}^d, v_{-1} \in \mathbb{R}, \mathbf{V}_{11} \in \mathbb{R}^{d\times d}$, and $*$ denotes entries that do not enter the prediction. Then, the prediction of LTB function can be written as

$$f(\mathbf{E}) = \boldsymbol{\gamma}^\top \mathbf{x} + \begin{pmatrix} \mathbf{v}_{21}^\top & v_{-1} \end{pmatrix} \cdot \frac{\mathbf{E} M_M \mathbf{E}^\top}{M} \cdot \begin{pmatrix} \mathbf{V}_{11} \\ \mathbf{v}_{12}^\top \end{pmatrix} \cdot \mathbf{x}$$

$$= \left[ \boldsymbol{\gamma}^\top + \mathbf{v}_{21}^\top \cdot \frac{1}{M}\mathbf{X}^\top \mathbf{X} \cdot \mathbf{V}_{11} + \mathbf{v}_{21}^\top \cdot \frac{1}{M}\mathbf{X}^\top \mathbf{y} \cdot \mathbf{v}_{12}^\top + v_{-1} \cdot \frac{1}{M}\mathbf{y}^\top \mathbf{X} \cdot \mathbf{V}_{11} + v_{-1} \cdot \frac{1}{M}\mathbf{y}^\top \mathbf{y} \cdot \mathbf{v}_{12}^\top \right] \cdot \mathbf{x}.$$

**Step 1: simplify the risk function.** We use $\widetilde{\boldsymbol{\beta}}$ to denote the task parameter. From the Assumption 3.1 and Definition A.1, we have

$$\mathbf{y} = \mathbf{X}\widetilde{\boldsymbol{\beta}} + \boldsymbol{\varepsilon}, \quad y = \left\langle \widetilde{\boldsymbol{\beta}}, \mathbf{x} \right\rangle + \varepsilon, \quad \widetilde{\boldsymbol{\beta}} \sim \mathcal{N}\left(\boldsymbol{\beta}^*, \boldsymbol{\Psi}\right), \quad \widetilde{\boldsymbol{\beta}} = \boldsymbol{\beta}^* + \boldsymbol{\Psi}^{\frac{1}{2}}\boldsymbol{\theta};$$

and

$$\mathbf{X}[i], \mathbf{x} \overset{\text{i.i.d.}}{\sim} \mathcal{N}\left(\mathbf{0}, \mathbf{H}\right), \quad \boldsymbol{\varepsilon}[i], \varepsilon \overset{\text{i.i.d.}}{\sim} \mathcal{N}\left(0, \sigma^2\right), \quad \boldsymbol{\theta} \sim \mathcal{N}\left(\mathbf{0}, \mathbf{I}_d\right).$$

Then the model output can be written as

$$f(\mathbf{E}) = \left[ \boldsymbol{\gamma}^\top + \mathbf{v}_{21}^\top \cdot \frac{1}{M}\mathbf{X}^\top \mathbf{X} \cdot \mathbf{V}_{11} + \mathbf{v}_{21}^\top \cdot \frac{1}{M}\mathbf{X}^\top \mathbf{y} \cdot \mathbf{v}_{12}^\top + v_{-1} \cdot \frac{1}{M}\mathbf{y}^\top \mathbf{X} \cdot \mathbf{V}_{11} + v_{-1} \cdot \frac{1}{M}\mathbf{y}^\top \mathbf{y} \cdot \mathbf{v}_{12}^\top \right] \cdot \mathbf{x}$$

$$= \left[ \boldsymbol{\gamma}^\top + \left( \mathbf{v}_{21} + v_{-1}\widetilde{\boldsymbol{\beta}} \right)^\top \cdot \frac{1}{M}\mathbf{X}^\top \mathbf{X} \cdot \left( \mathbf{V}_{11} + \widetilde{\boldsymbol{\beta}}\mathbf{v}_{12}^\top \right) \right] \cdot \mathbf{x}$$

$$+ \left[ \mathbf{v}_{21}^\top \cdot \frac{1}{M}\mathbf{X}^\top \boldsymbol{\varepsilon} \cdot \mathbf{v}_{12}^\top + v_{-1} \cdot \frac{1}{M}\boldsymbol{\varepsilon}^\top \mathbf{X} \cdot \mathbf{V}_{11} + v_{-1} \cdot \frac{2}{M}\boldsymbol{\varepsilon}^\top \mathbf{X}\widetilde{\boldsymbol{\beta}}\mathbf{v}_{12}^\top + \frac{1}{M}\boldsymbol{\varepsilon}^\top \boldsymbol{\varepsilon} \cdot v_{-1}\mathbf{v}_{12}^\top \right] \cdot \mathbf{x}.$$

To simplify the presentation, we denote

$$\mathbf{z}_1^\top = \left( \mathbf{v}_{21} + v_{-1}\widetilde{\boldsymbol{\beta}} \right)^\top \cdot \frac{1}{M}\mathbf{X}^\top \mathbf{X} \cdot \left( \mathbf{V}_{11} + \widetilde{\boldsymbol{\beta}}\mathbf{v}_{12}^\top \right),$$

$$\mathbf{z}_2^\top = \mathbf{v}_{21}^\top \cdot \frac{1}{M}\mathbf{X}^\top \boldsymbol{\varepsilon} \cdot \mathbf{v}_{12}^\top + v_{-1} \cdot \frac{1}{M}\boldsymbol{\varepsilon}^\top \mathbf{X} \cdot \mathbf{V}_{11} + v_{-1} \cdot \frac{2}{M}\boldsymbol{\varepsilon}^\top \mathbf{X}\widetilde{\boldsymbol{\beta}}\mathbf{v}_{12}^\top$$

$$\mathbf{z}_3^\top = \frac{1}{M}\boldsymbol{\varepsilon}^\top \boldsymbol{\varepsilon} \cdot v_{-1}\mathbf{v}_{12}^\top.$$

Since $\mathbf{x}, \mathbf{X}, \boldsymbol{\varepsilon}, \widetilde{\boldsymbol{\beta}}$ are independent, we have

$$\mathcal{R}\left(f\right) = \mathbb{E}\left( f(\mathbf{E}) - \left\langle \widetilde{\boldsymbol{\beta}}, \mathbf{x} \right\rangle - \varepsilon \right)^2$$

$$= \sigma^2 + \mathbb{E}\left(f(\mathbf{E}) - \left\langle \widetilde{\boldsymbol{\beta}}, \mathbf{x} \right\rangle\right)^2 \quad (\varepsilon \text{ is independent from other variables and zero-mean})$$

$$= \mathbb{E}\left[\left\langle \mathbf{z}_1 + \mathbf{z}_2 + \mathbf{z}_3 + \boldsymbol{\gamma} - \widetilde{\boldsymbol{\beta}}, \mathbf{x} \right\rangle^2\right] + \sigma^2$$

$$= \left\langle \mathbf{H}, \mathbb{E}\left(\mathbf{z}_1 + \mathbf{z}_2 + \mathbf{z}_3 + \boldsymbol{\gamma} - \widetilde{\boldsymbol{\beta}}\right)\left(\mathbf{z}_1 + \mathbf{z}_2 + \mathbf{z}_3 + \boldsymbol{\gamma} - \widetilde{\boldsymbol{\beta}}\right)^\top \right\rangle + \sigma^2.$$

Note that $\mathbf{z}_1$ does not contain $\varepsilon$, $\mathbf{z}_2$ is a linear form of $\varepsilon$, and $\mathbf{z}_3$ is a quadratic form of $\varepsilon$. Using $\varepsilon \sim \mathcal{N}(\mathbf{0}, \sigma^2 \mathbf{I}_d)$, we have $\mathbb{E}[(\mathbf{z}_1 + \boldsymbol{\gamma} - \widetilde{\boldsymbol{\beta}}) \cdot \mathbf{z}_2^\top] = \mathbf{0}$ and $\mathbb{E}[\mathbf{z}_2 \mathbf{z}_3^\top] = \mathbf{0}$. Therefore, we have

$$\mathcal{R}(f) - \sigma^2 = \underbrace{\left\langle \mathbf{H}, \mathbb{E}\left(\mathbf{z}_1 + \boldsymbol{\gamma} - \widetilde{\boldsymbol{\beta}}\right)\left(\mathbf{z}_1 + \boldsymbol{\gamma} - \widetilde{\boldsymbol{\beta}}\right)^\top \right\rangle}_{S_1'} + \underbrace{\left\langle \mathbf{H}, \mathbb{E}\mathbf{z}_2 \mathbf{z}_2^\top \right\rangle}_{S_2'} + \underbrace{\left\langle \mathbf{H}, \mathbb{E}\mathbf{z}_3 \mathbf{z}_3^\top \right\rangle}_{S_3'}$$

$$+ \underbrace{2\left\langle \mathbf{H}, \mathbb{E}\left(\mathbf{z}_1 + \boldsymbol{\gamma} - \widetilde{\boldsymbol{\beta}}\right)\mathbf{z}_3^\top \right\rangle}_{S_4'}. \tag{D.2}$$

**Step 2: compute the risk function.** Let's compute the risk function. Note that, the risk function is in the same form as the risk function of LSA layer, which is in the proof of Theorem 4.1 (see Section B and equation (B.2)). There are two differences: one is we replace $\mathbf{U}_{11}, \mathbf{u}_{12}, \mathbf{u}_{21}, u_{-1}$ here with $\mathbf{V}_{11}, \mathbf{v}_{12}, \mathbf{v}_{21}, v_{-1}$. The second difference is that we replace $\widetilde{\boldsymbol{\beta}}$ in (B.2) with $\widetilde{\boldsymbol{\beta}} - \boldsymbol{\gamma}$. This is equivalent to replacing $\boldsymbol{\beta}^*$ with $\boldsymbol{\beta}^* - \boldsymbol{\gamma}$, since $\widetilde{\boldsymbol{\beta}} - \boldsymbol{\gamma} \sim \mathcal{N}(\boldsymbol{\beta}^* - \boldsymbol{\gamma}, \boldsymbol{\Psi})$. Therefore, similar to (B.15), the risk function in (D.2) can be written as

$$\mathcal{R}(f) - \sigma^2$$

$$= \boldsymbol{b}^\top \left(\frac{M+1}{M}\mathbf{H}\boldsymbol{\Psi}\mathbf{H} + \frac{1}{M}\mathrm{tr}(\mathbf{H}\boldsymbol{\Psi})\mathbf{H}\right)\boldsymbol{b} \cdot \mathbf{v}_{12}^\top \mathbf{H}\mathbf{v}_{12} + \frac{\sigma^2}{M}\cdot \boldsymbol{b}^\top \mathbf{H}\boldsymbol{b} \cdot \mathbf{v}_{12}^\top \mathbf{H}\mathbf{v}_{12}$$

$$+ v_{-1}^2 \mathrm{tr}\left(\boldsymbol{A}^\top \left(\frac{M+1}{M}\mathbf{H}\boldsymbol{\Psi}\mathbf{H} + \frac{1}{M}\mathrm{tr}(\mathbf{H}\boldsymbol{\Psi})\mathbf{H}\right)\boldsymbol{A}\mathbf{H}\right) + \frac{\sigma^2}{M}\cdot \mathrm{tr}\left(\mathbf{H}\boldsymbol{A}\mathbf{H}\boldsymbol{A}^\top\right)$$

$$\underbrace{+ 2v_{-1}\boldsymbol{b}^\top \left[\frac{1}{M}\mathrm{tr}(\mathbf{H}\boldsymbol{\Psi})\mathbf{H} + \frac{M+1}{M}\mathbf{H}\boldsymbol{\Psi}\mathbf{H} + \frac{\sigma^2}{M}\cdot \mathbf{H}\right]\boldsymbol{A}\mathbf{H}\mathbf{v}_{12} - 2v_{-1}\mathrm{tr}\left(\mathbf{H}\boldsymbol{A}\mathbf{H}\boldsymbol{\Psi}\right) - 2\mathbf{v}_{12}^\top \mathbf{H}\boldsymbol{\Psi}\mathbf{H}\boldsymbol{b}}_{\mathrm{V}}$$

$$\underbrace{+ \frac{1}{M}\left[\boldsymbol{b}^\top \mathbf{H}\boldsymbol{b} \cdot \mathrm{tr}\left(\boldsymbol{A}^\top \mathbf{H}\boldsymbol{A}\mathbf{H}\right) + 4v_{-1}\cdot \boldsymbol{b}^\top \mathbf{H}\boldsymbol{\Psi}\mathbf{H}\boldsymbol{A}\mathbf{H}\mathbf{v}_{12} + 4v_{-1}^2 \cdot \mathrm{tr}\left(\mathbf{H}\boldsymbol{\Psi}\mathbf{H}\boldsymbol{\Psi}\right)\cdot \mathbf{v}_{12}^\top \mathbf{H}\mathbf{v}_{12}\right]}_{\mathrm{VI}}$$

$$\underbrace{+ \boldsymbol{b}^\top \left(\frac{M+1}{M}\mathbf{H}\boldsymbol{A}\mathbf{H}\boldsymbol{A}^\top \mathbf{H}\right)\boldsymbol{b} - 2\boldsymbol{b}^\top \mathbf{H}\boldsymbol{A}\mathbf{H}(\boldsymbol{\beta}^* - \boldsymbol{\gamma}) + 2v_{-1}\mathrm{tr}(\mathbf{H}\boldsymbol{\Psi})\cdot \boldsymbol{b}^\top \mathbf{H}\boldsymbol{A}\mathbf{H}\mathbf{v}_{12} + 2\sigma^2 \boldsymbol{b}^\top \mathbf{H}\boldsymbol{A}\mathbf{H}(v_{-1}\mathbf{v}_{12})}_{\mathrm{VII}}$$

$$+ (\boldsymbol{\beta}^* - \boldsymbol{\gamma})^\top \mathbf{H}(\boldsymbol{\beta}^* - \boldsymbol{\gamma}) - 2\left(\mathrm{tr}(\mathbf{H}\boldsymbol{\Psi}) + \sigma^2\right)\cdot (\boldsymbol{\beta}^* - \boldsymbol{\gamma})^\top \mathbf{H}(v_{-1}\mathbf{v}_{12}) + \mathrm{tr}(\mathbf{H}\boldsymbol{\Psi})$$

$$\underbrace{+ v_{-1}^2 \cdot \left[\left(2\mathrm{tr}(\mathbf{H}\boldsymbol{\Psi}\mathbf{H}\boldsymbol{\Psi}) + \frac{M+2}{M}\mathrm{tr}(\mathbf{H}\boldsymbol{\Psi})^2\right)\cdot \mathbf{v}_{12}^\top \mathbf{H}\mathbf{v}_{12} + \left(2 + \frac{4}{M}\right)\sigma^2 \mathrm{tr}(\mathbf{H}\boldsymbol{\Psi}) + \frac{M+2}{M}\sigma^4\right]\cdot \mathbf{v}_{12}^\top \mathbf{H}\mathbf{v}_{12}}_{\mathrm{VIII}}$$

$$= \mathrm{V} + \mathrm{VI} + \mathrm{VII} + \mathrm{VIII}, \tag{D.3}$$

where V, VI, VII, VIII are defined as above and $\boldsymbol{\Omega} = \frac{M+1}{M}\mathbf{H}^{\frac{1}{2}}\boldsymbol{\Psi}\mathbf{H}^{\frac{1}{2}} + \frac{\mathrm{tr}(\mathbf{H}\boldsymbol{\Psi}) + \sigma^2}{M}\cdot \mathbf{I}_d$, and $\boldsymbol{b} := \mathbf{v}_{21} + v_{-1}\boldsymbol{\beta}^* \in \mathbb{R}^d$, $\boldsymbol{A} := \mathbf{V}_{11} + \boldsymbol{\beta}^*\mathbf{v}_{12}^\top \in \mathbb{R}^{d\times d}$,

**Step 3: solve the global minimum of the risk function.** This is very similar to step 5 in Appendix B. The V term in (D.3) is actually equal to the I term in (B.15), so we have

$$
\mathrm{V} = \mathrm{tr}\left[\left(\mathbf{H}^{\frac{1}{2}}\mathbf{v}_{12}\boldsymbol{b}^{\top}\mathbf{H}^{\frac{1}{2}} + v_{-1}\mathbf{H}^{\frac{1}{2}}\boldsymbol{A}^{\top}\mathbf{H}^{\frac{1}{2}} - \mathbf{H}^{\frac{1}{2}}\boldsymbol{\Psi}\mathbf{H}^{\frac{1}{2}}\boldsymbol{\Omega}^{-1}\right)\boldsymbol{\Omega}\left(\mathbf{H}^{\frac{1}{2}}\mathbf{v}_{12}\boldsymbol{b}^{\top}\mathbf{H}^{\frac{1}{2}} + v_{-1}\mathbf{H}^{\frac{1}{2}}\boldsymbol{A}^{\top}\mathbf{H}^{\frac{1}{2}} - \mathbf{H}^{\frac{1}{2}}\boldsymbol{\Psi}\mathbf{H}^{\frac{1}{2}}\boldsymbol{\Omega}^{-1}\right)^{\top}\right]
$$

$$
- \mathrm{tr}\left(\mathbf{H}^{\frac{1}{2}}\boldsymbol{\Psi}\mathbf{H}^{\frac{1}{2}}\boldsymbol{\Omega}^{-1}\mathbf{H}^{\frac{1}{2}}\boldsymbol{\Psi}\mathbf{H}^{\frac{1}{2}}\right)
$$

$$
\geq -\mathrm{tr}\left(\mathbf{H}^{\frac{1}{2}}\boldsymbol{\Psi}\mathbf{H}^{\frac{1}{2}}\boldsymbol{\Omega}^{-1}\mathbf{H}^{\frac{1}{2}}\boldsymbol{\Psi}\mathbf{H}^{\frac{1}{2}}\right)
$$

$$
= -\mathrm{tr}\left(\left(\mathbf{H}^{\frac{1}{2}}\boldsymbol{\Psi}\mathbf{H}^{\frac{1}{2}}\right)^{2}\boldsymbol{\Omega}^{-1}\right),
$$

where the last line comes from the fact that $\boldsymbol{\Omega}$ and $\mathbf{H}^{\frac{1}{2}}\boldsymbol{\Psi}\mathbf{H}^{\frac{1}{2}}$ commute. For the same reason, we know that the $VI$ term above is equal to the II term in (B.15), which implies $\mathrm{VI} \geq 0$, since

$$
4v_{-1} \cdot \boldsymbol{b}^{\top}\mathbf{H}\boldsymbol{\Psi}\mathbf{H}\boldsymbol{A}\mathbf{H}\mathbf{v}_{12} \geq -4\left\|\boldsymbol{b}^{\top}\mathbf{H}^{\frac{1}{2}}\right\|_{F} \cdot \left\|v_{-1}\mathbf{H}^{\frac{1}{2}}\boldsymbol{\Psi}\mathbf{H}^{\frac{1}{2}}\right\|_{F} \cdot \left\|\mathbf{H}^{\frac{1}{2}}\boldsymbol{A}\mathbf{H}^{\frac{1}{2}}\right\|_{F} \cdot \left\|\mathbf{H}^{\frac{1}{2}}\mathbf{v}_{12}\right\|_{F}
$$

$$
\geq -\left\|\boldsymbol{b}^{\top}\mathbf{H}^{\frac{1}{2}}\right\|_{F}^{2}\left\|\mathbf{H}^{\frac{1}{2}}\boldsymbol{A}\mathbf{H}^{\frac{1}{2}}\right\|_{F}^{2} - 4\left\|v_{-1}\mathbf{H}^{\frac{1}{2}}\boldsymbol{\Psi}\mathbf{H}^{\frac{1}{2}}\right\|_{F}^{2}\left\|\mathbf{H}^{\frac{1}{2}}\mathbf{v}_{12}\right\|_{F}^{2}
$$

$$
= -\boldsymbol{b}^{\top}\mathbf{H}\boldsymbol{b} \cdot \mathrm{tr}\left(\boldsymbol{A}^{\top}\mathbf{H}\boldsymbol{A}\mathbf{H}\right) - 4v_{-1}^{2} \cdot \mathrm{tr}\left(\mathbf{H}\boldsymbol{\Psi}\mathbf{H}\boldsymbol{\Psi}\right) \cdot \mathbf{v}_{12}^{\top}\mathbf{H}\mathbf{v}_{12}, \quad \text{(D.4)}
$$

where the last line comes from the fact that $\|\boldsymbol{A}\|_{F}^{2} = \mathrm{tr}\left(\boldsymbol{A}\boldsymbol{A}^{\top}\right)$ for any matrix $\boldsymbol{A}$. The term VII above is equal to III term in (B.15), except that we replace $\boldsymbol{\beta}^{*}$ with $\boldsymbol{\beta}^{*} - \boldsymbol{\gamma}$. Therefore, we have

$$
\mathrm{VII} + \frac{M}{M+1}\left(\boldsymbol{\beta}^{*} - \boldsymbol{\gamma} - \left(\mathrm{tr}\left(\mathbf{H}\boldsymbol{\Psi}\right) + \sigma^{2}\right)v_{-1}\mathbf{v}_{12}\right)^{\top}\mathbf{H}\left(\boldsymbol{\beta}^{*} - \boldsymbol{\gamma} - \left(\mathrm{tr}\left(\mathbf{H}\boldsymbol{\Psi}\right) + \sigma^{2}\right)v_{-1}\mathbf{v}_{12}\right)
$$

$$
= \left[\boldsymbol{A}^{\top}\mathbf{H}\boldsymbol{b} - \frac{M}{M+1}\left(\boldsymbol{\beta}^{*} - \boldsymbol{\gamma} - \left(\mathrm{tr}\left(\mathbf{H}\boldsymbol{\Psi}\right) + \sigma^{2}\right)v_{-1}\mathbf{v}_{12}\right)\right]^{\top}\left(\frac{M+1}{M}\mathbf{H}\right)
$$

$$
\cdot \left[\boldsymbol{A}^{\top}\mathbf{H}\boldsymbol{b} - \frac{M}{M+1}\left(\boldsymbol{\beta}^{*} - \boldsymbol{\gamma} - \left(\mathrm{tr}\left(\mathbf{H}\boldsymbol{\Psi}\right) + \sigma^{2}\right)v_{-1}\mathbf{v}_{12}\right)\right]
$$

$$
\geq 0, \tag{D.5}
$$

which implies

$$
\mathrm{VII} \geq -\frac{M}{M+1}\left(\boldsymbol{\beta}^{*} - \boldsymbol{\gamma} - \left(\mathrm{tr}\left(\mathbf{H}\boldsymbol{\Psi}\right) + \sigma^{2}\right)v_{-1}\mathbf{v}_{12}\right)^{\top}\mathbf{H}\left(\boldsymbol{\beta}^{*} - \boldsymbol{\gamma} - \left(\mathrm{tr}\left(\mathbf{H}\boldsymbol{\Psi}\right) + \sigma^{2}\right)v_{-1}\mathbf{v}_{12}\right).
$$

Combining the three parts above, one has

$$
\mathcal{R}\left(f\right) - \sigma^{2}
$$

$$
\geq \mathrm{VIII} - \mathrm{tr}\left(\left(\mathbf{H}^{\frac{1}{2}}\boldsymbol{\Psi}\mathbf{H}^{\frac{1}{2}}\right)^{2}\boldsymbol{\Omega}^{-1}\right)
$$

$$
- \frac{M}{M+1}\left(\boldsymbol{\beta}^{*} - \boldsymbol{\gamma} - \left(\mathrm{tr}\left(\mathbf{H}\boldsymbol{\Psi}\right) + \sigma^{2}\right)v_{-1}\mathbf{v}_{12}\right)^{\top}\mathbf{H}\left(\boldsymbol{\beta}^{*} - \boldsymbol{\gamma} - \left(\mathrm{tr}\left(\mathbf{H}\boldsymbol{\Psi}\right) + \sigma^{2}\right)v_{-1}\mathbf{v}_{12}\right)
$$

$$
= \underbrace{\mathrm{tr}\left(\mathbf{H}\boldsymbol{\Psi}\right) - \mathrm{tr}\left(\left(\mathbf{H}^{\frac{1}{2}}\boldsymbol{\Psi}\mathbf{H}^{\frac{1}{2}}\right)^{2}\boldsymbol{\Omega}^{-1}\right)}_{\inf_{f \in \mathcal{F}_{\mathrm{GD}\text{-}\boldsymbol{\beta}}}\mathcal{R}(f) - \sigma^{2}} + \frac{1}{M+1}\left(\boldsymbol{\beta}^{*} - \boldsymbol{\gamma}\right)^{\top}\mathbf{H}\left(\boldsymbol{\beta}^{*} - \boldsymbol{\gamma}\right)
$$

$$
- \frac{2\left(\mathrm{tr}\left(\mathbf{H}\boldsymbol{\Psi}\right) + \sigma^{2}\right)}{M+1}\left(\boldsymbol{\beta}^{*} - \boldsymbol{\gamma}\right)^{\top}\mathbf{H}\left(v_{-1}\mathbf{v}_{12}\right)
$$

$$
+ \left[2\mathrm{tr}\left(\mathbf{H}\boldsymbol{\Psi}\mathbf{H}\boldsymbol{\Psi}\right) + \frac{3M+2}{M(M+1)}\left(\mathrm{tr}\left(\mathbf{H}\boldsymbol{\Psi}\right) + \sigma^{2}\right)^{2}\right]\left(v_{-1}\mathbf{v}_{12}\right)^{\top}\mathbf{H}\left(v_{-1}\mathbf{v}_{12}\right).
$$

Here, the global minimum of GD-$\boldsymbol{\beta}$ class is taken from Theorem 5.2, whose proof does not reply on the proof here. Therefore, one has

$$\mathcal{R}(f) - \inf_{f \in \mathcal{F}_{\text{GD-}\boldsymbol{\beta}}} \mathcal{R}(f) \geq \frac{1}{M+1} (\boldsymbol{\beta}^* - \boldsymbol{\gamma})^\top \mathbf{H} (\boldsymbol{\beta}^* - \boldsymbol{\gamma}) - \frac{2 \left( \text{tr}(\mathbf{H\Psi}) + \sigma^2 \right)}{M+1} (\boldsymbol{\beta}^* - \boldsymbol{\gamma})^\top \mathbf{H} (v_{-1}\mathbf{v}_{12})$$
$$+ \left[ 2\text{tr}(\mathbf{H\Psi H\Psi}) + \frac{3M+2}{M(M+1)} \left( \text{tr}(\mathbf{H\Psi}) + \sigma^2 \right)^2 \right] (v_{-1}\mathbf{v}_{12})^\top \mathbf{H} (v_{-1}\mathbf{v}_{12}).$$

The right hand side in the above inequality is a quadratic function of $v_{-1}\mathbf{v}_{12}$, so we can take its global minimizer:

$$v_{-1}\mathbf{v}_{12} = \frac{\frac{\left( \text{tr}(\mathbf{H\Psi}) + \sigma^2 \right)}{M+1}}{2\text{tr}(\mathbf{H\Psi H\Psi}) + \frac{3M+2}{M(M+1)} \left( \text{tr}(\mathbf{H\Psi}) + \sigma^2 \right)^2} (\boldsymbol{\beta}^* - \boldsymbol{\gamma})$$

to lower bound the risk gap as

$$\mathcal{R}(f) - \inf_{f \in \mathcal{F}_{\text{GD-}\boldsymbol{\beta}}} \mathcal{R}(f) \geq \left[ \frac{1}{M+1} - \frac{\frac{\left( \text{tr}(\mathbf{H\Psi}) + \sigma^2 \right)^2}{(M+1)^2}}{2\text{tr}(\mathbf{H\Psi H\Psi}) + \frac{3M+2}{M(M+1)} \left( \text{tr}(\mathbf{H\Psi}) + \sigma^2 \right)^2} \right] \|\boldsymbol{\beta}^* - \boldsymbol{\gamma}\|_{\mathbf{H}}^2 \geq 0.$$

Note that, taking infimum on the left hand side, we get

$$\inf_{f \in \mathcal{F}_{\text{LTB}}} \mathcal{R}(f) - \inf_{f \in \mathcal{F}_{\text{GD-}\boldsymbol{\beta}}} \mathcal{R}(f) \geq 0.$$

On the other hand, in the main text we have showed that $\mathcal{F}_{\text{GD-}\boldsymbol{\beta}} \subset \mathcal{F}_{\text{LTB}}$, which implies

$$\inf_{f \in \mathcal{F}_{\text{LTB}}} \mathcal{R}(f) - \inf_{f \in \mathcal{F}_{\text{GD-}\boldsymbol{\beta}}} \mathcal{R}(f) \leq 0.$$

Therefore, we conclude

$$\inf_{f \in \mathcal{F}_{\text{LTB}}} \mathcal{R}(f) = \inf_{f \in \mathcal{F}_{\text{GD-}\boldsymbol{\beta}}} \mathcal{R}(f) = \text{tr}(\mathbf{H\Psi}) - \text{tr}\left( \left( \mathbf{H}^{\frac{1}{2}} \mathbf{\Psi} \mathbf{H}^{\frac{1}{2}} \right)^2 \mathbf{\Omega}^{-1} \right),$$

where the last equation is from Theorem 5.2.

**Step 4: sufficient and necessary conditions for global minimizers.** Let's now verify the conditions for the global minimizers. For a function in LTB class, the sufficient and necessary condition for it to be a global minimizer is that inequalities in the above step all hold, which are

$$\mathbf{H}^{\frac{1}{2}} \mathbf{v}_{12} \boldsymbol{b}^\top \mathbf{H}^{\frac{1}{2}} + v_{-1} \mathbf{H}^{\frac{1}{2}} \boldsymbol{A}^\top \mathbf{H}^{\frac{1}{2}} = \mathbf{H}^{\frac{1}{2}} \mathbf{\Psi} \mathbf{H}^{\frac{1}{2}} \mathbf{\Omega}^{-1}, \tag{D.6}$$

$$\left\| \boldsymbol{b}^\top \mathbf{H}^{\frac{1}{2}} \right\|_F^2 \left\| \mathbf{H}^{\frac{1}{2}} \boldsymbol{A} \mathbf{H}^{\frac{1}{2}} \right\|_F^2 = 4 \left\| v_{-1} \mathbf{H}^{\frac{1}{2}} \mathbf{\Psi} \mathbf{H}^{\frac{1}{2}} \right\|_F^2 \left\| \mathbf{H}^{\frac{1}{2}} \mathbf{v}_{12} \right\|_F^2, \tag{D.7}$$

$$v_{-1} \cdot \boldsymbol{b}^\top \mathbf{H} \mathbf{\Psi} \mathbf{H} \boldsymbol{A} \mathbf{H} \mathbf{v}_{12} = \left\| \boldsymbol{b}^\top \mathbf{H}^{\frac{1}{2}} \right\|_F \cdot \left\| v_{-1} \mathbf{H}^{\frac{1}{2}} \mathbf{\Psi} \mathbf{H}^{\frac{1}{2}} \right\|_F \cdot \left\| \mathbf{H}^{\frac{1}{2}} \boldsymbol{A} \mathbf{H}^{\frac{1}{2}} \right\|_F \cdot \left\| \mathbf{H}^{\frac{1}{2}} \mathbf{v}_{12} \right\|_F, \tag{D.8}$$

$$\mathbf{H}^{\frac{1}{2}} \left[ \boldsymbol{A}^\top \mathbf{H} \boldsymbol{b} - \frac{M}{M+1} \left( \boldsymbol{\beta}^* - \boldsymbol{\gamma} - \left( \text{tr}(\mathbf{H\Psi}) + \sigma^2 \right) v_{-1} \mathbf{v}_{12} \right) \right] = \mathbf{0}_{d \times d}, \tag{D.9}$$

$$v_{-1} \mathbf{v}_{12} = \frac{\frac{\left( \text{tr}(\mathbf{H\Psi}) + \sigma^2 \right)}{M+1}}{2\text{tr}(\mathbf{H\Psi H\Psi}) + \frac{3M+2}{M(M+1)} \left( \text{tr}(\mathbf{H\Psi}) + \sigma^2 \right)^2} (\boldsymbol{\beta}^* - \boldsymbol{\gamma}) \tag{D.10}$$

$$\|\boldsymbol{\beta}^* - \boldsymbol{\gamma}\|_{\mathbf{H}} = 0. \tag{D.11}$$

Let's first verify the necessary conditions of the system defined above. From (D.11) and Lemma H.5, we know $\boldsymbol{\gamma} \in \boldsymbol{\beta}^* + \text{null}\left( \mathbf{H}^{\frac{1}{2}} \right) = \boldsymbol{\beta}^* + \text{null}(\mathbf{H})$. Then, we have $v_{-1}\mathbf{v}_{12} \in \text{null}(\mathbf{H})$. Then, in (D.7), we know

$$\left\| \boldsymbol{b}^\top \mathbf{H}^{\frac{1}{2}} \right\|_F^2 \left\| \mathbf{H}^{\frac{1}{2}} \boldsymbol{A} \mathbf{H}^{\frac{1}{2}} \right\|_F^2 = 4 \left\| v_{-1} \mathbf{H}^{\frac{1}{2}} \mathbf{\Psi} \mathbf{H}^{\frac{1}{2}} \right\|_F^2 \left\| \mathbf{H}^{\frac{1}{2}} \mathbf{v}_{12} \right\|_F^2 = 4 \left\| \mathbf{H}^{\frac{1}{2}} \mathbf{\Psi} \mathbf{H}^{\frac{1}{2}} \right\|_F^2 \left\| v_{-1} \mathbf{H}^{\frac{1}{2}} \mathbf{v}_{12} \right\|_F^2 = 0,$$

which implies either $\mathbf{H}^{\frac{1}{2}}\boldsymbol{b} = \mathbf{0}_d$ or $\mathbf{H}^{\frac{1}{2}}\boldsymbol{A}\mathbf{H}^{\frac{1}{2}} = \mathbf{0}_{d\times d}$. If we assume $\mathbf{H}^{\frac{1}{2}}\boldsymbol{A}\mathbf{H}^{\frac{1}{2}} = \mathbf{0}_{d\times d}$, then (D.6) implies $\mathbf{H}^{\frac{1}{2}}\mathbf{v}_{12}\boldsymbol{b}^{\top}\mathbf{H}^{\frac{1}{2}} = \mathbf{H}^{\frac{1}{2}}\boldsymbol{\Psi}\mathbf{H}^{\frac{1}{2}}\boldsymbol{\Omega}^{-1}$. From our assumption, we know $\mathsf{rank}(\mathbf{H}^{\frac{1}{2}}\boldsymbol{\Psi}^{\frac{1}{2}}) \geq 2$, which implies $\mathsf{rank}(\mathbf{H}^{\frac{1}{2}}\boldsymbol{\Psi}\mathbf{H}^{\frac{1}{2}}) \geq 2$, since for any matrix $\boldsymbol{Z}$, it holds that $\mathsf{rank}(\boldsymbol{Z}) = \mathsf{rank}(\boldsymbol{Z}\boldsymbol{Z}^{\top})$. Since multiplication by an invertible matrix does not change the rank, we know $\mathsf{rank}\left(\mathbf{H}^{\frac{1}{2}}\boldsymbol{\Psi}\mathbf{H}^{\frac{1}{2}}\boldsymbol{\Omega}^{-1}\right) \geq 2$, while $\mathbf{H}^{\frac{1}{2}}\mathbf{v}_{12}\boldsymbol{b}^{\top}\mathbf{H}^{\frac{1}{2}}$ is a matrix of rank at most one, which contradicts with (D.6). Therefore, we have

$$\mathbf{H}^{\frac{1}{2}}\boldsymbol{b} = \mathbf{0}, \tag{D.12}$$

$$v_{-1}\mathbf{H}^{\frac{1}{2}}\boldsymbol{A}^{\top}\mathbf{H}^{\frac{1}{2}} = \mathbf{H}^{\frac{1}{2}}\boldsymbol{\Psi}\mathbf{H}^{\frac{1}{2}}\boldsymbol{\Omega}^{-1}. \tag{D.13}$$

Recall in Theorem 5.2, we have defined

$$\boldsymbol{\Gamma}^{*} := \boldsymbol{\Psi}\mathbf{H}^{\frac{1}{2}}\boldsymbol{\Omega}^{-1}\mathbf{H}^{-\frac{1}{2}}.$$

Simple calculation shows

$$\mathbf{H}^{\frac{1}{2}}\boldsymbol{\Gamma}^{*}\mathbf{H}^{\frac{1}{2}} = \mathbf{H}^{\frac{1}{2}}\boldsymbol{\Psi}\mathbf{H}^{\frac{1}{2}}\boldsymbol{\Omega}^{-1}\mathbf{H}^{-\frac{1}{2}}\mathbf{H}^{\frac{1}{2}} = \boldsymbol{\Omega}^{-1}\mathbf{H}^{\frac{1}{2}}\boldsymbol{\Psi}\mathbf{H}^{\frac{1}{2}}\mathbf{H}^{-\frac{1}{2}}\mathbf{H}^{\frac{1}{2}} = \boldsymbol{\Omega}^{-1}\mathbf{H}^{\frac{1}{2}}\boldsymbol{\Psi}\mathbf{H}^{\frac{1}{2}} = \mathbf{H}^{\frac{1}{2}}\boldsymbol{\Psi}\mathbf{H}^{\frac{1}{2}}\boldsymbol{\Omega}^{-1},$$

where the second and the last equalities come from the fact that $\boldsymbol{\Omega}$ and $\mathbf{H}^{\frac{1}{2}}\boldsymbol{\Psi}\mathbf{H}^{\frac{1}{2}}$ commute, and the third equality comes from the property of Moor Penrose pseudo-inverse. Therefore, one solution of (D.13) is $v_{-1}\boldsymbol{A} = \boldsymbol{\Gamma}^{*\top}$. Since (D.13) is a linear equation to $v_{-1}\boldsymbol{A}$, we know its full solution is

$$v_{-1}\boldsymbol{A} \in \boldsymbol{\Gamma}^{*\top} + \left\{\boldsymbol{Z} : \mathbf{H}^{\frac{1}{2}}\boldsymbol{Z}\mathbf{H}^{\frac{1}{2}} = \mathbf{0}_{d\times d}\right\} = \boldsymbol{\Gamma}^{*\top} + \mathsf{Im}\left(\mathbf{H}^{\otimes 2}\right).$$

The solution to (D.12) is

$$\mathbf{v}_{21} = -v_{-1}\boldsymbol{\beta}^{*} + \mathsf{null}\left(\mathbf{H}^{\frac{1}{2}}\right) = -v_{-1}\boldsymbol{\beta}^{*} + \mathsf{null}\left(\mathbf{H}\right).$$

Therefore, we know the necessary conditions for (D.6) to (D.11) are

$$\begin{cases} v_{-1} \neq 0, \\ v_{-1}\mathbf{v}_{12} \in \mathsf{null}\left(\mathbf{H}\right), \\ \mathbf{v}_{21} = -v_{-1}\boldsymbol{\beta}^{*} + \mathsf{null}\left(\mathbf{H}\right), \\ v_{-1}\mathbf{V}_{11} \in \boldsymbol{\Gamma}^{*\top} - v_{-1}\boldsymbol{\beta}^{*}\mathbf{v}_{12}^{\top} + \mathsf{Im}\left(\mathbf{H}^{\otimes 2}\right), \\ \boldsymbol{\gamma} = \boldsymbol{\beta}^{*} + \mathsf{null}\left(\mathbf{H}\right) \end{cases} \tag{D.14}$$

It is easy to verify these equations above are also sufficient conditions of (D.6) to (D.11) by directly replacing each variable with its value and validating equaions from (D.6) to (D.11).

Specially, if $\mathbf{H}$ is positive definite and hence, invertible, the global minimizer is unique up to a scaling to $v_{-1}$:

$$v_{-1} \neq 0, \quad \mathbf{v}_{12} = \mathbf{0}_d, \quad \mathbf{v}_{21} = -v_{-1}\boldsymbol{\beta}^{*}, \quad \mathbf{V}_{11} = \frac{1}{v_{-1}} \cdot \boldsymbol{\Gamma}^{*\top}, \quad \boldsymbol{\gamma} = \boldsymbol{\beta}^{*}. \tag{D.15}$$

**Step 5: equivalence in the hypothesis class.** Finally, we will prove that any global minimizer of $\mathcal{F}_{\mathsf{LTB}}$ is actually equivalent to one single function in $\mathcal{F}_{\mathsf{LTB}}$ almost surely. More concretely, let's take a function $f \in \mathcal{F}_{\mathsf{LTB}}$ with parameters $\mathbf{W}_K$, $\mathbf{W}_Q$, $\mathbf{W}_P$, $\mathbf{W}_V$, $\mathbf{W}_1$, $\mathbf{W}_2$ and recall the parameter transformation in (D.1). We assume equations in (D.14) and Assumption 3.1 hold. Then, for any vector $\mathbf{a} \in \mathsf{null}\left(\mathbf{H}\right)$, one has $\mathbf{x}^{\top}\mathbf{a} = \mathbf{x}_i^{\top}\mathbf{a} = 0$ since $\mathbf{x}, \mathbf{x}_i \overset{\text{i.i.d.}}{\sim} \mathcal{N}\left(\mathbf{0}_d, \mathbf{H}\right)$. We denote

$$\mathbf{v}_{21} = -v_{-1}\boldsymbol{\beta}^{*} + \mathbf{a}_1, \quad v_{-1}\mathbf{V}_{11} = \boldsymbol{\Gamma}^{*\top} - v_{-1}\boldsymbol{\beta}^{*}\mathbf{v}_{12}^{\top} + \boldsymbol{Z}, \quad \boldsymbol{\gamma} = \boldsymbol{\beta}^{*} + \mathbf{a}_2,$$

where $\mathbf{a}_1, \mathbf{a}_2 \in \mathbf{H}, \mathbf{H}^{\frac{1}{2}}\boldsymbol{Z}\mathbf{H}^{\frac{1}{2}} = \mathbf{0}_{d\times d}$. Then, we have

$$f(\mathbf{E}) = \left[\mathbf{W}_2^{\top}\mathbf{W}_1\left(\mathbf{E} + \mathbf{W}_P^{\top}\mathbf{W}_V\mathbf{E}\mathbf{M}\frac{\mathbf{E}^{\top}\mathbf{W}_K^{\top}\mathbf{W}_Q\mathbf{E}}{M}\right)\right]_{-1,-1}$$

$$= (\boldsymbol{\beta}^* + \mathbf{a}_2)^\top \mathbf{x} + \left( -v_{-1} \boldsymbol{\beta}^{*\top} + \mathbf{a}_1^\top \quad v_{-1} \right) \cdot \frac{1}{M} \begin{pmatrix} \mathbf{X}^\top \mathbf{X} & \mathbf{X}^\top \mathbf{y} \\ \mathbf{y}^\top \mathbf{X} & \mathbf{y}^\top \mathbf{y} \end{pmatrix} \cdot \begin{pmatrix} \mathbf{V}_{11} \\ \mathbf{v}_{12}^\top \end{pmatrix} \cdot \mathbf{x}$$

$$= \boldsymbol{\beta}^{*\top} \mathbf{x} + \left( -\boldsymbol{\beta}^{*\top} \quad 1 \right) \cdot \frac{1}{M} \begin{pmatrix} \mathbf{X}^\top \mathbf{X} & \mathbf{X}^\top \mathbf{y} \\ \mathbf{y}^\top \mathbf{X} & \mathbf{y}^\top \mathbf{y} \end{pmatrix} \cdot \begin{pmatrix} v_{-1} \mathbf{V}_{11} \mathbf{x} \\ v_{-1} \mathbf{v}_{12}^\top \mathbf{x} \end{pmatrix} \qquad (\mathbf{X} \mathbf{a}_1 = \mathbf{0}, \mathbf{a}_2^\top \mathbf{x} = 0)$$

$$= \boldsymbol{\beta}^{*\top} \mathbf{x} + \left( -\boldsymbol{\beta}^{*\top} \quad 1 \right) \cdot \frac{1}{M} \begin{pmatrix} \mathbf{X}^\top \mathbf{X} & \mathbf{X}^\top \mathbf{y} \\ \mathbf{y}^\top \mathbf{X} & \mathbf{y}^\top \mathbf{y} \end{pmatrix} \cdot \begin{pmatrix} \boldsymbol{\Gamma}^{*\top} \mathbf{x} \\ \mathbf{0}_d^\top \end{pmatrix} \qquad (\mathbf{v}_{12}^\top \mathbf{x} = 0, \mathbf{X} \mathbf{V}_{11} \mathbf{x} = \mathbf{0})$$

$$= \left\langle \boldsymbol{\beta}^* - \frac{\boldsymbol{\Gamma}^*}{M} \mathbf{X}^\top (\mathbf{X} \boldsymbol{\beta} - \mathbf{y}), \mathbf{x} \right\rangle = f_{\boldsymbol{\beta}^*, \boldsymbol{\Gamma}^*} (\mathbf{E}),$$

where $f_{\boldsymbol{\beta}^*, \boldsymbol{\Gamma}^*} (\cdot)$ is the GD-$\boldsymbol{\beta}$ function defined in (5.1). $\qquad\qquad \square$

# E   Proof of Corollary 6.2

*Proof.* We denote $\phi_1 \geq \phi_2 \geq ... \geq \phi_d \geq 0$ are ordered eigenvalues of $\mathbf{\Psi}^{\frac{1}{2}}\mathbf{H}\mathbf{\Psi}^{\frac{1}{2}}$. From Theorem 5.2, we have

$$\inf_{f \in \mathcal{F}_{\text{GD-}\boldsymbol{\beta}}} \mathcal{R}(f) - \sigma^2 = \text{tr}\left(\mathbf{H}^{\frac{1}{2}}\mathbf{\Psi}\mathbf{H}^{\frac{1}{2}}\right) - \text{tr}\left(\left(\mathbf{H}^{\frac{1}{2}}\mathbf{\Psi}\mathbf{H}^{\frac{1}{2}}\right)^2 \mathbf{\Omega}^{-1}\right),$$

where

$$\mathbf{\Omega} := \frac{M+1}{M}\mathbf{H}^{\frac{1}{2}}\mathbf{\Psi}\mathbf{H}^{\frac{1}{2}} + \frac{\text{tr}\left(\mathbf{H}\mathbf{\Psi}\right) + \sigma^2}{M} \cdot \mathbf{I}_d.$$

Therefore, we have

$$\inf_{f \in \mathcal{F}_{\text{GD-}\boldsymbol{\beta}}} \mathcal{R}(f) - \sigma^2 = \text{tr}\left(\mathbf{H}^{\frac{1}{2}}\mathbf{\Psi}\mathbf{H}^{\frac{1}{2}} \cdot \left(\mathbf{\Omega} - \mathbf{H}^{\frac{1}{2}}\mathbf{\Psi}\mathbf{H}^{\frac{1}{2}}\right)\mathbf{\Omega}^{-1}\right)$$

$$= \frac{1}{M}\text{tr}\left(\mathbf{\Omega}^{-1}\mathbf{H}^{\frac{1}{2}}\mathbf{\Psi}\mathbf{H}^{\frac{1}{2}} \cdot \left(\mathbf{H}^{\frac{1}{2}}\mathbf{\Psi}\mathbf{H}^{\frac{1}{2}} + \left(\text{tr}\left(\mathbf{H}^{\frac{1}{2}}\mathbf{\Psi}\mathbf{H}^{\frac{1}{2}}\right) + \sigma^2\right) \cdot \mathbf{I}_d\right)\right).$$

Since

$$\left(\text{tr}\left(\mathbf{H}^{\frac{1}{2}}\mathbf{\Psi}\mathbf{H}^{\frac{1}{2}}\right) + \sigma^2\right)\cdot\mathbf{I}_d \preceq \mathbf{H}^{\frac{1}{2}}\mathbf{\Psi}\mathbf{H}^{\frac{1}{2}} + \left(\text{tr}\left(\mathbf{H}^{\frac{1}{2}}\mathbf{\Psi}\mathbf{H}^{\frac{1}{2}}\right) + \sigma^2\right)\cdot\mathbf{I}_d \preceq 2\left(\text{tr}\left(\mathbf{H}^{\frac{1}{2}}\mathbf{\Psi}\mathbf{H}^{\frac{1}{2}}\right) + \sigma^2\right)\cdot\mathbf{I}_d,$$

we have

$$\inf_{f \in \mathcal{F}_{\text{GD-}\boldsymbol{\beta}}} \mathcal{R}(f) - \sigma^2 \simeq \frac{\text{tr}\left(\mathbf{H}^{\frac{1}{2}}\mathbf{\Psi}\mathbf{H}^{\frac{1}{2}}\right) + \sigma^2}{M}\text{tr}\left(\mathbf{\Omega}^{-1}\mathbf{H}^{\frac{1}{2}}\mathbf{\Psi}\mathbf{H}^{\frac{1}{2}}\right)$$

$$= \bar{\phi} \cdot \sum_{i=1}^{d} \frac{\phi_i}{\frac{M+1}{M}\phi_i + \bar{\phi}}$$

$$\simeq \sum_{i=1}^{d} \min\left\{\phi_i, \bar{\phi}\right\}.$$

Therefore, we conclude.   $\square$

# F Proof of Theorem 6.1

## F.1 Proof of first equation

*Proof.* From the classical bias-variance decomposition, we know

$$\mathcal{L}\left(g; \mathbf{X}\right) := \mathbb{E}\left[\left(g(\mathbf{X}, \mathbf{y}, \mathbf{x}) - y\right)^2 \mid \mathbf{X}\right]$$

$$= \mathbb{E}\left[\left(g(\mathbf{X}, \mathbf{y}, \mathbf{x}) - \mathbb{E}\left[y \mid \mathbf{X}, \mathbf{y}, \mathbf{x}\right]\right)^2 \mid \mathbf{X}\right] + \mathbb{E}\left[\left(\mathbb{E}\left[y \mid \mathbf{X}, \mathbf{y}, \mathbf{x}\right] - y\right)^2 \mid \mathbf{X}\right]$$

since the cross term vanishes. The second term does not depend on $g$ and hence, the Bayesian optimal estimator is given by the posterior mean, i.e.,

$$\widehat{y}_{\mathsf{Bayes}} = \mathbb{E}\left[y \mid \mathbf{X}, \mathbf{y}, \mathbf{x}\right] = \left\langle \mathbb{E}\left[\widetilde{\boldsymbol{\beta}} \mid \mathbf{X}, \mathbf{y}, \mathbf{x}\right], \mathbf{x}\right\rangle.$$

Since $\widetilde{\boldsymbol{\beta}} \sim \mathcal{N}\left(\boldsymbol{\beta}^*, \boldsymbol{\Psi}\right)$, we have there exists a random vector $\widetilde{\boldsymbol{\theta}} \sim \mathcal{N}\left(\mathbf{0}_d, \mathbf{I}_d\right)$ such that $\widetilde{\boldsymbol{\beta}} = \boldsymbol{\beta}^* + \boldsymbol{\Psi}^{\frac{1}{2}}\widetilde{\boldsymbol{\theta}}$ almost surely. Therefore, one has

$$\widehat{y}_{\mathsf{Bayes}} = \langle \boldsymbol{\beta}^*, \mathbf{x}\rangle + \left\langle \mathbb{E}\left[\widetilde{\boldsymbol{\theta}} \mid \mathbf{X}, \mathbf{y}, \mathbf{x}\right], \boldsymbol{\Psi}^{\frac{1}{2}}\mathbf{x}\right\rangle.$$

To compute $\mathbb{E}\left[\widetilde{\boldsymbol{\theta}} \mid \mathbf{X}, \mathbf{y}, \mathbf{x}\right]$, it suffices to solve the posterior distribution of $\boldsymbol{\beta}$ given $\mathbf{X}, \mathbf{y}$. From $\widetilde{\boldsymbol{\theta}} \sim \mathcal{N}\left(\mathbf{0}_d, \mathbf{I}_d\right)$, we have

$$\mathbb{P}\left(\widetilde{\boldsymbol{\theta}} \mid \mathbf{X}, \mathbf{y}, \mathbf{x}\right) \propto \mathbb{P}\left(\widetilde{\boldsymbol{\theta}}\right) \mathbb{P}\left(\mathbf{y} \mid \mathbf{X}, \widetilde{\boldsymbol{\theta}}\right) \propto \exp\left(-\frac{\left\|\mathbf{y} - \mathbf{X}\left(\boldsymbol{\beta}^* + \boldsymbol{\Psi}^{\frac{1}{2}}\widetilde{\boldsymbol{\theta}}\right)\right\|_2^2}{2\sigma^2} - \frac{1}{2}\widetilde{\boldsymbol{\theta}}^\top \cdot \widetilde{\boldsymbol{\theta}}\right)$$

$$\propto \exp\left(-\frac{1}{2\sigma^2}\left[\widetilde{\boldsymbol{\theta}}^\top \left(\boldsymbol{\Psi}^{\frac{1}{2}}\mathbf{X}^\top \mathbf{X}\boldsymbol{\Psi}^{\frac{1}{2}} + \sigma^2\mathbf{I}_d\right)\widetilde{\boldsymbol{\theta}} - 2\left(\mathbf{y} - \mathbf{X}\boldsymbol{\beta}^*\right)^\top \mathbf{X}\boldsymbol{\Psi}^{\frac{1}{2}}\widetilde{\boldsymbol{\theta}}\right]\right).$$

Note that, the function above matches the probability density function of a multivariate Gaussian distrbution. Therefore, the posterior mean is given by

$$\mathbb{E}\left[\boldsymbol{\theta} \mid \mathbf{X}, \mathbf{y}, \mathbf{x}\right] = \left(\boldsymbol{\Psi}^{\frac{1}{2}}\mathbf{X}^\top \mathbf{X}\boldsymbol{\Psi}^{\frac{1}{2}} + \sigma^2\mathbf{I}_d\right)^{-1}\boldsymbol{\Psi}^{\frac{1}{2}}\mathbf{X}^\top \left(\mathbf{y} - \mathbf{X}\boldsymbol{\beta}^*\right).$$

Therefore, we conclude

$$\widehat{y}_{\mathsf{Bayes}} = \mathbf{x}^\top \boldsymbol{\Psi}^{\frac{1}{2}}\left(\boldsymbol{\Psi}^{\frac{1}{2}}\mathbf{X}^\top \mathbf{X}\boldsymbol{\Psi}^{\frac{1}{2}} + \sigma^2\mathbf{I}_d\right)^{-1}\boldsymbol{\Psi}^{\frac{1}{2}}\mathbf{X}^\top \left(\mathbf{y} - \mathbf{X}\boldsymbol{\beta}^*\right) + \mathbf{x}^\top \boldsymbol{\beta}^*. \qquad \text{(F.1)}$$

$\square$

## F.2 Proof of second equation

*Proof.* Let's first do a variable transformation. We denote

$$\widetilde{\mathbf{X}} = \mathbf{X}\boldsymbol{\Psi}^{\frac{1}{2}}, \quad \widetilde{\mathbf{x}} = \boldsymbol{\Psi}^{\frac{1}{2}}\mathbf{x}, \quad \widetilde{\mathbf{y}} = \mathbf{y} - \mathbf{X}\boldsymbol{\beta}^*, \quad \boldsymbol{\Lambda} = \boldsymbol{\Psi}^{\frac{1}{2}}\mathbf{H}\boldsymbol{\Psi}^{\frac{1}{2}}. \qquad \text{(F.2)}$$

Then, we know $\widetilde{\mathbf{X}}[i], \widetilde{\mathbf{x}} \overset{\text{i.i.d.}}{\sim} \mathcal{N}\left(\mathbf{0}_d, \boldsymbol{\Lambda}\right)$. We can write the Bayesian optimal estimator in (F.1) as

$$\widehat{y}_{\mathsf{Bayes}} = \widetilde{\mathbf{x}}^\top \left(\widetilde{\mathbf{X}}^\top \widetilde{\mathbf{X}} + \sigma^2\mathbf{I}_d\right)^{-1}\widetilde{\mathbf{X}}\widetilde{\mathbf{y}} + \mathbf{x}^\top \boldsymbol{\beta}^*.$$

The true label is

$$y = \mathbf{x}^\top \widetilde{\boldsymbol{\beta}} + \varepsilon = \mathbf{x}^\top \left(\boldsymbol{\beta}^* + \boldsymbol{\Psi}^{\frac{1}{2}}\widetilde{\boldsymbol{\theta}}\right).$$

Therefore, we have

$$\mathcal{L}\left(\widehat{y}_{\mathsf{Bayes}}; \mathbf{X}\right) - \sigma^2 = \mathbb{E}\left(\widetilde{\mathbf{x}}^\top \left(\widetilde{\mathbf{X}}^\top \widetilde{\mathbf{X}} + \sigma^2\mathbf{I}_d\right)^{-1}\widetilde{\mathbf{X}}\widetilde{\mathbf{y}} + \mathbf{x}^\top \boldsymbol{\beta}^* - \mathbf{x}^\top \left(\boldsymbol{\beta}^* + \boldsymbol{\Psi}^{\frac{1}{2}}\widetilde{\boldsymbol{\theta}}\right)\right)^2$$

$$= \mathbb{E}\left(\widetilde{\mathbf{x}}^{\top}\left[\left(\widetilde{\mathbf{X}}^{\top}\widetilde{\mathbf{X}} + \sigma^2 \mathbf{I}_d\right)^{-1}\widetilde{\mathbf{X}}\widetilde{\mathbf{y}} - \widehat{\boldsymbol{\theta}}\right]\right)^2 = \mathbb{E}\left\|\widehat{\boldsymbol{\theta}} - \widetilde{\boldsymbol{\theta}}\right\|_{\boldsymbol{\Lambda}}^2,$$

where

$$\widehat{\boldsymbol{\theta}} := \left(\widetilde{\mathbf{X}}^{\top}\widetilde{\mathbf{X}} + \sigma^2 \mathbf{I}_d\right)^{-1}\widetilde{\mathbf{X}}\widetilde{\mathbf{y}}.$$

Note that, this is equivalent to the risk of estimating the ground true linear weight $\widetilde{\boldsymbol{\theta}}$ using $\widehat{\boldsymbol{\theta}}$ under the Gaussian prior $\widetilde{\boldsymbol{\theta}} \sim \mathcal{N}(\mathbf{0}, \mathbf{I}_d)$. The estimator $\widehat{\boldsymbol{\theta}}$ is a function of transformed input-output pairs in the context $\left(\widetilde{\mathbf{X}}, \widetilde{\mathbf{y}}\right)$ and the transformed query input $\mathbf{x}$, and takes the form of standard ridge estimator with regularization coefficient being $\sigma^2$. We then revoke the standard results for the risk of a ridge estimator. Applying the Theorem 1 and 2 in [33], we know for a fixed weight vector $\widetilde{\boldsymbol{\theta}}$, with probability at least $1 - \exp\left(-\Omega(M)\right)$ we have

$$\left\|\widehat{\boldsymbol{\theta}} - \widetilde{\boldsymbol{\theta}}\right\|_{\mathbf{H}}^2 \simeq \left(\frac{\sigma^2 + \sum_{i>k^*}\phi_i}{M}\right)^2\left\|\widetilde{\boldsymbol{\theta}}\right\|_{\mathbf{H}_{0:k^*}^{-1}}^2 + \left\|\widetilde{\boldsymbol{\theta}}\right\|_{\mathbf{H}_{k^*:\infty}}^2 + \sigma^2\left(\frac{k^*}{M} + \frac{M\sum_{i>k^*}\phi_i^2}{\sigma^2 + \sum_{i>k^*}\phi_i}\right),$$

where $\phi_1 \geq \phi_2 \geq ... \geq \phi_d \geq 0$ are ordered eigenvalues of $\boldsymbol{\Lambda}$.

$$k^* := \min\left\{k : \phi_k \geq c \cdot \frac{\sigma^2 + \sum_{i>k^*}\phi_i}{M}\right\}$$

and $c > 1$ is an absolute constant. Here, $\mathbf{H}_{0:k^*}$ is SVD approximation with respect to the largest $k^*$ singular values and $\mathbf{H}_{k^*:\infty}$ is the SVD approximation in the remaining singular values. Namely, if we have the eigen-decomposition of $\mathbf{H} = \mathbf{Q} \cdot \operatorname{diag}(\phi_1, \phi_2, ..., \phi_d) \cdot \mathbf{Q}^{\top}$, where $\mathbf{Q}$ is an orthogonal matrix, then $\mathbf{H}_{0:k^*}$ and $\mathbf{H}_{k^*:\infty}$ are given by

$$\mathbf{H}_{0:k^*} = \mathbf{Q} \cdot \operatorname{diag}(\phi_1, \phi_2, ..., \phi_{k^*}, 0, 0, ..., 0) \cdot \mathbf{Q}^{\top}, \quad \mathbf{H}_{k^*:\infty} = \mathbf{H} - \mathbf{H}_{0:k^*}.$$

Taking expectation over $\widetilde{\boldsymbol{\theta}} \sim \mathcal{N}(0, \mathbf{I}_d)$, we have

$$\mathcal{L}\left(\widehat{y}_{\mathsf{Bayes}}; \mathbf{X}\right) - \sigma^2 = \mathbb{E}_{\widetilde{\boldsymbol{\theta}} \sim \mathcal{N}(0, \mathbf{I}_d)}\left\|\widehat{\boldsymbol{\theta}}_{\mathsf{Bayes}} - \widetilde{\boldsymbol{\theta}}\right\|_{\mathbf{H}}^2$$

$$= \left(\frac{\sigma^2 + \sum_{i>k^*}\phi_i}{M}\right)^2 \cdot \sum_{i\leq k^*}\frac{1}{\phi_i} + \sum_{i>k^*}\phi_i + \sigma^2\left(\frac{k^*}{M} + \frac{M\sum_{i>k^*}\phi_i^2}{\sigma^2 + \sum_{i>k^*}\phi_i}\right)$$

Now we simplify this expression. First, we define

$$\bar{\phi} := c \cdot \frac{\sigma^2 + \sum_{i>k^*}\phi_i}{M}.$$

From the definition, we see $\bar{\phi} \geq \frac{c\sigma^2}{M}$. On the other hand, from the assumption that $\operatorname{tr}(\mathbf{H}) = \operatorname{tr}\left(\boldsymbol{\Psi}^{\frac{1}{2}}\mathbf{H}\boldsymbol{\Psi}^{\frac{1}{2}}\right) = \sum_{i=1}^d \phi_i \lesssim \sigma^2$, we know that $\bar{\phi} \lesssim \frac{c\sigma^2}{M}$. Combining two parts, we get

$$\bar{\phi} \simeq \frac{\sigma^2}{M}. \tag{F.3}$$

Therefore, we have

$$\mathcal{L}\left(\widehat{y}_{\mathsf{Bayes}}; \mathbf{X}\right) - \sigma^2 \simeq \bar{\phi}^2 \cdot \sum_{i\leq k^*}\frac{1}{\phi_i} + \sum_{i>k^*}\phi_i + \frac{\sigma^2}{M}\cdot\left(k^* + \frac{\sum_{i>k^*}\phi_i^2}{\bar{\phi}^2}\right)$$

$$\simeq \sum_i \min\left\{\frac{\bar{\phi}^2}{\phi_i}, \phi_i\right\} + \bar{\phi} \cdot \sum_i \min\left\{1, \frac{\phi_i^2}{\bar{\phi}^2}\right\} \qquad \text{(from (F.3))}$$

$$\simeq \sum_i\left(\min\left\{\frac{\bar{\phi}^2}{\phi_i}, \phi_i\right\} + \min\left\{\bar{\phi}, \frac{\phi_i^2}{\bar{\phi}}\right\}\right)$$

$$\simeq \sum_i \min\left\{\phi_i, \bar{\phi}\right\}.$$

This finishes the proof. $\qquad\qquad\qquad\qquad\qquad\qquad\qquad\qquad\qquad\qquad\qquad\qquad\qquad\quad\square$

# G  Proof of Theorem 6.3

## G.1  Training dynamics

Now we consider doing gradient flow on the risk function, or equivalently, on the excess risk function which differs by only a constant:

$$\frac{\mathrm{d}\boldsymbol{\beta}}{\mathrm{d}t} = -\frac{1}{2}\frac{\partial}{\partial\boldsymbol{\beta}}\left[\mathcal{R}(\boldsymbol{\beta},\boldsymbol{\Gamma}) - \min\mathcal{R}(\cdot,\cdot)\right]; \tag{G.1}$$

$$\frac{\mathrm{d}\boldsymbol{\Gamma}}{\mathrm{d}t} = -\frac{1}{2}\frac{\partial}{\partial\boldsymbol{\Gamma}}\left[\mathcal{R}(\boldsymbol{\beta},\boldsymbol{\Gamma}) - \min\mathcal{R}(\cdot,\cdot)\right]. \tag{G.2}$$

First, we have the following corollary of Theorem 5.2, which computes the excess risk $\mathcal{R}(\boldsymbol{\beta},\boldsymbol{\Gamma}) - \min\mathcal{R}(\cdot,\cdot)$.

**Corollary G.1** (Excess Risk). *We fix $M$ as the context length. Consider the ICL risk in (3.5), assume the data is generated following Assumption 3.1. Then, we have*

$$\mathcal{R}(\boldsymbol{\beta},\boldsymbol{\Gamma}) - \min\mathcal{R}(\cdot,\cdot)$$

$$= (\boldsymbol{\beta}-\boldsymbol{\beta}^*)^\top\left[(\mathbf{I}_d - \boldsymbol{\Gamma}\mathbf{H})^\top\mathbf{H}(\mathbf{I}_d - \boldsymbol{\Gamma}\mathbf{H}) + \frac{\mathrm{tr}\left(\mathbf{H}\boldsymbol{\Gamma}^\top\mathbf{H}\boldsymbol{\Gamma}\right)}{M}\mathbf{H} + \frac{1}{M}\mathbf{H}\boldsymbol{\Gamma}^\top\mathbf{H}\boldsymbol{\Gamma}\mathbf{H}\right](\boldsymbol{\beta}-\boldsymbol{\beta}^*)$$

$$+ \mathrm{tr}\left[\left(\mathbf{H}^{\frac{1}{2}}\boldsymbol{\Gamma}\mathbf{H}^{\frac{1}{2}} - \mathbf{H}^{\frac{1}{2}}\boldsymbol{\Psi}\mathbf{H}^{\frac{1}{2}}\boldsymbol{\Omega}^{-1}\right)\boldsymbol{\Omega}\left(\mathbf{H}^{\frac{1}{2}}\boldsymbol{\Gamma}\mathbf{H}^{\frac{1}{2}} - \mathbf{H}^{\frac{1}{2}}\boldsymbol{\Psi}\mathbf{H}^{\frac{1}{2}}\boldsymbol{\Omega}^{-1}\right)^\top\right]. \tag{G.3}$$

*Proof.* This is obtained directly from equation (C.5) and (C.7). $\qquad\square$

Now, we write out the differential equations explicitly in the following lemma.

**Lemma G.2** (Dynamical system). *The dynamical system of gradient flow described in (G.1) and (G.2) is*

$$\frac{\mathrm{d}\boldsymbol{\beta}}{\mathrm{d}t} = -\left[(\mathbf{I}_d - \boldsymbol{\Gamma}\mathbf{H})^\top\mathbf{H}(\mathbf{I}_d - \boldsymbol{\Gamma}\mathbf{H}) + \frac{\mathrm{tr}\left(\mathbf{H}\boldsymbol{\Gamma}^\top\mathbf{H}\boldsymbol{\Gamma}\right)}{M}\mathbf{H} + \frac{1}{M}\mathbf{H}\boldsymbol{\Gamma}^\top\mathbf{H}\boldsymbol{\Gamma}\mathbf{H}\right](\boldsymbol{\beta}-\boldsymbol{\beta}^*) \tag{G.4}$$

$$\frac{\mathrm{d}\boldsymbol{\Gamma}}{\mathrm{d}t} = -\left(\mathbf{H}\boldsymbol{\Gamma}\mathbf{H}^{\frac{1}{2}}\boldsymbol{\Omega}\mathbf{H}^{\frac{1}{2}} - \mathbf{H}\boldsymbol{\Psi}\mathbf{H}\right) - \frac{M+1}{M}\mathbf{H}\boldsymbol{\Gamma}\mathbf{H}(\boldsymbol{\beta}-\boldsymbol{\beta}^*)(\boldsymbol{\beta}-\boldsymbol{\beta}^*)^\top\mathbf{H}$$

$$- \frac{1}{M}(\boldsymbol{\beta}-\boldsymbol{\beta}^*)^\top\mathbf{H}(\boldsymbol{\beta}-\boldsymbol{\beta}^*)\cdot\mathbf{H}\boldsymbol{\Gamma}\mathbf{H} + \mathbf{H}(\boldsymbol{\beta}-\boldsymbol{\beta}^*)(\boldsymbol{\beta}-\boldsymbol{\beta}^*)^\top\mathbf{H}. \tag{G.5}$$

*Proof.* This can be obtained by directly calculating the derivatives over the excess risk (G.3). To write out the dynamics of $\boldsymbol{\beta}$, it suffices to notice that $\boldsymbol{\beta}$ only attends the first term of the RHS of (G.3), which is a standard quadratic function. For the derivatives of $\boldsymbol{\Gamma}$, we first have

$$\frac{1}{2}\frac{\partial}{\partial\boldsymbol{\Gamma}}\mathrm{tr}\left[\left(\mathbf{H}^{\frac{1}{2}}\boldsymbol{\Gamma}\mathbf{H}^{\frac{1}{2}} - \mathbf{H}^{\frac{1}{2}}\boldsymbol{\Psi}\mathbf{H}^{\frac{1}{2}}\boldsymbol{\Omega}^{-1}\right)\boldsymbol{\Omega}\left(\mathbf{H}^{\frac{1}{2}}\boldsymbol{\Gamma}\mathbf{H}^{\frac{1}{2}} - \mathbf{H}^{\frac{1}{2}}\boldsymbol{\Psi}\mathbf{H}^{\frac{1}{2}}\boldsymbol{\Omega}^{-1}\right)^\top\right]$$

$$= \mathbf{H}^{\frac{1}{2}}\left(\mathbf{H}^{\frac{1}{2}}\boldsymbol{\Gamma}\mathbf{H}^{\frac{1}{2}} - \mathbf{H}^{\frac{1}{2}}\boldsymbol{\Psi}\mathbf{H}^{\frac{1}{2}}\boldsymbol{\Omega}^{-1}\right)\boldsymbol{\Omega}^{\frac{1}{2}}\cdot\boldsymbol{\Omega}^{\frac{1}{2}}\mathbf{H}^{\frac{1}{2}} \qquad\text{(the sixth equation in Lemma H.3)}$$

$$= \mathbf{H}\boldsymbol{\Gamma}\mathbf{H}^{\frac{1}{2}}\boldsymbol{\Omega}\mathbf{H}^{\frac{1}{2}} - \mathbf{H}\boldsymbol{\Psi}\mathbf{H}.$$

Now it suffices to compute

$$\frac{\partial}{\partial\boldsymbol{\Gamma}}\frac{1}{2}(\boldsymbol{\beta}-\boldsymbol{\beta}^*)^\top\left[(\mathbf{I}_d - \boldsymbol{\Gamma}\mathbf{H})^\top\mathbf{H}(\mathbf{I}_d - \boldsymbol{\Gamma}\mathbf{H}) + \frac{\mathrm{tr}\left(\mathbf{H}\boldsymbol{\Gamma}^\top\mathbf{H}\boldsymbol{\Gamma}\right)}{M}\mathbf{H} + \frac{1}{M}\mathbf{H}\boldsymbol{\Gamma}^\top\mathbf{H}\boldsymbol{\Gamma}\mathbf{H}\right](\boldsymbol{\beta}-\boldsymbol{\beta}^*).$$

Let's compute the partial derivativess separately. From Lemma H.3, we have

$$\frac{\partial}{\partial \mathbf{\Gamma}} \frac{1}{2} (\boldsymbol{\beta} - \boldsymbol{\beta}^*)^\top \left(-\mathbf{H}\mathbf{\Gamma}^\top\mathbf{H}\right)(\boldsymbol{\beta} - \boldsymbol{\beta}^*) = \frac{\partial}{\partial \mathbf{\Gamma}} \frac{1}{2} (\boldsymbol{\beta} - \boldsymbol{\beta}^*)^\top \left(-\mathbf{H}\mathbf{\Gamma}\mathbf{H}\right)(\boldsymbol{\beta} - \boldsymbol{\beta}^*)$$

$$= -\frac{1}{2}\mathbf{H}(\boldsymbol{\beta} - \boldsymbol{\beta}^*)(\boldsymbol{\beta} - \boldsymbol{\beta}^*)^\top \mathbf{H};$$

$$\frac{\partial}{\partial \mathbf{\Gamma}} \frac{1}{2} \left(\frac{M+1}{M}(\boldsymbol{\beta} - \boldsymbol{\beta}^*)^\top \mathbf{H}\mathbf{\Gamma}^\top\mathbf{H}\mathbf{\Gamma}\mathbf{H}(\boldsymbol{\beta} - \boldsymbol{\beta}^*)\right) = \frac{M+1}{M}\mathbf{H}\mathbf{\Gamma} \cdot \mathbf{H}(\boldsymbol{\beta} - \boldsymbol{\beta}^*)(\boldsymbol{\beta} - \boldsymbol{\beta}^*)^\top \mathbf{H};$$

$$\frac{\partial}{\partial \mathbf{\Gamma}} \frac{1}{2} \left(\frac{\operatorname{tr}\left\{\mathbf{H}\mathbf{\Gamma}^\top\mathbf{H}\mathbf{\Gamma}\right\}}{M} \cdot (\boldsymbol{\beta} - \boldsymbol{\beta}^*)^\top \mathbf{H}(\boldsymbol{\beta} - \boldsymbol{\beta}^*)\right) = \frac{1}{M}(\boldsymbol{\beta} - \boldsymbol{\beta}^*)^\top \mathbf{H}(\boldsymbol{\beta} - \boldsymbol{\beta}^*) \cdot \mathbf{H}\mathbf{\Gamma}\mathbf{H}.$$

Summing over the three equations above and applying the definition of gradient flow, we conclude the dynamics of $\mathbf{\Gamma}$ in (G.5). $\qquad\square$

## G.2 Proof of the global convergence

Let's now prove Theorem 6.3. Since the first part of Theorem 6.3 is directly implied by the second part, we only deal with the case with general $\mathbf{H}$. In this section, we denote $\lambda_{-1} > 0$ as the minimal non-zero eigenvalue of $\mathbf{H}$. As in the main text, we define

$$\mathcal{H} := \operatorname{Im}(\mathbf{H})$$

and $\mathcal{H}^\perp$ as its orthogonal complement. First, let's prove the convergence of $\boldsymbol{\beta}$.

**Lemma G.3** (Convergence of $\boldsymbol{\beta}$). *Under the dynamical system* (G.4) *and* (G.5)*, one has*

$$\|\mathcal{P}_\mathcal{H}(\boldsymbol{\beta}(t)) - \mathcal{P}_\mathcal{H}(\boldsymbol{\beta}^*)\|_2^2 \leq \exp\left(\frac{-2\lambda_{-1}t}{M+1}\right)\|\mathcal{P}_\mathcal{H}(\boldsymbol{\beta}(0)) - \mathcal{P}_\mathcal{H}(\boldsymbol{\beta}^*)\|_2^2, \tag{G.6}$$

*which implies*

$$\left\|\mathbf{H}^{\frac{1}{2}}(\boldsymbol{\beta}(t) - \boldsymbol{\beta}^*)\right\|_2^2 \leq \lambda_1 \exp\left(\frac{-2\lambda_{-1}t}{M+1}\right)\|\boldsymbol{\beta}(0) - \boldsymbol{\beta}^*\|_2^2, \tag{G.7}$$

*and*

$$\mathcal{P}_\mathcal{H}(\boldsymbol{\beta}(t)) \to \mathcal{P}_\mathcal{H}(\boldsymbol{\beta}^*)$$

*when* $t \to \infty$, *from arbitrary initialization* $\boldsymbol{\beta}(0)$ *and* $\mathbf{\Gamma}(0)$. *Moreover, one has for any* $t > 0$, *it holds that*

$$\mathcal{P}_{\mathcal{H}^\perp}(\boldsymbol{\beta}(t)) = \mathcal{P}_{\mathcal{H}^\perp}(\boldsymbol{\beta}(0)) \tag{G.8}$$

*Proof.* We first consider the orthogonal projection operator $\mathcal{P}$. From Lemma H.5 and equation (G.4), we know

$$\frac{\mathrm{d}\mathcal{P}_\mathcal{H}(\boldsymbol{\beta}(t) - \boldsymbol{\beta}^*)}{\mathrm{d}t} = -\mathbf{H}\mathbf{H}^+\mathbf{H}_\mathbf{\Gamma}(\boldsymbol{\beta} - \boldsymbol{\beta}^*),$$

where

$$\mathbf{H}_\mathbf{\Gamma} := (\mathbf{I}_d - \mathbf{\Gamma}\mathbf{H})^\top \mathbf{H}(\mathbf{I}_d - \mathbf{\Gamma}\mathbf{H}) + \frac{\operatorname{tr}\left(\mathbf{H}\mathbf{\Gamma}^\top\mathbf{H}\mathbf{\Gamma}\right)}{M}\mathbf{H} + \frac{1}{M}\mathbf{H}\mathbf{\Gamma}^\top\mathbf{H}\mathbf{\Gamma}\mathbf{H}.$$

From the property of pseudo-inverse, we know

$$\frac{\mathrm{d}\mathcal{P}_\mathcal{H}(\boldsymbol{\beta}(t) - \boldsymbol{\beta}^*)}{\mathrm{d}t} = -\mathbf{H}_\mathbf{\Gamma}(\boldsymbol{\beta} - \boldsymbol{\beta}^*) = -\mathbf{H}_\mathbf{\Gamma}\mathbf{H}^+\mathbf{H}(\boldsymbol{\beta} - \boldsymbol{\beta}^*) = -\mathbf{H}_\mathbf{\Gamma}\mathcal{P}_\mathcal{H}(\boldsymbol{\beta} - \boldsymbol{\beta}^*).$$

Therefore, we have

$$\frac{\mathrm{d}}{\mathrm{d}t}\left[\frac{1}{2}\|\mathcal{P}_\mathcal{H}(\boldsymbol{\beta}(t) - \boldsymbol{\beta}^*)\|_2^2\right] = -\mathcal{P}_\mathcal{H}(\boldsymbol{\beta} - \boldsymbol{\beta}^*)^\top \cdot \mathbf{H}_\mathbf{\Gamma} \cdot \mathcal{P}_\mathcal{H}(\boldsymbol{\beta} - \boldsymbol{\beta}^*).$$

Notice that

$$\mathbf{H}_\mathbf{\Gamma} \succeq \left(\sqrt{\frac{M}{M+1}}\mathbf{I} - \sqrt{\frac{M+1}{M}}\mathbf{\Gamma}\mathbf{H}\right)^\top \mathbf{H}\left(\sqrt{\frac{M}{M+1}}\mathbf{I} - \sqrt{\frac{M+1}{M}}\mathbf{\Gamma}\mathbf{H}\right) + \frac{1}{M+1}\mathbf{H} \succeq \frac{1}{M+1}\mathbf{H}.$$

This implies

$$\frac{\mathrm{d}}{\mathrm{d}t}\left[\frac{1}{2}\left\|\mathcal{P}_{\mathcal{H}}(\boldsymbol{\beta}(t)-\boldsymbol{\beta}^*)\right\|_2^2\right] \leq -\frac{1}{M+1}\mathcal{P}_{\mathcal{H}}\left(\boldsymbol{\beta}-\boldsymbol{\beta}^*\right)^\top \cdot \mathbf{H} \cdot \mathcal{P}_{\mathcal{H}}\left(\boldsymbol{\beta}-\boldsymbol{\beta}^*\right)$$

$$\leq -\frac{\lambda_{-1}}{M+1}\left\|\mathcal{P}_{\mathcal{H}}\left(\boldsymbol{\beta}-\boldsymbol{\beta}^*\right)\right\|_2^2,$$

where the last line comes from Lemma H.5 and $\lambda_{-1}$ is the minimal non-zero eigenvalue of $\mathbf{H}$. Via standard integration method in ODE, we know this suggests

$$\left\|\mathcal{P}_{\mathcal{H}}\left(\boldsymbol{\beta}(t)-\boldsymbol{\beta}^*\right)\right\|_2^2 \leq \exp\left(\frac{-2\lambda_{-1}t}{M+1}\right)\left\|\mathcal{P}_{\mathcal{H}}\left(\boldsymbol{\beta}(0)-\boldsymbol{\beta}^*\right)\right\|_2^2 \to 0 \quad \text{when } t \to \infty.$$

This indicates that $\mathcal{P}_{\mathcal{H}}\left(\boldsymbol{\beta}(t)\right) \to \mathcal{P}_{\mathcal{H}}\left(\boldsymbol{\beta}^*\right)$ when $t \to \infty$. Finally, we have

$$\frac{\mathrm{d}\mathcal{P}_{\mathcal{H}^\perp}\left(\boldsymbol{\beta}(t)-\boldsymbol{\beta}^*\right)}{\mathrm{d}t} = -\left(\mathbf{I}_d - \mathbf{H}\mathbf{H}^+\right)\mathbf{H}_{\boldsymbol{\Gamma}}\left(\boldsymbol{\beta}-\boldsymbol{\beta}^*\right) = \mathbf{0}_d,$$

which implies $\mathcal{P}_{\mathcal{H}^\perp}\left(\boldsymbol{\beta}(t)-\boldsymbol{\beta}^*\right) = \mathcal{P}_{\mathcal{H}^\perp}\left(\boldsymbol{\beta}(0)-\boldsymbol{\beta}^*\right)$ and hence, $\mathcal{P}_{\mathcal{H}^\perp}\left(\boldsymbol{\beta}(t)\right) = \mathcal{P}_{\mathcal{H}^\perp}\left(\boldsymbol{\beta}(0)\right)$ for any $t > 0$. $\qquad\square$

Next, let's consider the convergence of $\boldsymbol{\Gamma}$. As defined in the main text, we have

$$\mathcal{Z} := \mathsf{Im}\left(\mathbf{H} \otimes \mathbf{H}\right)$$

and $\mathcal{Z}^\perp$ is its orthogonal complement. We have the following lemma.

**Lemma G.4.** *Under the dynamical system* (G.4) *and* (G.5)*, one has*

$$\frac{\mathrm{d}}{\mathrm{d}t}\mathcal{P}_{\mathcal{Z}^\perp}\left(\boldsymbol{\Gamma}-\boldsymbol{\Gamma}^*\right) = \mathbf{0}_{d\times d}, \tag{G.9}$$

$$\begin{aligned}
\frac{\mathrm{d}}{\mathrm{d}t}\mathcal{P}_{\mathcal{Z}}\left(\boldsymbol{\Gamma}-\boldsymbol{\Gamma}^*\right) = &-\mathbf{H}\cdot\mathcal{P}_{\mathcal{Z}}\left(\boldsymbol{\Gamma}-\boldsymbol{\Gamma}^*\right)\cdot\mathbf{H}^{\frac{1}{2}}\boldsymbol{\Omega}\mathbf{H}^{\frac{1}{2}} - \frac{M+1}{M}\mathbf{H}\cdot\mathcal{P}_{\mathcal{Z}}\left(\boldsymbol{\Gamma}-\boldsymbol{\Gamma}^*\right)\cdot\mathbf{H}\left(\boldsymbol{\beta}-\boldsymbol{\beta}^*\right)\left(\boldsymbol{\beta}-\boldsymbol{\beta}^*\right)^\top\mathbf{H} \\
&-\frac{1}{M}\left(\boldsymbol{\beta}-\boldsymbol{\beta}^*\right)^\top\mathbf{H}\left(\boldsymbol{\beta}-\boldsymbol{\beta}^*\right)\cdot\mathbf{H}\cdot\mathcal{P}_{\mathcal{Z}}\left(\boldsymbol{\Gamma}-\boldsymbol{\Gamma}^*\right)\cdot\mathbf{H} \\
&-\frac{1}{M}\left(\boldsymbol{\beta}-\boldsymbol{\beta}^*\right)^\top\mathbf{H}\left(\boldsymbol{\beta}-\boldsymbol{\beta}^*\right)\cdot\mathbf{H}\boldsymbol{\Gamma}^*\mathbf{H} - \frac{1}{M}\mathbf{H}\boldsymbol{\Gamma}^*\mathbf{H}\left(\boldsymbol{\beta}-\boldsymbol{\beta}^*\right)\left(\boldsymbol{\beta}-\boldsymbol{\beta}^*\right)^\top\mathbf{H} \\
&+\frac{1}{M}\mathbf{H}^{\frac{1}{2}}\boldsymbol{\Omega}^{-1}\left(\mathbf{H}^{\frac{1}{2}}\boldsymbol{\Psi}\mathbf{H}^{\frac{1}{2}} + \left(\mathrm{tr}\left(\mathbf{H}^{\frac{1}{2}}\boldsymbol{\Psi}\mathbf{H}^{\frac{1}{2}}\right) + \sigma^2\right)\mathbf{I}_d\right)\mathbf{H}^{\frac{1}{2}}\left(\boldsymbol{\beta}-\boldsymbol{\beta}^*\right)\left(\boldsymbol{\beta}-\boldsymbol{\beta}^*\right)^\top\mathbf{H}.
\end{aligned} \tag{G.10}$$

*Proof.* The first ODE is trivial since the RHS of (G.5) always lies in $\mathcal{Z} := \mathsf{Im}\left(\mathbf{H} \otimes \mathbf{H}\right)$, so its projection on $\mathcal{Z}^\perp$ always vanishes. To prove the second equation, we can rewrite (G.5) as

$$\begin{aligned}
\frac{\mathrm{d}\boldsymbol{\Gamma}}{\mathrm{d}t} = &-\left(\mathbf{H}\left(\boldsymbol{\Gamma}-\boldsymbol{\Gamma}^*\right)\mathbf{H}^{\frac{1}{2}}\boldsymbol{\Omega}\mathbf{H}^{\frac{1}{2}}\underbrace{-\mathbf{H}\boldsymbol{\Psi}\mathbf{H} + \mathbf{H}\boldsymbol{\Gamma}^*\mathbf{H}^{\frac{1}{2}}\boldsymbol{\Omega}\mathbf{H}^{\frac{1}{2}}}_{A}\right) \\
&-\frac{M+1}{M}\mathbf{H}\left(\boldsymbol{\Gamma}-\boldsymbol{\Gamma}^*\right)\mathbf{H}\left(\boldsymbol{\beta}-\boldsymbol{\beta}^*\right)\left(\boldsymbol{\beta}-\boldsymbol{\beta}^*\right)^\top\mathbf{H} \\
&-\frac{1}{M}\left(\boldsymbol{\beta}-\boldsymbol{\beta}^*\right)^\top\mathbf{H}\left(\boldsymbol{\beta}-\boldsymbol{\beta}^*\right)\cdot\mathbf{H}\left(\boldsymbol{\Gamma}-\boldsymbol{\Gamma}^*\right)\mathbf{H} + \underbrace{\mathbf{H}\left(\boldsymbol{\beta}-\boldsymbol{\beta}^*\right)\left(\boldsymbol{\beta}-\boldsymbol{\beta}^*\right)^\top\mathbf{H}}_{B} \\
&-\frac{1}{M}\left(\boldsymbol{\beta}-\boldsymbol{\beta}^*\right)^\top\mathbf{H}\left(\boldsymbol{\beta}-\boldsymbol{\beta}^*\right)\cdot\mathbf{H}\boldsymbol{\Gamma}^*\mathbf{H}\underbrace{-\frac{M+1}{M}\mathbf{H}\boldsymbol{\Gamma}^*\mathbf{H}\left(\boldsymbol{\beta}-\boldsymbol{\beta}^*\right)\left(\boldsymbol{\beta}-\boldsymbol{\beta}^*\right)^\top\mathbf{H}}_{C}.
\end{aligned}$$

For term A, recalling $\boldsymbol{\Gamma}^* = \boldsymbol{\Psi}\mathbf{H}^{\frac{1}{2}}\boldsymbol{\Omega}^{-1}\mathbf{H}^{-\frac{1}{2}}$, we have

$$\begin{aligned}
\mathbf{H}\boldsymbol{\Gamma}^*\mathbf{H}^{\frac{1}{2}}\boldsymbol{\Omega}\mathbf{H}^{\frac{1}{2}} &= \mathbf{H}\boldsymbol{\Psi}\mathbf{H}^{\frac{1}{2}}\boldsymbol{\Omega}^{-1}\mathbf{H}^{-\frac{1}{2}}\mathbf{H}^{\frac{1}{2}}\boldsymbol{\Omega}\mathbf{H}^{\frac{1}{2}} \\
&= \mathbf{H}^{\frac{1}{2}}\boldsymbol{\Omega}^{-1}\mathbf{H}^{\frac{1}{2}}\boldsymbol{\Psi}\mathbf{H}^{\frac{1}{2}}\mathbf{H}^{-\frac{1}{2}}\mathbf{H}^{\frac{1}{2}}\boldsymbol{\Omega}\mathbf{H}^{\frac{1}{2}} \qquad (\boldsymbol{\Omega} \text{ and } \mathbf{H}^{\frac{1}{2}}\boldsymbol{\Psi}\mathbf{H}^{\frac{1}{2}} \text{ commute.})
\end{aligned}$$

$$= \mathbf{H}^{\frac{1}{2}}\boldsymbol{\Omega}^{-1}\mathbf{H}^{\frac{1}{2}}\boldsymbol{\Psi}\mathbf{H}^{\frac{1}{2}}\boldsymbol{\Omega}\mathbf{H}^{\frac{1}{2}} \qquad\qquad (\mathbf{H}^{\frac{1}{2}}\mathbf{H}^{-\frac{1}{2}}\mathbf{H}^{\frac{1}{2}} = \mathbf{H}^{\frac{1}{2}})$$

$$= \mathbf{H}\boldsymbol{\Psi}\mathbf{H}^{\frac{1}{2}}\boldsymbol{\Omega}^{-1}\boldsymbol{\Omega}\mathbf{H}^{\frac{1}{2}} \qquad\qquad (\boldsymbol{\Omega} \text{ and } \mathbf{H}^{\frac{1}{2}}\boldsymbol{\Psi}\mathbf{H}^{\frac{1}{2}} \text{ commute.})$$

$$= \mathbf{H}\boldsymbol{\Psi}\mathbf{H}.$$

This suggests $A = 0$. For $B + C$, we have

$$B + C = -\frac{1}{M}\mathbf{H}\boldsymbol{\Gamma}^*\mathbf{H}\left(\boldsymbol{\beta} - \boldsymbol{\beta}^*\right)\left(\boldsymbol{\beta} - \boldsymbol{\beta}^*\right)^\top\mathbf{H} + \left(\mathbf{I}_d - \mathbf{H}\boldsymbol{\Gamma}^*\right)\mathbf{H}\left(\boldsymbol{\beta} - \boldsymbol{\beta}^*\right)\left(\boldsymbol{\beta} - \boldsymbol{\beta}^*\right)^\top\mathbf{H}.$$

We can compute $\left(\mathbf{I}_d - \mathbf{H}\boldsymbol{\Gamma}^*\right)\mathbf{H}$ as

$$\begin{aligned}
\left(\mathbf{I}_d - \mathbf{H}\boldsymbol{\Gamma}^*\right)\mathbf{H} &= \mathbf{H}^{\frac{1}{2}}\left(\mathbf{H}^{\frac{1}{2}} - \mathbf{H}^{\frac{1}{2}}\boldsymbol{\Psi}\mathbf{H}^{\frac{1}{2}}\boldsymbol{\Omega}^{-1}\mathbf{H}^{-\frac{1}{2}}\mathbf{H}\right)\\
&= \mathbf{H}^{\frac{1}{2}}\left(\mathbf{H}^{\frac{1}{2}} - \boldsymbol{\Omega}^{-1}\mathbf{H}^{\frac{1}{2}}\boldsymbol{\Psi}\mathbf{H}^{\frac{1}{2}}\mathbf{H}^{-\frac{1}{2}}\mathbf{H}\right) \qquad (\boldsymbol{\Omega} \text{ and } \mathbf{H}^{\frac{1}{2}}\boldsymbol{\Psi}\mathbf{H}^{\frac{1}{2}} \text{ commute.})\\
&= \mathbf{H}^{\frac{1}{2}}\left(\mathbf{H}^{\frac{1}{2}} - \boldsymbol{\Omega}^{-1}\mathbf{H}^{\frac{1}{2}}\boldsymbol{\Psi}\mathbf{H}\right) \qquad\qquad (\mathbf{H}^{\frac{1}{2}}\mathbf{H}^{-\frac{1}{2}}\mathbf{H}^{\frac{1}{2}} = \mathbf{H}^{\frac{1}{2}})\\
&= \mathbf{H}^{\frac{1}{2}}\left(\boldsymbol{\Omega}^{-1}\boldsymbol{\Omega}\mathbf{H}^{\frac{1}{2}} - \boldsymbol{\Omega}^{-1}\mathbf{H}^{\frac{1}{2}}\boldsymbol{\Psi}\mathbf{H}\right)\\
&= \frac{1}{M}\mathbf{H}^{\frac{1}{2}}\boldsymbol{\Omega}^{-1}\left(\mathbf{H}^{\frac{1}{2}}\boldsymbol{\Psi}\mathbf{H}^{\frac{1}{2}} + \left(\text{tr}\left(\mathbf{H}^{\frac{1}{2}}\boldsymbol{\Psi}\mathbf{H}^{\frac{1}{2}}\right) + \sigma^2\right)\mathbf{I}_d\right)\mathbf{H}^{\frac{1}{2}}.
\end{aligned}$$

Therefore,

$$\begin{aligned}
B + C = &-\frac{1}{M}\mathbf{H}\boldsymbol{\Gamma}^*\mathbf{H}\left(\boldsymbol{\beta} - \boldsymbol{\beta}^*\right)\left(\boldsymbol{\beta} - \boldsymbol{\beta}^*\right)^\top\mathbf{H}\\
&+ \frac{1}{M}\mathbf{H}^{\frac{1}{2}}\boldsymbol{\Omega}^{-1}\left(\mathbf{H}^{\frac{1}{2}}\boldsymbol{\Psi}\mathbf{H}^{\frac{1}{2}} + \left(\text{tr}\left(\mathbf{H}^{\frac{1}{2}}\boldsymbol{\Psi}\mathbf{H}^{\frac{1}{2}}\right) + \sigma^2\right)\mathbf{I}_d\right)\mathbf{H}^{\frac{1}{2}}\left(\boldsymbol{\beta} - \boldsymbol{\beta}^*\right)\left(\boldsymbol{\beta} - \boldsymbol{\beta}^*\right)^\top\mathbf{H}.
\end{aligned}$$

Bridging $A = 0$ and the result of $B + C$ into the ODE, we have

$$\begin{aligned}
&\frac{\mathrm{d}\left(\boldsymbol{\Gamma} - \boldsymbol{\Gamma}^*\right)}{\mathrm{d}t}\\
&= -\mathbf{H}\left(\boldsymbol{\Gamma} - \boldsymbol{\Gamma}^*\right)\mathbf{H}^{\frac{1}{2}}\boldsymbol{\Omega}\mathbf{H}^{\frac{1}{2}} - \frac{M+1}{M}\mathbf{H}\left(\boldsymbol{\Gamma} - \boldsymbol{\Gamma}^*\right)\mathbf{H}\left(\boldsymbol{\beta} - \boldsymbol{\beta}^*\right)\left(\boldsymbol{\beta} - \boldsymbol{\beta}^*\right)^\top\mathbf{H}\\
&\quad - \frac{1}{M}\left(\boldsymbol{\beta} - \boldsymbol{\beta}^*\right)^\top\mathbf{H}\left(\boldsymbol{\beta} - \boldsymbol{\beta}^*\right)\cdot\mathbf{H}\left(\boldsymbol{\Gamma} - \boldsymbol{\Gamma}^*\right)\mathbf{H}\\
&\quad - \frac{1}{M}\left(\boldsymbol{\beta} - \boldsymbol{\beta}^*\right)^\top\mathbf{H}\left(\boldsymbol{\beta} - \boldsymbol{\beta}^*\right)\cdot\mathbf{H}\boldsymbol{\Gamma}^*\mathbf{H} - \frac{1}{M}\mathbf{H}\boldsymbol{\Gamma}^*\mathbf{H}\left(\boldsymbol{\beta} - \boldsymbol{\beta}^*\right)\left(\boldsymbol{\beta} - \boldsymbol{\beta}^*\right)^\top\mathbf{H}\\
&\quad + \frac{1}{M}\mathbf{H}^{\frac{1}{2}}\boldsymbol{\Omega}^{-1}\left(\mathbf{H}^{\frac{1}{2}}\boldsymbol{\Psi}\mathbf{H}^{\frac{1}{2}} + \left(\text{tr}\left(\mathbf{H}^{\frac{1}{2}}\boldsymbol{\Psi}\mathbf{H}^{\frac{1}{2}}\right) + \sigma^2\right)\mathbf{I}_d\right)\mathbf{H}^{\frac{1}{2}}\left(\boldsymbol{\beta} - \boldsymbol{\beta}^*\right)\left(\boldsymbol{\beta} - \boldsymbol{\beta}^*\right)^\top\mathbf{H}. \quad \text{(G.11)}
\end{aligned}$$

Note that, all terms in the RHS above lie in $\mathcal{Z} = \text{Im}\left(\mathbf{H}^{\otimes 2}\right)$, so this summation also equals $\frac{\mathrm{d}}{\mathrm{d}t}\mathcal{P}_{\mathcal{Z}}\left(\boldsymbol{\Gamma} - \boldsymbol{\Gamma}^*\right)$. Moreover, since

$$\begin{aligned}
\mathbf{H}^{\frac{1}{2}}\left(\boldsymbol{\Gamma} - \boldsymbol{\Gamma}^*\right)\mathbf{H}^{\frac{1}{2}} &= \mathbf{H}^{\frac{1}{2}}\left[\mathcal{P}_{\text{Im}\left(\mathbf{H}^{\frac{1}{2}}\otimes\mathbf{H}^{\frac{1}{2}}\right)}\left(\boldsymbol{\Gamma} - \boldsymbol{\Gamma}^*\right) + \mathcal{P}_{\text{null}\left(\mathbf{H}^{\frac{1}{2}}\otimes\mathbf{H}^{\frac{1}{2}}\right)}\left(\boldsymbol{\Gamma} - \boldsymbol{\Gamma}^*\right)\right]\mathbf{H}^{\frac{1}{2}}\\
&= \mathbf{H}^{\frac{1}{2}}\cdot\mathcal{P}_{\text{Im}\left(\mathbf{H}^{\frac{1}{2}}\otimes\mathbf{H}^{\frac{1}{2}}\right)}\left(\boldsymbol{\Gamma} - \boldsymbol{\Gamma}^*\right)\mathbf{H}^{\frac{1}{2}}. \qquad\qquad\qquad \text{(G.12)}
\end{aligned}$$

Bridging (G.12) into (G.11), we conclude. $\qquad\qquad\qquad\qquad\qquad\qquad\qquad\qquad\qquad\qquad\quad\square$

Then, let's upper bound the dynamics of $\left\|\mathcal{P}_{\mathcal{Z}}\left(\boldsymbol{\Gamma} - \boldsymbol{\Gamma}^*\right)\right\|_F^2$ in the following lemma.

**Lemma G.5** (Dynamics of Frobenius norm). *Under the dynamical system* (G.4) *and* (G.5)*, one has*

$$\frac{\mathrm{d}}{\mathrm{d}t}\left[\frac{1}{2}\left\|\mathcal{P}_{\mathcal{Z}}\left(\boldsymbol{\Gamma} - \boldsymbol{\Gamma}^*\right)\right\|_F^2\right] \leq -A_1\left\|\mathcal{P}_{\mathcal{Z}}\left(\boldsymbol{\Gamma} - \boldsymbol{\Gamma}^*\right)\right\|_F^2 + A_2\exp\left(\frac{-2\lambda_{-1}t}{M+1}\right)\left\|\mathcal{P}_{\mathcal{Z}}\left(\boldsymbol{\Gamma} - \boldsymbol{\Gamma}^*\right)\right\|_F,$$

$$\text{(G.13)}$$

*where*

$$A_1 = \lambda_{-1}^2 \lambda_{\min}(\boldsymbol{\Omega}),$$

$$A_2 = \left(2 + \frac{2}{M}\right) \lambda_1^2 \|\boldsymbol{\beta}(0) - \boldsymbol{\beta}^*\|_2^2. \tag{G.14}$$

*are two positive constant. Here, $\lambda_{-1} > 0$ is the minimal positive eigenvalue of $\mathbf{H}$, $\lambda_1$ is the maximal eigenvalue of $\mathbf{H}$, $\lambda_{\min}(\boldsymbol{\Omega})$ is the minimal eigenvalue of $\boldsymbol{\Omega}$ (defined in (A.3)) and is strictly positive.*

*Proof.* From the dynamics of $\mathcal{P}_{\mathcal{Z}}\boldsymbol{\Gamma}$ in (G.10), we know

$$\frac{\mathrm{d}}{\mathrm{d}t}\left[\frac{1}{2}\|\mathcal{P}_{\mathcal{Z}}(\boldsymbol{\Gamma} - \boldsymbol{\Gamma}^*)\|_F^2\right] = \mathrm{tr}\left(\frac{\mathrm{d}}{\mathrm{d}t}\mathcal{P}_{\mathcal{Z}}(\boldsymbol{\Gamma} - \boldsymbol{\Gamma}^*) \cdot \mathcal{P}_{\mathcal{Z}}(\boldsymbol{\Gamma} - \boldsymbol{\Gamma}^*)^\top\right)$$

$$= G_1 + G_2 + G_3, \tag{G.15}$$

where

$$G_1 = -\mathrm{tr}\left[\mathbf{H} \cdot \mathcal{P}_{\mathcal{Z}}(\boldsymbol{\Gamma} - \boldsymbol{\Gamma}^*) \cdot \mathbf{H}^{\frac{1}{2}}\boldsymbol{\Omega}\mathbf{H}^{\frac{1}{2}} \cdot \mathcal{P}_{\mathcal{Z}}(\boldsymbol{\Gamma} - \boldsymbol{\Gamma}^*)^\top\right],$$

$$G_2 = -\mathrm{tr}\left[\frac{M+1}{M}\mathbf{H} \cdot \mathcal{P}_{\mathcal{Z}}(\boldsymbol{\Gamma} - \boldsymbol{\Gamma}^*) \cdot \mathbf{H}(\boldsymbol{\beta} - \boldsymbol{\beta}^*)(\boldsymbol{\beta} - \boldsymbol{\beta}^*)^\top \mathbf{H} \cdot \mathcal{P}_{\mathcal{Z}}(\boldsymbol{\Gamma} - \boldsymbol{\Gamma}^*)^\top\right]$$

$$- \mathrm{tr}\left[\frac{1}{M}(\boldsymbol{\beta} - \boldsymbol{\beta}^*)^\top \mathbf{H}(\boldsymbol{\beta} - \boldsymbol{\beta}^*) \cdot \mathbf{H} \cdot \mathcal{P}_{\mathcal{Z}}(\boldsymbol{\Gamma} - \boldsymbol{\Gamma}^*) \cdot \mathbf{H} \cdot \mathcal{P}_{\mathcal{Z}}(\boldsymbol{\Gamma} - \boldsymbol{\Gamma}^*)^\top\right],$$

$$G_3 = -\mathrm{tr}\left[\frac{1}{M}(\boldsymbol{\beta} - \boldsymbol{\beta}^*)^\top \mathbf{H}(\boldsymbol{\beta} - \boldsymbol{\beta}^*) \cdot \mathbf{H}\boldsymbol{\Gamma}^*\mathbf{H} \cdot \mathcal{P}_{\mathcal{Z}}(\boldsymbol{\Gamma} - \boldsymbol{\Gamma}^*)^\top\right]$$

$$+ \frac{1}{M}\mathrm{tr}\left[\mathbf{H}\boldsymbol{\Gamma}^*\mathbf{H}(\boldsymbol{\beta} - \boldsymbol{\beta}^*)(\boldsymbol{\beta} - \boldsymbol{\beta}^*)^\top \mathbf{H} \cdot \mathcal{P}_{\mathcal{Z}}(\boldsymbol{\Gamma} - \boldsymbol{\Gamma}^*)^\top\right]$$

$$+ \mathrm{tr}\left[\frac{1}{M}\mathbf{H}^{\frac{1}{2}}\boldsymbol{\Omega}^{-1}\left(\mathbf{H}^{\frac{1}{2}}\boldsymbol{\Psi}\mathbf{H}^{\frac{1}{2}} + \left(\mathrm{tr}\left(\mathbf{H}^{\frac{1}{2}}\boldsymbol{\Psi}\mathbf{H}^{\frac{1}{2}}\right) + \sigma^2\right)\mathbf{I}_d\right)\mathbf{H}^{\frac{1}{2}}(\boldsymbol{\beta} - \boldsymbol{\beta}^*)(\boldsymbol{\beta} - \boldsymbol{\beta}^*)^\top \mathbf{H} \cdot \mathcal{P}_{\mathcal{Z}}(\boldsymbol{\Gamma} - \boldsymbol{\Gamma}^*)^\top\right]. \tag{G.16}$$

From Lemma H.2, we know that $G_2 \leq 0$. Then, we consider $G_1$ and $G_3$. For $G_1$, we have

$$G_1 = -\mathrm{tr}\left[\left(\mathbf{H}^{\frac{1}{2}}\mathcal{P}_{\mathcal{Z}}(\boldsymbol{\Gamma} - \boldsymbol{\Gamma}^*)^\top \mathbf{H}^{\frac{1}{2}}\right) \cdot \left(\mathbf{H}^{\frac{1}{2}}\mathcal{P}_{\mathcal{Z}}(\boldsymbol{\Gamma} - \boldsymbol{\Gamma}^*)\mathbf{H}^{\frac{1}{2}}\right) \cdot \boldsymbol{\Omega}\right]$$

$$\leq -\lambda_{-1}^2 \mathrm{tr}\left[\mathcal{P}_{\mathcal{Z}}(\boldsymbol{\Gamma} - \boldsymbol{\Gamma}^*)^\top \cdot \mathcal{P}_{\mathcal{Z}}(\boldsymbol{\Gamma} - \boldsymbol{\Gamma}^*) \cdot \boldsymbol{\Omega}\right] \quad \text{((3) in Lemma H.6 and (2) in Lemma H.2)}$$

$$\leq -\lambda_{-1}^2 \lambda_{\min}(\boldsymbol{\Omega}) \|\mathcal{P}_{\mathcal{Z}}(\boldsymbol{\Gamma} - \boldsymbol{\Gamma}^*)\|_F^2,$$

where $\lambda_{\min}(\cdot)$ denote the minimal eigenvalue. Since $\boldsymbol{\Omega}$ is positive definite, we know $\lambda_{\min}(\boldsymbol{\Omega}) > 0$. Finally, let's upper bound $G_3$. First, we have

$$- \mathrm{tr}\left[\frac{1}{M}(\boldsymbol{\beta} - \boldsymbol{\beta}^*)^\top \mathbf{H}(\boldsymbol{\beta} - \boldsymbol{\beta}^*) \cdot \mathbf{H}\boldsymbol{\Gamma}^*\mathbf{H} \cdot \mathcal{P}_{\mathcal{Z}}(\boldsymbol{\Gamma} - \boldsymbol{\Gamma}^*)^\top\right]$$

$$\leq \frac{\sqrt{d}}{M}(\boldsymbol{\beta} - \boldsymbol{\beta}^*)^\top \mathbf{H}(\boldsymbol{\beta} - \boldsymbol{\beta}^*) \cdot \left\|\mathbf{H}^{\frac{1}{2}}\boldsymbol{\Gamma}^*\mathbf{H}^{\frac{1}{2}}\right\|_2 \left\|\mathbf{H}^{\frac{1}{2}} \cdot \mathcal{P}_{\mathcal{Z}}(\boldsymbol{\Gamma} - \boldsymbol{\Gamma}^*)^\top \mathbf{H}^{\frac{1}{2}}\right\|_F \quad \text{(Lemma H.1)}$$

$$\leq \frac{\sqrt{d}\lambda_1}{M}\exp\left(\frac{-2\lambda_{-1}t}{M+1}\right)\|\boldsymbol{\beta}(0) - \boldsymbol{\beta}^*\|_2^2 \lambda_{\max}\left(\mathbf{H}^{\frac{1}{2}}\boldsymbol{\Gamma}^*\mathbf{H}^{\frac{1}{2}}\right) \cdot \|\mathbf{H}\|_F \|\mathcal{P}_{\mathcal{Z}}(\boldsymbol{\Gamma} - \boldsymbol{\Gamma}^*)\|_F$$

$$\text{(from equation (G.7))}$$

$$\leq \frac{\sqrt{d}\lambda_1^2}{M}\exp\left(\frac{-2\lambda_{-1}t}{M+1}\right)\|\boldsymbol{\beta}(0) - \boldsymbol{\beta}^*\|_2^2 \lambda_{\max}\left(\mathbf{H}^{\frac{1}{2}}\boldsymbol{\Gamma}^*\mathbf{H}^{\frac{1}{2}}\right) \cdot \|\mathcal{P}_{\mathcal{Z}}(\boldsymbol{\Gamma} - \boldsymbol{\Gamma}^*)\|_F.$$

For $\lambda_{\max}\left(\mathbf{H}^{\frac{1}{2}}\boldsymbol{\Gamma}^*\mathbf{H}^{\frac{1}{2}}\right)$, we recall $\boldsymbol{\Gamma}^* = \boldsymbol{\Psi}\mathbf{H}^{\frac{1}{2}}\boldsymbol{\Omega}^{-1}\mathbf{H}^{-\frac{1}{2}}$ and obtain

$$\mathbf{H}^{\frac{1}{2}}\boldsymbol{\Gamma}^*\mathbf{H}^{\frac{1}{2}} = \mathbf{H}^{\frac{1}{2}}\boldsymbol{\Psi}\mathbf{H}^{\frac{1}{2}}\boldsymbol{\Omega}^{-1}\mathbf{H}^{-\frac{1}{2}}\mathbf{H}^{\frac{1}{2}} = \boldsymbol{\Omega}^{-1}\mathbf{H}^{\frac{1}{2}}\boldsymbol{\Psi}\mathbf{H}^{\frac{1}{2}}\mathbf{H}^{-\frac{1}{2}}\mathbf{H}^{\frac{1}{2}} \quad (\boldsymbol{\Omega} \text{ and } \mathbf{H}^{\frac{1}{2}}\boldsymbol{\Psi}\mathbf{H}^{\frac{1}{2}} \text{ commute})$$

$$= \boldsymbol{\Omega}^{-1}\mathbf{H}^{\frac{1}{2}}\boldsymbol{\Psi}\mathbf{H}^{\frac{1}{2}}. \quad (\mathbf{H}^{\frac{1}{2}}\mathbf{H}^{-\frac{1}{2}}\mathbf{H}^{\frac{1}{2}} = \mathbf{H}^{\frac{1}{2}})$$

Since $\boldsymbol{\Omega}^{-1}$ and $\mathbf{H}^{\frac{1}{2}}\mathbf{H}^{-\frac{1}{2}}\mathbf{H}^{\frac{1}{2}}$ commute, they are simultaneously diagonalizable. The eigenvalues of $\boldsymbol{\Omega}^{-1}\mathbf{H}^{\frac{1}{2}}\boldsymbol{\Psi}\mathbf{H}^{\frac{1}{2}}$ are

$$\frac{\phi_i}{\frac{M+1}{M}\phi_i + \frac{\sum_i \phi_i + \sigma^2}{M}}, \quad i = 1, 2, ..., d.$$

Note that, every eigenvalue is upper bounded by 1, so we simply get

$$\lambda_{\max}\left(\mathbf{H}^{\frac{1}{2}}\boldsymbol{\Gamma}^*\mathbf{H}^{\frac{1}{2}}\right) \leq 1. \tag{G.17}$$

Therefore, we have

$$-\operatorname{tr}\left[\frac{1}{M}(\boldsymbol{\beta}-\boldsymbol{\beta}^*)^\top \mathbf{H}(\boldsymbol{\beta}-\boldsymbol{\beta}^*)\cdot\mathbf{H}\boldsymbol{\Gamma}^*\mathbf{H}\cdot\mathcal{P}_{\mathcal{Z}}(\boldsymbol{\Gamma}-\boldsymbol{\Gamma}^*)^\top\right]$$

$$\leq \frac{\sqrt{d}\lambda_1^2}{M}\exp\left(\frac{-2\lambda_{-1}t}{M+1}\right)\|\boldsymbol{\beta}(0)-\boldsymbol{\beta}^*\|_2^2\cdot\|\mathcal{P}_{\mathcal{Z}}(\boldsymbol{\Gamma}-\boldsymbol{\Gamma}^*)\|_F. \tag{G.18}$$

Similarly, we have

$$-\operatorname{tr}\left[\frac{1}{M}\mathbf{H}\boldsymbol{\Gamma}^*\mathbf{H}(\boldsymbol{\beta}-\boldsymbol{\beta}^*)(\boldsymbol{\beta}-\boldsymbol{\beta}^*)^\top\mathbf{H}\cdot\mathcal{P}_{\mathcal{Z}}(\boldsymbol{\Gamma}-\boldsymbol{\Gamma}^*)^\top\right]$$

$$\leq \frac{\sqrt{d}}{M}\left\|\mathcal{P}_{\mathcal{Z}}(\boldsymbol{\Gamma}-\boldsymbol{\Gamma}^*)^\top\right\|_F \left\|\mathbf{H}^{\frac{1}{2}}(\boldsymbol{\beta}-\boldsymbol{\beta}^*)(\boldsymbol{\beta}-\boldsymbol{\beta}^*)^\top\mathbf{H}^{\frac{1}{2}}\right\|_F \|\mathbf{H}\|_F \left\|\mathbf{H}^{\frac{1}{2}}\boldsymbol{\Gamma}^*\mathbf{H}^{\frac{1}{2}}\right\|_2 \quad \text{(Lemma H.1)}$$

$$\leq \frac{\sqrt{d}\lambda_1^2}{M}\exp\left(\frac{-2\lambda_{-1}t}{M+1}\right)\|\boldsymbol{\beta}(0)-\boldsymbol{\beta}^*\|_2^2\cdot\|\mathcal{P}_{\mathcal{Z}}(\boldsymbol{\Gamma}-\boldsymbol{\Gamma}^*)\|_F, \tag{G.19}$$

and

$$\operatorname{tr}\left[\frac{1}{M}\mathbf{H}^{\frac{1}{2}}\boldsymbol{\Omega}^{-1}\left(\mathbf{H}^{\frac{1}{2}}\boldsymbol{\Psi}\mathbf{H}^{\frac{1}{2}}+\left(\operatorname{tr}\left(\mathbf{H}^{\frac{1}{2}}\boldsymbol{\Psi}\mathbf{H}^{\frac{1}{2}}\right)+\sigma^2\right)\mathbf{I}_d\right)\mathbf{H}^{\frac{1}{2}}(\boldsymbol{\beta}-\boldsymbol{\beta}^*)(\boldsymbol{\beta}-\boldsymbol{\beta}^*)^\top\mathbf{H}\cdot\mathcal{P}_{\mathcal{Z}}(\boldsymbol{\Gamma}-\boldsymbol{\Gamma}^*)^\top\right]$$

$$\leq \frac{\sqrt{d}}{M}\left\|\boldsymbol{\Omega}^{-1}\left(\mathbf{H}^{\frac{1}{2}}\boldsymbol{\Psi}\mathbf{H}^{\frac{1}{2}}+\left(\operatorname{tr}\left(\mathbf{H}^{\frac{1}{2}}\boldsymbol{\Psi}\mathbf{H}^{\frac{1}{2}}\right)+\sigma^2\right)\mathbf{I}_d\right)\right\|_2 \left\|\mathbf{H}^{\frac{1}{2}}(\boldsymbol{\beta}-\boldsymbol{\beta}^*)(\boldsymbol{\beta}-\boldsymbol{\beta}^*)^\top\mathbf{H}\cdot\mathcal{P}_{\mathcal{Z}}(\boldsymbol{\Gamma}-\boldsymbol{\Gamma}^*)^\top\mathbf{H}^{\frac{1}{2}}\right\|_F$$

$$\text{(Lemma H.1)}$$

$$\leq \frac{\sqrt{d}}{M}\cdot M\cdot\left\|\mathbf{H}^{\frac{1}{2}}(\boldsymbol{\beta}-\boldsymbol{\beta}^*)(\boldsymbol{\beta}-\boldsymbol{\beta}^*)^\top\mathbf{H}^{\frac{1}{2}}\right\|_F\cdot\|\mathbf{H}\|_F\cdot\|\mathcal{P}_{\mathcal{Z}}(\boldsymbol{\Gamma}-\boldsymbol{\Gamma}^*)\|_F$$

$$\leq \sqrt{d}\lambda_1^2\exp\left(\frac{-2\lambda_{-1}t}{M+1}\right)\|\boldsymbol{\beta}(0)-\boldsymbol{\beta}^*\|_2^2\cdot\|\mathcal{P}_{\mathcal{Z}}(\boldsymbol{\Gamma}-\boldsymbol{\Gamma}^*)\|_F \tag{G.20}$$

Bridging (G.18), (G.19) and (G.20) into (G.16), we get

$$G_3 \leq \left(2+\frac{2}{M}\right)\lambda_1^2\exp\left(\frac{-2\lambda_{-1}t}{M+1}\right)\|\boldsymbol{\beta}(0)-\boldsymbol{\beta}^*\|_2^2\cdot\|\mathcal{P}_{\mathcal{Z}}(\boldsymbol{\Gamma}-\boldsymbol{\Gamma}^*)\|_F \tag{G.21}$$

$\square$

Finally, we finish this section by proving the convergence of $\boldsymbol{\Gamma}$.

**Lemma G.6** (Convergence of $\boldsymbol{\Gamma}$). *Under the dynamical system* (G.4) *and* (G.5)*, one has*

$$\mathcal{P}_{\mathcal{Z}}(\boldsymbol{\Gamma}(t)) \to \mathcal{P}_{\mathcal{Z}}(\boldsymbol{\Gamma}^*), \quad \mathcal{P}_{\mathcal{Z}^\perp}(\boldsymbol{\Gamma}(t)) = \mathcal{P}_{\mathcal{Z}^\perp}(\boldsymbol{\Gamma}(0))$$

*when* $t \to \infty$, *from arbitrary initialization* $\boldsymbol{\beta}(0)$ *and* $\boldsymbol{\Gamma}(0)$.

*Proof.* We observe that

$$\frac{\mathrm{d}}{\mathrm{d}t}\left[\frac{1}{2}\|\mathcal{P}_{\mathcal{Z}}(\boldsymbol{\Gamma}-\boldsymbol{\Gamma}^*)\|_F^2\right] = \|\mathcal{P}_{\mathcal{Z}}(\boldsymbol{\Gamma}-\boldsymbol{\Gamma}^*)\|_F\cdot\frac{\mathrm{d}}{\mathrm{d}t}\|\mathcal{P}_{\mathcal{Z}}(\boldsymbol{\Gamma}-\boldsymbol{\Gamma}^*)\|_F.$$

Combining it with Lemma G.5, we have

$$\frac{\mathrm{d}}{\mathrm{d}t}\|\mathcal{P}_{\mathcal{Z}}(\boldsymbol{\Gamma}-\boldsymbol{\Gamma}^*)\|_F \leq -A_1\|\mathcal{P}_{\mathcal{Z}}(\boldsymbol{\Gamma}-\boldsymbol{\Gamma}^*)\|_F + A_2\exp\left(\frac{-2\lambda_{-1}t}{M+1}\right),$$

where $A_1 > 0, A_2 > 0$ are defined in (G.14). Simple calculation shows that

$$\frac{\mathrm{d}}{\mathrm{d}t} \left[ \exp\left(A_1 t\right) \| \mathcal{P}_{\mathcal{Z}} \left(\mathbf{\Gamma} - \mathbf{\Gamma}^*\right) \|_F \right] \le A_2 \exp\left[ \left( A_1 - \frac{2\lambda_{-1}}{M+1} \right) t \right]. \tag{G.22}$$

When $A_1 \ne \frac{2\lambda_{-1}}{M+1}$, we integrate both sides from $t = 0$ to $t = T$, then divide them by $\exp\left(A_1 T\right)$. This gives

$$\| \mathcal{P}_{\mathcal{Z}} \left(\mathbf{\Gamma}(T) - \mathbf{\Gamma}^*\right) \|_F \le \frac{\| \mathcal{P}_{\mathcal{Z}} \left(\mathbf{\Gamma}(0) - \mathbf{\Gamma}^*\right) \|_F}{\exp\left(A_1 T\right)} + \frac{A_2}{A_1 - \frac{2\lambda_{-1}}{M+1}} \cdot \frac{\exp\left[ \left( A_1 - \frac{2\lambda_{-1}}{M+1} \right) T \right] - 1}{\exp\left(A_1 T\right)}$$

$$= \frac{\| \mathcal{P}_{\mathcal{Z}} \left(\mathbf{\Gamma}(0) - \mathbf{\Gamma}^*\right) \|_F}{\exp\left(A_1 T\right)} + \frac{A_2}{A_1 - \frac{2\lambda_{-1}}{M+1}} \cdot \left[ \exp\left( -\frac{2\lambda_{-1} T}{M+1} \right) - \exp\left(-A_1 T\right) \right] \to 0$$

when $T \to \infty$. Otherwise, if $A_1 = \frac{2\lambda_{-1}}{M+1}$, the right hand side of (G.22) reduces to a constant, which implies $\exp\left(A_1 t\right) \| \mathcal{P}_{\mathcal{Z}} \left(\mathbf{\Gamma}(t) - \mathbf{\Gamma}^*\right) \|_F$ grows at most at a linear rate, which implies $\| \mathcal{P}_{\mathcal{Z}} \left(\mathbf{\Gamma}(t) - \mathbf{\Gamma}^*\right) \|_F \to 0$ when $t \to \infty$. Together, in all cases, we have $\| \mathcal{P}_{\mathcal{Z}} \left(\mathbf{\Gamma}(t) - \mathbf{\Gamma}^*\right) \|_F \to 0$. From (G.9), we soon get $\mathcal{P}_{\mathcal{Z}} \left(\mathbf{\Gamma}(t)\right) = \mathcal{P}_{\mathcal{Z}} \left(\mathbf{\Gamma}(0)\right)$ foo all $t > 0$. Therefore, we conclude. $\qquad \square$

The Theorem 6.3 is proved by simplly combining Lemma G.3 and Lemma G.6.

# H Technical lemmas

**Lemma H.1** (Von-Neumann's Trace Inequality). *Let $U, V \in \mathbb{R}^{d \times n}$ with $d \leq n$. We have*

$$\text{tr}\left(U^\top V\right) \leq \sum_{i=1}^{d} \sigma_i(U)\sigma_i(V) \leq \|U\|_{\text{op}} \times \sum_{i=1}^{d} \sigma_i(V) \leq \sqrt{d} \cdot \|U\|_{\text{op}}\|V\|_F$$

*where $\sigma_1(X) \geq \sigma_2(X) \geq \cdots \geq \sigma_d(X)$ are the ordered singular values of $X \in \mathbb{R}^{d \times n}$.*

**Lemma H.2** ([22]). *For any two positive semi-definite matrices $A, B \in \mathbb{R}^{d \times d}$, we have*

- $\text{tr}[AB] \geq 0$.

- $AB \succeq 0$ *if and only if $A$ and $B$ commute.*

**Lemma H.3** (Derivatives, [27]). *We denote $\mathbf{A}, \mathbf{B}, \mathbf{C}, \mathbf{D}, \mathbf{X}$ as matrices and $\mathbf{a}, \mathbf{b}, \mathbf{x}$ as vectors. Then, we have*

- $\frac{\partial}{\partial \mathbf{X}} \text{tr}\left[\mathbf{X}^\top \mathbf{B}\mathbf{X}\mathbf{C}\right] = \mathbf{B}\mathbf{X}\mathbf{C} + \mathbf{B}^\top \mathbf{X}\mathbf{C}^\top$.

- $\frac{\partial \mathbf{x}^\top \mathbf{B}\mathbf{x}}{\partial \mathbf{x}} = \left(\mathbf{B} + \mathbf{B}^\top\right)\mathbf{x}$.

- $\frac{\partial \mathbf{a}^\top \mathbf{X}\mathbf{b}}{\partial \mathbf{X}} = \mathbf{a}\mathbf{b}^\top$.

- $\frac{\partial \mathbf{a}^\top \mathbf{X}^\top \mathbf{b}}{\partial \mathbf{X}} = \mathbf{b}\mathbf{a}^\top$.

- $\frac{\partial \mathbf{b}^\top \mathbf{X}^\top \mathbf{D}\mathbf{X}\mathbf{c}}{\partial \mathbf{X}} = \mathbf{D}^\top \mathbf{X}\mathbf{b}\mathbf{c}^\top + \mathbf{D}\mathbf{X}\mathbf{c}\mathbf{b}^\top$.

- $\frac{\partial}{\partial \mathbf{X}} \text{tr}\left[(\mathbf{A}\mathbf{X}\mathbf{B} + \mathbf{C})(\mathbf{A}\mathbf{X}\mathbf{B} + \mathbf{C})^\top\right] = 2\mathbf{A}^\top(\mathbf{A}\mathbf{X}\mathbf{B} + \mathbf{C})\mathbf{B}^\top$

**Lemma H.4** (Lemma D.2 in [38], Lemma 4.2 in [37]). *If $\mathbf{x}$ is Gaussian random vector of $d$ dimension, mean zero and covariance matrix $\mathbf{H}$, and $A \in \mathbb{R}^{d \times d}$ is a fixed matrix. Then*

$$\mathbb{E}\left[\mathbf{x}\mathbf{x}^\top A\mathbf{x}\mathbf{x}^\top\right] = \mathbf{H}\left(A + A^\top\right)\mathbf{H} + \text{tr}(A\mathbf{H})\mathbf{H}.$$

*If $A$ is symmetric and the rows in $\mathbf{X} \in \mathbb{R}^{M \times d}$ are generated independently from*

$$\mathbf{X}[i] \sim \mathcal{N}(0, \mathbf{H}), \quad i = 1, \ldots, M.$$

*Then, it holds that*

$$\mathbb{E}\left[\mathbf{X}^\top \mathbf{X}A\mathbf{X}^\top \mathbf{X}\right] = M \cdot \text{tr}\left(\mathbf{H}A\right) \cdot \mathbf{H} + M(M+1) \cdot \mathbf{H}A\mathbf{H}.$$

**Lemma H.5** (Linear Algebra). *Suppose $\mathbf{H}$ is a (non-zero) positive semi-definite matrix in $\mathbb{R}^{d \times d}$ and $\mathbf{H}^{\frac{1}{2}}$ is its principle square root. We denote $\lambda_{-1}$ as the minimal non-zero eigenvector of $\mathbf{H}$. Then, we have*

- *1.* $\text{null}\left(\mathbf{H}\right) = \text{null}\left(\mathbf{H}^{\frac{1}{2}}\right), \text{Im}\left(\mathbf{H}\right) = \text{Im}\left(\mathbf{H}^{\frac{1}{2}}\right).$

- *2.* $\mathbf{H}\mathbf{H}^+ = \mathbf{H}^+\mathbf{H}$, *where $(\cdot)^+$ denotes the Moore-Penrose pseudo-inverse.*

- *3. For any vector $\boldsymbol{\alpha} \in \mathbb{R}^d$, the orthogonal projection operator on $\text{Im}\left(\mathbf{H}\right)$ and $\text{null}\left(\mathbf{H}\right)$ are respectively*

  $$\mathcal{P}_{\text{Im}(\mathbf{H})}\left(\boldsymbol{\alpha}\right) = \mathbf{H}\mathbf{H}^+\boldsymbol{\alpha} = \mathbf{H}^+\mathbf{H}\boldsymbol{\alpha}, \quad \mathcal{P}_{\text{null}(\mathbf{H})}\left(\boldsymbol{\alpha}\right) = \left(\mathbf{I}_d - \mathbf{H}\mathbf{H}^+\right)\boldsymbol{\alpha} = \left(\mathbf{I}_d - \mathbf{H}^+\mathbf{H}\right)\boldsymbol{\alpha}.$$

- *4. For any vector $\boldsymbol{\alpha} \in \mathbb{R}^d$, we have*

  $$\mathcal{P}_{\text{Im}(\mathbf{H})}\left(\boldsymbol{\alpha}\right)^\top \mathbf{H}\mathcal{P}_{\text{Im}(\mathbf{H})}\left(\boldsymbol{\alpha}\right) \geq \lambda_{-1}\left\|\mathcal{P}_{\text{Im}(\mathbf{H})}\left(\boldsymbol{\alpha}\right)\right\|_2^2$$

*Proof.* We consider the eigen-decomposition of matrix $\mathbf{H}$, which is

$$\mathbf{H} = \mathbf{Q}D\mathbf{Q}^\top, \tag{H.1}$$

where $D = \text{diag}\,(\lambda_1, \lambda_2, ..., \lambda_d)$ is a diagonal matrix with diagonal entries being the eigenvalues of $\mathbf{H}$, and $\mathbf{Q}$ is an orthogonal matrix. Then, from the definition of principle square root, we know

$$\mathbf{H}^{\frac{1}{2}} = \mathbf{Q}D^{\frac{1}{2}}\mathbf{Q}^\top, \tag{H.2}$$

where $D^{\frac{1}{2}} = \text{diag}\,\left(\sqrt{\lambda_1}, \sqrt{\lambda_2}, ..., \sqrt{\lambda_d}\right)$. We denote columns of $\mathbf{Q}$ as $\mathbf{q}_1, ...\mathbf{q}_d$. We know they are the eigenvectors of $\mathbf{H}$. From the eigen-decomposition of $\mathbf{H}$ and $\mathbf{H}^{\frac{1}{2}}$, we know they share the same set of eigenvectors. Without loss of generality, we assume $\text{rank}(\mathbf{H}) = r$ and $\lambda_1 \geq \lambda_2 \geq ... \geq \lambda_r > 0, \lambda_i = 0, i = r + 1, ..., d$. Then, we denote $\mathcal{H}$ as the linear vector space spanned by $\mathbf{q}_1, ..., \mathbf{q}_r$ and $\mathcal{H}^\perp$ as its orthogonal complement (which is the subspace spanned by $\mathbf{q}_{r+1}, ..., \mathbf{q}_d$). For any vector $\boldsymbol{\alpha} \in \mathbb{R}^d$, we can write its orthogonal decomposition as $\boldsymbol{\alpha} = \sum_{i=1}^d a_i \mathbf{q}_i$, so we have

$$\mathbf{H}\boldsymbol{\alpha} = \sum_{i=1}^d a_i \mathbf{H}\mathbf{q}_i = \sum_{i=1}^r a_i \lambda_i \mathbf{q}_i \in \mathcal{H},$$

which implies $\text{Im}\,(\mathbf{H}) \subset \mathcal{H}$. On the other hand, we know for any $\boldsymbol{\alpha} \in \mathcal{H}$, we can write it as $\boldsymbol{\alpha} = \sum_{i=1}^r a_i \mathbf{q}_i$ for some $a_1, ..., a_r$, then we have

$$\boldsymbol{\alpha} = \sum_{i=1}^r \frac{a_i}{\lambda_i} \cdot \lambda_i \mathbf{q}_i = \mathbf{H}\left(\sum_{i=1}^r \frac{a_i}{\lambda_i} \mathbf{q}_i\right) \in \text{Im}\,(\mathbf{H}),$$

which implies $\mathcal{H} \subset \text{Im}\,(\mathbf{H})$. This shows

$$\text{Im}\,(\mathbf{H}) = \mathcal{H} = \text{span}\,\{\mathbf{q}_1, ...\mathbf{q}_r\}\,,$$

and

$$\text{null}\,(\mathbf{H}) = \mathcal{H} = \text{span}\,\{\mathbf{q}_{r+1}, ...\mathbf{q}_d\}\,,$$

since $\text{null}\,(\mathbf{H})$ is the orthogonal complement of $\text{Im}\,(\mathbf{H})$. Then, (1) is proved by noticing that $\mathbf{H}$ and $\mathbf{H}^{\frac{1}{2}}$ share the same set of eigenvectors. To prove (2), it suffices to notice that

$$\mathbf{H}\mathbf{H}^+ = \mathbf{Q} \cdot \text{diag}(\underbrace{1, 1, ..., 1}_{r}, 0, 0, .., 0) \cdot \mathbf{Q}^\top = \mathbf{H}^+\mathbf{H}.$$

To prove (3), it suffices to notice that actually $\mathbf{H}\mathbf{H}^+\boldsymbol{\alpha} \in \text{Im}\,(\mathbf{H})$ and

$$\left\langle \mathbf{H}\mathbf{H}^+\boldsymbol{\alpha}, \left(\mathbf{I}_d - \mathbf{H}\mathbf{H}^+\right)\boldsymbol{\alpha}\right\rangle = \left\langle \mathbf{H}^+\mathbf{H}\boldsymbol{\alpha}, \left(\mathbf{I}_d - \mathbf{H}\mathbf{H}^+\right)\boldsymbol{\alpha}\right\rangle = \boldsymbol{\alpha}^\top\left(\mathbf{H}\mathbf{H}^+ - \mathbf{H}\mathbf{H}^+\mathbf{H}\mathbf{H}^+\right)\boldsymbol{\alpha} = 0.$$

Finally, to prove (4), we can write $\boldsymbol{\alpha} = \sum_{i=1}^d a_i \mathbf{q}_i$ for some $a_1, ..., a_d$. Then, by orthogonality we have

$$\mathcal{P}_{\text{Im}(\mathbf{H})}\,(\boldsymbol{\alpha})^\top \mathbf{H}\mathcal{P}_{\text{Im}(\mathbf{H})}\,(\boldsymbol{\alpha}) = \left(\sum_{i=1}^d a_i \mathbf{q}_i\right)^\top \mathbf{H}\left(\sum_{i=1}^d a_i \mathbf{q}_i\right) = \sum_{i=1}^r a_i^2 \lambda_i \mathbf{q}_i^\top \mathbf{q}_i \geq \lambda_{-1} \cdot \sum_{i=1}^r a_i^2 \mathbf{q}_i^\top \mathbf{q}_i$$

$$= \lambda_{-1}\left\|\mathcal{P}_{\text{Im}(\mathbf{H})}\,(\boldsymbol{\alpha})\right\|_2^2.$$

Therefore, we conclude. $\qquad\square$

**Lemma H.6** (Tensor product). *Suppose $\mathbf{H}$ is a (non-zero) positive semi-definite matrix in $\mathbb{R}^{d\times d}$ and $\mathbf{H}^{\frac{1}{2}}$ is its principle square root. We denote $\otimes$ as Kronecker product, which is defined as*

$$(\boldsymbol{A} \otimes \boldsymbol{B}) \circ \boldsymbol{C} = \boldsymbol{B}\boldsymbol{C}\boldsymbol{A}^\top.$$

*We define*

$$\text{Im}\,(\mathbf{H} \otimes \mathbf{H}) := \left\{(\mathbf{H} \otimes \mathbf{H}) \circ \boldsymbol{Z} : \boldsymbol{Z} \in \mathbb{R}^{d\times d}\right\}, \quad \text{null}\,(\mathbf{H} \otimes \mathbf{H}) := \left\{\boldsymbol{Z} \in \mathbb{R}^{d\times d} : (\mathbf{H} \otimes \mathbf{H}) \circ \boldsymbol{Z} = \mathbf{0}\right\},$$

*and define* $\text{Im}\,\left(\mathbf{H}^{\frac{1}{2}} \otimes \mathbf{H}^{\frac{1}{2}}\right)$ *and* $\text{null}\,\left(\mathbf{H}^{\frac{1}{2}} \otimes \mathbf{H}^{\frac{1}{2}}\right)$ *similarly. Then, we have*

- *1*. $\mathsf{Im}\left(\mathbf{H}^{\frac{1}{2}} \otimes \mathbf{H}^{\frac{1}{2}}\right) = \mathsf{Im}\left(\mathbf{H} \otimes \mathbf{H}\right), \mathsf{null}\left(\mathbf{H}^{\frac{1}{2}} \otimes \mathbf{H}^{\frac{1}{2}}\right) = \mathsf{null}\left(\mathbf{H} \otimes \mathbf{H}\right).$

- *2*. *We denote* $\mathcal{Z} = \mathsf{Im}\left(\mathbf{H}^{\frac{1}{2}} \otimes \mathbf{H}^{\frac{1}{2}}\right) = \mathsf{Im}\left(\mathbf{H} \otimes \mathbf{H}\right)$ *and* $\mathcal{Z}^{\perp} = \mathsf{null}\left(\mathbf{H}^{\frac{1}{2}} \otimes \mathbf{H}^{\frac{1}{2}}\right) = \mathsf{null}\left(\mathbf{H} \otimes \mathbf{H}\right).$ *For any matrix* $\mathbf{Z} \in \mathbb{R}^{d \times d}$, *we have*

$$\mathcal{P}_{\mathcal{Z}}\left(\mathbf{Z}\right) = \left(\mathbf{H}\mathbf{H}^{+}\right)^{\otimes 2} \circ \mathbf{Z} = \mathbf{H}\mathbf{H}^{+}\mathbf{Z}\mathbf{H}^{+}\mathbf{H} = \mathbf{H}^{\frac{1}{2}}\mathbf{H}^{-\frac{1}{2}}\mathbf{Z}\mathbf{H}^{-\frac{1}{2}}\mathbf{H}^{\frac{1}{2}} \tag{H.3}$$

$$\mathcal{P}_{\mathcal{Z}^{\perp}}\left(\mathbf{Z}\right) = \left[\mathbf{I}_d^{\otimes 2} - \left(\mathbf{H}\mathbf{H}^{+}\right)^{\otimes 2}\right] \circ \mathbf{Z} = \mathbf{Z} - \mathbf{H}\mathbf{H}^{+}\mathbf{Z}\mathbf{H}^{+}\mathbf{H} = \mathbf{Z} - \mathbf{H}^{\frac{1}{2}}\mathbf{H}^{-\frac{1}{2}}\mathbf{Z}\mathbf{H}^{-\frac{1}{2}}\mathbf{H}^{\frac{1}{2}}. \tag{H.4}$$

- *3*. *For any matrix* $\mathbf{Z} \in \mathbb{R}^{d \times d}$, *it holds that*

$$\left(\left(\mathbf{H}^{\frac{1}{2}} \otimes \mathbf{H}^{\frac{1}{2}}\right) \circ \mathcal{P}_{\mathcal{Z}}\left(\mathbf{Z}\right)\right) \cdot \left(\left(\mathbf{H}^{\frac{1}{2}} \otimes \mathbf{H}^{\frac{1}{2}}\right) \circ \mathcal{P}_{\mathcal{Z}}\left(\mathbf{Z}\right)\right)^{\top} \succeq \lambda_{-1}^{2} \mathcal{P}_{\mathcal{Z}}\left(\mathbf{Z}\right) \cdot \mathcal{P}_{\mathcal{Z}}\left(\mathbf{Z}\right)^{\top}, \tag{H.5}$$

*where* $\lambda_{-1}$ *is the minimal positive eigenvector of* $\mathbf{H}$.

*Proof.* Let's first prove the second part. From the definition of tensor product and the fact that $\mathbf{H}\mathbf{H}^{+} = \mathbf{H}^{+}\mathbf{H}$ (see Lemma H.5), we know the second equation in H.3 holds. To prove the first equation, it suffices to notice that for any matrix $\mathbf{Z} \in \mathbb{R}^{d \times d}$, the matrix $\mathbf{H}\mathbf{H}^{+}\mathbf{Z}\mathbf{H}^{+}\mathbf{H}$ is in $\mathsf{Im}\left(\mathbf{H}^{\otimes 2}\right)$, together with the fact that

$$\begin{aligned}
\left\langle \mathbf{H}\mathbf{H}^{+}\mathbf{Z}\mathbf{H}^{+}\mathbf{H}, \mathbf{Z} - \mathbf{H}\mathbf{H}^{+}\mathbf{Z}\mathbf{H}^{+}\mathbf{H} \right\rangle &= \mathrm{tr}\left(\mathbf{H}\mathbf{H}^{+}\mathbf{Z}\mathbf{H}^{+}\mathbf{H}\mathbf{Z}^{\top}\right) - \mathrm{tr}\left(\mathbf{H}\mathbf{H}^{+}\mathbf{Z}\mathbf{H}^{+}\mathbf{H}\mathbf{H}\mathbf{H}^{+}\mathbf{Z}^{\top}\mathbf{H}^{+}\mathbf{H}\right) \\
&= \mathrm{tr}\left(\mathbf{H}\mathbf{H}^{+}\mathbf{Z}\mathbf{H}^{+}\mathbf{H}\mathbf{Z}^{\top}\right) - \mathrm{tr}\left(\mathbf{H}\mathbf{H}^{+}\mathbf{H}\mathbf{H}^{+}\mathbf{Z}\mathbf{H}^{+}\mathbf{H}\mathbf{H}^{+}\mathbf{H}\mathbf{Z}^{\top}\right) \\
&\qquad\qquad\qquad\qquad\qquad\qquad\qquad\qquad\qquad (\mathbf{H}\mathbf{H}^{+} = \mathbf{H}^{+}\mathbf{H}) \\
&= 0.
\end{aligned}$$

The proof of (H.4) is similar. Then, from Lemma H.5, we know $\mathsf{Im}\left(\mathbf{H}\right) = \mathsf{Im}\left(\mathbf{H}^{\frac{1}{2}}\right)$, which implies the projection operator onto those two subspace are identical, indicating $\mathbf{H}\mathbf{H}^{+} = \mathbf{H}^{\frac{1}{2}}\mathbf{H}^{-\frac{1}{2}}$. Therefore, we have

$$\mathcal{P}_{\mathsf{Im}(\mathbf{H} \otimes \mathbf{H})}\left(\mathbf{Z}\right) = \left(\mathbf{H}\mathbf{H}^{+}\right)^{\otimes 2} \circ \mathbf{Z} = \mathbf{H}\mathbf{H}^{+}\mathbf{Z}\mathbf{H}^{+}\mathbf{H} = \mathbf{H}^{\frac{1}{2}}\mathbf{H}^{-\frac{1}{2}}\mathbf{Z}\mathbf{H}^{-\frac{1}{2}}\mathbf{H}^{\frac{1}{2}} = \mathcal{P}_{\mathsf{Im}\left(\mathbf{H}^{\frac{1}{2}} \otimes \mathbf{H}^{\frac{1}{2}}\right)}\left(\mathbf{Z}\right).$$

Since this holds for any matrix $\mathbf{Z} \in \mathbb{R}^{d \times d}$, this suggests $\mathsf{Im}\left(\mathbf{H}^{\frac{1}{2}} \otimes \mathbf{H}^{\frac{1}{2}}\right) = \mathsf{Im}\left(\mathbf{H} \otimes \mathbf{H}\right)$. Similarly, we also have $\mathsf{null}\left(\mathbf{H}^{\frac{1}{2}} \otimes \mathbf{H}^{\frac{1}{2}}\right) = \mathsf{null}\left(\mathbf{H} \otimes \mathbf{H}\right)$. Finally, to prove (3), we use the eigendecomposition of $\mathbf{H}$ and $\mathbf{H}^{\frac{1}{2}}$:

$$\mathbf{H} = \mathbf{Q}D\mathbf{Q}^{\top}, \quad \mathbf{H}^{\frac{1}{2}} = \mathbf{Q}D^{\frac{1}{2}}\mathbf{Q}^{\top},$$

where $\mathbf{Q} = (\mathbf{q}_1\,\mathbf{q}_2\,...\,\mathbf{q}_d)$ is an orthogonal matrix and $D = \mathrm{diag}\left(\lambda_1, ...., \lambda_d\right)$, where $\lambda_1 \geq \lambda_2 \geq ....\lambda_d \geq 0$ are ordered eigenvalues. We assume $\mathsf{rank}(\mathbf{H}) = r$ and $\lambda_r > 0 = \lambda_{r+1}$. Then, from the property of Kronecker product (eg. see [27]), the eigenvalues of $\mathbf{H}^{\frac{1}{2}} \otimes \mathbf{H}^{\frac{1}{2}}$ are $\left\{\sqrt{\lambda_i \lambda_j} : 1 \leq i, j \leq d\right\}$. The eigenvectors are $\{\mathbf{q}_i \otimes \mathbf{q}_j : 1 \leq i, j \leq d\}$, which forms an orthogonal unit basis of $\mathbb{R}^{d \times d}$. We define

$$\mathcal{S} := \mathsf{span}\left\{\mathbf{q}_i \otimes \mathbf{q}_j : 1 \leq i, j \leq r\right\}$$

as the eigenspace spanned by all eigenvectors corresponding to positive eigenvalues. For any matrix $\mathbf{Z}$, we can decompose it as

$$\mathbf{Z} = \sum_{i,j} a_{i,j} \cdot \mathbf{q}_i \otimes \mathbf{q}_j,$$

so

$$\left(\mathbf{H} \otimes \mathbf{H}\right) \circ \mathbf{Z} = \sum_{i,j=1}^{d} a_{i,j}\left(\mathbf{H} \otimes \mathbf{H}\right) \circ \left(\mathbf{q}_i \otimes \mathbf{q}_j\right) = \sum_{i,j}\left(\mathbf{H}\mathbf{q}_i\right) \otimes \left(\mathbf{H}\mathbf{q}_j\right)$$

$$= \sum_{i,j=1}^{d} a_{i,j} \lambda_i \lambda_j \mathbf{q}_i \otimes \mathbf{q}_j = \sum_{i,j=1}^{r} a_{i,j} \lambda_i \lambda_j \mathbf{q}_i \otimes \mathbf{q}_j \in \mathcal{S},$$

which implies $\mathcal{Z} \subset \mathcal{S}$. On the other hand, for any $\mathbf{Z} \in \mathcal{S}$, we can write it as $\mathbf{Z} = \sum_{i,j=1}^{r} b_{i,j} \cdot \mathbf{q}_i \otimes \mathbf{q}_j$, so we have

$$\mathbf{Z} = \sum_{i,j=1}^{r} \frac{b_{i,j}}{\lambda_i \lambda_j} \cdot (\lambda_i \mathbf{q}_i) \otimes (\lambda_j \mathbf{q}_j) = \sum_{i,j=1}^{r} \frac{b_{i,j}}{\lambda_i \lambda_j} \cdot (\mathbf{H} \mathbf{q}_i) \otimes (\mathbf{H} \mathbf{q}_j) = (\mathbf{H} \otimes \mathbf{H}) \circ \left[ \sum_{i,j=1}^{r} \frac{b_{i,j}}{\lambda_i \lambda_j} (\mathbf{q}_i \otimes \mathbf{q}_j) \right] \in \mathcal{Z},$$

which implies $\mathcal{S} \subset \mathcal{Z}$. Combining two directions, we have $\mathcal{S} = \mathcal{Z}$. Therefore, for any matrix $\mathbf{Z} \in \mathbb{R}^{d \times d}$, we can write it as $\mathbf{Z} = \sum_{i,j=1}^{d} c_{i,j} \cdot \mathbf{q}_i \otimes \mathbf{q}_j$ for some $c_{i,j}$. Then, we have

$$\left( \mathbf{H}^{\frac{1}{2}} \otimes \mathbf{H}^{\frac{1}{2}} \right) \circ \mathcal{P}_{\mathcal{Z}} (\mathbf{Z}) = \left( \mathbf{H}^{\frac{1}{2}} \otimes \mathbf{H}^{\frac{1}{2}} \right) \circ \mathcal{P}_{\mathcal{S}} (\mathbf{Z}) = \left( \mathbf{H}^{\frac{1}{2}} \otimes \mathbf{H}^{\frac{1}{2}} \right) \circ \sum_{i,j=1}^{r} c_{i,j} \cdot \mathbf{q}_i \otimes \mathbf{q}_j$$

$$= \sum_{i,j=1}^{r} c_{i,j} \left( \mathbf{H}^{\frac{1}{2}} \mathbf{q}_i \right) \otimes \left( \mathbf{H}^{\frac{1}{2}} \mathbf{q}_j \right) = \sum_{i,j=1}^{r} c_{i,j} \sqrt{\lambda_i \lambda_j} \cdot \mathbf{q}_i \otimes \mathbf{q}_j.$$

Therefore, we have

$$\left( \left( \mathbf{H}^{\frac{1}{2}} \otimes \mathbf{H}^{\frac{1}{2}} \right) \circ \mathcal{P}_{\mathcal{Z}} (\mathbf{Z}) \right) \left( \left( \mathbf{H}^{\frac{1}{2}} \otimes \mathbf{H}^{\frac{1}{2}} \right) \circ \mathcal{P}_{\mathcal{Z}} (\mathbf{Z}) \right)^{\top}$$

$$= \left( \sum_{i,j=1}^{r} c_{i,j} \sqrt{\lambda_i \lambda_j} \cdot \mathbf{q}_i \otimes \mathbf{q}_j \right) \left( \sum_{k,l=1}^{r} c_{k,l} \sqrt{\lambda_k \lambda_l} \cdot \mathbf{q}_k \otimes \mathbf{q}_l \right)^{\top}$$

$$= \sum_{i,j=1}^{r} c_{i,j}^2 \lambda_i \lambda_j \left( \mathbf{q}_i \otimes \mathbf{q}_j \right) \left( \mathbf{q}_i \otimes \mathbf{q}_j \right)^{\top} \qquad \text{(By orthogonality)}$$

$$\succeq \lambda_{-1}^2 \sum_{i,j=1}^{r} c_{i,j}^2 \left( \mathbf{q}_i \otimes \mathbf{q}_j \right) \left( \mathbf{q}_i \otimes \mathbf{q}_j \right)^{\top}$$

$$= \lambda_{-1}^2 \mathcal{P}_{\mathcal{Z}} (\mathbf{Z}) \cdot \mathcal{P}_{\mathcal{Z}} (\mathbf{Z})^{\top}.$$

Therefore, we conclude. $\qquad\square$

## I  Experiment Details

Our experiments on GPT2 mostly follow the setting in [14], except that we use the token matrix defined in (3.1). In our experiments, we use $d = 20$, $M = 40$, $\mathbf{\Psi} = \mathbf{H} = \mathbf{I}_d$ and $\sigma = 0$. We train the GPT2 with and without MLP layers on two settings: $\boldsymbol{\beta}^* = (0, 0, ..., 0)^{\top}$ and $\boldsymbol{\beta}^* = (10, 10, ..., 10)^{\top}$. For GPT2 with and without MLP layers, we initialize the $\mathbf{W}_V$ and $\mathbf{W}_2$ by normal distribution with standard deviation $0.02/\sqrt{\text{number of residual connections}}$, where the number of residual connections equals the number of layers for GPT2 without MLP layers. For GPT2 with MLP layers, this is 2 times the number of layers. Initialization for other matrices follow the default setting in [36]. We use Adam with learning rate $0.0001$. We train the model for $200000$ steps and we sample a batch of $256$ new tasks for each step.

## J  Does scratchpad help?

In this section, we show the limitations of adding a scratchpad to the token matrix. We will show that by using a single LSA layer and the token matrix with scratchpad, one cannot recover the GD-$\boldsymbol{\beta}$ estimator defined in Section 5. We leave it as future work to see whether the token matrix with scratchpad could implement other types of estimators that more effectively address the linear regression tasks defined in Assumption 3.1, as well as whether additional structures could help alleviate this approximation error. We follow the notations in previous sections. The token matrix with scratchpad is defined as

$$\mathbf{E} := \begin{pmatrix} \mathbf{X}^\top & \mathbf{x} \\ \mathbf{1}_M^\top & 1 \\ \mathbf{y}^\top & 0 \end{pmatrix} \in \mathbb{R}^{(d+2)\times(M+1)}. \tag{J.1}$$

where $\mathbf{1}_M \in \mathbb{R}^M$ refers to the vector filled with ones.

A LSA model $f$, is defined by

$$f(\mathbf{E}) := \left[\mathbf{E} + \mathbf{W}_P^\top \mathbf{W}_V \mathbf{E} \mathbf{M} \frac{\mathbf{E}^\top \mathbf{W}_K^\top \mathbf{W}_Q \mathbf{E}}{M}\right]_{-1,-1} = \left[\mathbf{W}_P^\top \mathbf{W}_V \mathbf{E} \mathbf{M} \frac{\mathbf{E}^\top \mathbf{W}_K^\top \mathbf{W}_Q \mathbf{E}}{M}\right]_{-1,-1},$$

where the second equality is because the bottom right entry of $\mathbf{E}$ is zero (see (3.1)). Note that the prediction is the bottom right entry of the output matrix. So only the last row of $\mathbf{W}_P^\top \mathbf{W}_V$ and the last column of $\mathbf{E}$ attend the prediction. Denote

$$\mathbf{W}_P^\top \mathbf{W}_V = \begin{pmatrix} * & * & * \\ \boldsymbol{w}^\top & \boldsymbol{a}_1 & \boldsymbol{a}_2 \end{pmatrix}, \quad \mathbf{W}_K^\top \mathbf{W}_Q = \begin{pmatrix} \boldsymbol{Q} & \boldsymbol{b} & * \\ \boldsymbol{q}_1^\top & b_1 & * \\ \boldsymbol{q}_2^\top & b_2 & * \end{pmatrix},$$

where

$$\boldsymbol{w} \in \mathbb{R}^d, \; \boldsymbol{a}_1, \boldsymbol{a}_2 \in \mathbb{R}, \; \boldsymbol{Q} \in \mathbb{R}^{d\times d}, \; \boldsymbol{b}, \boldsymbol{q}_1, \boldsymbol{q}_2 \in \mathbb{R}^d, \; b_1, b_2 \in \mathbb{R},$$

and $*$ denotes entries that do not enter the final prediction. Then we have

$$f(\mathbf{E}) = \begin{pmatrix} \boldsymbol{w}^\top & \boldsymbol{a}_1 & \boldsymbol{a}_2 \end{pmatrix} \frac{\mathbf{E}\mathbf{M}_M\mathbf{E}^\top}{M} \begin{pmatrix} \boldsymbol{Q} & \boldsymbol{b} & * \\ \boldsymbol{q}_1^\top & b_1 & * \\ \boldsymbol{q}_2^\top & b_2 & * \end{pmatrix} \begin{pmatrix} \mathbf{x} \\ 1 \\ 0 \end{pmatrix}$$

$$= \begin{pmatrix} \boldsymbol{w}^\top & \boldsymbol{a}_1 & \boldsymbol{a}_2 \end{pmatrix} \frac{\mathbf{E}\mathbf{M}_M\mathbf{E}^\top}{M} \begin{pmatrix} \boldsymbol{Q} & \boldsymbol{b} \\ \boldsymbol{q}_1^\top & b_1 \\ \boldsymbol{q}_2^\top & b_2 \end{pmatrix} \begin{pmatrix} \mathbf{x} \\ 1 \end{pmatrix}$$

$$= \begin{pmatrix} \boldsymbol{w}^\top & \boldsymbol{a}_1 & \boldsymbol{a}_2 \end{pmatrix} \frac{1}{M} \begin{pmatrix} \mathbf{X}^\top\mathbf{X} & \mathbf{X}^\top\mathbf{1}_M & \mathbf{X}^\top\mathbf{y} \\ \mathbf{1}_M^\top\mathbf{X} & M & \mathbf{1}_M^\top\mathbf{y} \\ \mathbf{y}^\top\mathbf{X} & \mathbf{y}^\top\mathbf{1}_M & \mathbf{y}^\top\mathbf{y} \end{pmatrix} \begin{pmatrix} \boldsymbol{Q} & \boldsymbol{b} \\ \boldsymbol{q}_1^\top & b_1 \\ \boldsymbol{q}_2^\top & b_2 \end{pmatrix} \begin{pmatrix} \mathbf{x} \\ 1 \end{pmatrix}.$$

Following the Assumption 3.1, we use $\widetilde{\boldsymbol{\beta}}$ to refer to the task parameter, then

$$\widetilde{\boldsymbol{\beta}} := \boldsymbol{\beta}^* + \boldsymbol{\Psi}^{\frac{1}{2}}\widetilde{\boldsymbol{\theta}}, \quad \text{where} \quad \widetilde{\boldsymbol{\theta}} \sim \mathcal{N}\left(\mathbf{0}_d, \mathbf{I}_d\right).$$

We then decompose $f(\mathbf{E})$ as

$$f(\mathbf{E}) = \underbrace{\begin{pmatrix} \boldsymbol{w}^\top & \boldsymbol{a}_1 & \boldsymbol{a}_2 \end{pmatrix} \frac{1}{M} \begin{pmatrix} \mathbf{X}^\top\mathbf{X} & \mathbf{X}^\top\mathbf{1}_M & \mathbf{X}^\top\mathbf{y} \\ \mathbf{1}_M^\top\mathbf{X} & M & \mathbf{1}_M^\top\mathbf{y} \\ \mathbf{y}^\top\mathbf{X} & \mathbf{y}^\top\mathbf{1}_M & \mathbf{y}^\top\mathbf{y} \end{pmatrix} \begin{pmatrix} \boldsymbol{Q} \\ \boldsymbol{q}_1^\top \\ \boldsymbol{q}_2^\top \end{pmatrix}}_{\text{I}} \cdot \mathbf{x} + \text{II},$$

where terms I and II are independent of $\mathbf{x}$. Therefore, we have

$$\begin{aligned}
\mathcal{R}\left(f\right) &:= \mathbb{E}\left(f(\mathbf{E}) - y\right)^2 \\
&= \mathbb{E}\left(f(\mathbf{E}) - \langle\widetilde{\boldsymbol{\beta}}, \mathbf{x}\rangle\right)^2 + \sigma^2 && \text{since } y|\widetilde{\boldsymbol{\beta}}, \mathbf{x} \sim \mathcal{N}(0, \sigma^2) \\
&= \mathbb{E}\left(\text{I} \cdot \mathbf{x} + \text{II} - \langle\widetilde{\boldsymbol{\beta}}, \mathbf{x}\rangle\right)^2 + \sigma^2 && \text{since } f(\mathbf{E}) = \text{I} \cdot \mathbf{x} + \text{II} \\
&= \mathbb{E}\|\text{I}^\top - \widetilde{\boldsymbol{\beta}}\|_{\mathbf{H}}^2 + \mathbb{E}\text{II}^2 + \sigma^2 && \text{since } \mathbf{x} \sim \mathcal{N}(0, \mathbf{H})
\end{aligned}$$

$$\geq \mathbb{E}\|\mathbf{I}^\top - \widetilde{\boldsymbol{\beta}}\|_{\mathbf{H}}^2 + \sigma^2.$$

Note that the above equation holds if $\mathrm{II} = 0$, which can be easily achieved by setting $\boldsymbol{b} = \mathbf{0}_d, \boldsymbol{b}_1 = \boldsymbol{b}_2 = 0$. Therefore, without loss of generality, we can consider the LSA function which takes the following form:

$$f(\mathbf{E}) = \mathbf{I} \cdot \mathbf{x} = \begin{pmatrix} \boldsymbol{w}^\top & \boldsymbol{a}_1 & \boldsymbol{a}_2 \end{pmatrix} \frac{1}{M} \begin{pmatrix} \mathbf{X}^\top \mathbf{X} & \mathbf{X}^\top \mathbf{1}_M & \mathbf{X}^\top \mathbf{y} \\ \mathbf{1}_M^\top \mathbf{X} & M & \mathbf{1}_M^\top \mathbf{y} \\ \mathbf{y}^\top \mathbf{X} & \mathbf{y}^\top \mathbf{1}_M & \mathbf{y}^\top \mathbf{y} \end{pmatrix} \begin{pmatrix} \boldsymbol{Q} \\ \boldsymbol{q}_1^\top \\ \boldsymbol{q}_2^\top \end{pmatrix} \cdot \mathbf{x}. \qquad \text{(J.2)}$$

Let's then try to determine whether such a function can effectively represent a GD-$\boldsymbol{\beta}$ function, which takes the form of

$$f_{\text{GD-}\boldsymbol{\beta}}(\mathbf{E}) = \langle \boldsymbol{\beta}^*, \mathbf{x} \rangle - \left\langle \frac{\boldsymbol{\Gamma}^* \mathbf{X}^\top (\mathbf{X}\boldsymbol{\beta}^* - \mathbf{y})}{M}, \mathbf{x} \right\rangle, \qquad \text{(J.3)}$$

where $\boldsymbol{\beta}^*$ is the prior mean in Assumption 3.1 in our submission, and $\boldsymbol{\Gamma}^* = \boldsymbol{\Psi}\mathbf{H}^{\frac{1}{2}}\boldsymbol{\Omega}^{-1}\mathbf{H}^{\frac{1}{2}}$ and $\boldsymbol{\Omega} = \frac{M+1}{M}\mathbf{H}^{\frac{1}{2}}\boldsymbol{\Psi}\mathbf{H}^{\frac{1}{2}} + \frac{\sigma^2 + \text{tr}(\mathbf{H}\boldsymbol{\Psi})}{M}\mathbf{I}_d$ is defined in (5.3) in our submission. In order to achieve this, one simple way is to let

$$\boldsymbol{w} = -\boldsymbol{\beta}^*, \quad \boldsymbol{a}_1 = \boldsymbol{a}_2 = 1, \quad \boldsymbol{Q} = (\boldsymbol{\Gamma}^*)^\top, \quad \boldsymbol{q}_1 = \boldsymbol{\beta}^*, \quad \boldsymbol{q}_2 = \mathbf{0}_d. \qquad \text{(J.4)}$$

However, this will incur some additive terms and will potentially enlarge the ICL risk. Inserting the parameters above, we have

$$f(\mathbf{E}) = f_{\text{GD-}\boldsymbol{\beta}}(\mathbf{E}) + \frac{\mathbf{1}_M^\top}{M}\left(\mathbf{X}(\boldsymbol{\Gamma}^*)^\top - \mathbf{x}\boldsymbol{\beta}^*(\boldsymbol{\beta}^*)^\top + \mathbf{y}(\boldsymbol{\beta}^*)^\top\right) \cdot \mathbf{x}$$

This shows that the extended token with a single LSA cannot easily implement the GD-$\boldsymbol{\beta}$ function class and may incur an additive ICL risk depending on $\boldsymbol{\beta}^*$ (as shown in Theorem 4.1).

