# OpenReview forum: "In-Context Learning of a Linear Transformer Block: Benefits of the MLP Component and One-Step GD Initialization"
_NeurIPS.cc/2024/Conference — NeurIPS 2024 poster_

### Official Review · Reviewer_XdPR · 2024-06-27

**Soundness:** 3
**Presentation:** 4
**Contribution:** 3
**Rating:** 7
**Confidence:** 5

**Summary:**

This paper studies the in-context learning (ICL) in the linear regression setting with a Gaussian prior with a non-zero mean. They first prove that linear Transformer block (LTB) will enjoy a smaller approximation error than linear self-attention (LSA) in this setting, where the non-zero mean of Gaussian emerges. Second, they show that the core benefit brought by linear MLP is that LTB can implement the one-step gradient descent with learnable initialization (GD-$\beta$). Besides, all global minimizers in the LTB class are equivalent to the unique global minimizer in the GD-$\beta$ class. Finally, they investigate the non-convex dynamics of the GD-$\beta$ class and prove that it will converge to the unique global minimizer. Simulations on the GPT-2 support their theory.

**Strengths:**

1. Clarity: The paper is presented very clearly and is easy to follow.
2. Significance: The paper characterizes the benefit brought by the linear MLP in the ICL for linear regression setting, which is undoubtedly an important topic in the ICL theory community.
3. Quality: The paper investigates the additional effect of linear MLP from approximation and optimization perspectives in detail, and the derivation is solid.

**Weaknesses:**

1. In this paper, all non-linear parts in the transformer block (softmax and ReLU) are dropped. However, similar works on the non-linear transformer block have emerged in the ICL theory. [a] considers the ReLU MLP followed by a linear attention layer in the linear regression setting. [b] considers the standard transformer block in the in-context classification regime. Authors are suggested to discuss the potential impacts of these studies about non-linear transformers on the setting in this paper.
2. Though the LTB class enjoys the same global minimizer as the GD-$\beta$ class and the training dynamics of the GD-$\beta$ class can converge to the global minima, whether the training dynamics of the LTB class can converge to the global minimizer is not discussed by the authors. The authors should clarify this point.

[a] Transformers Learn Nonlinear Features In Context: Nonconvex Mean-field Dynamics on the Attention Landscape, ICML, 2024

[b] Training Nonlinear Transformers for Efficient In-Context Learning: A Theoretical Learning and Generalization Analysis, ICML, 2024

**Questions:**

1. In the LTB class, the paper does not merge $W_K^TW_Q$ and $W_P^TW_V$ into one matrix parameter, respectively. Are there any benefits? Does it will improve the approximation ability of the LTB?
2. In line 272, I think the $\Gamma$ is a precondition but not equivalent to the Newton step. Is it a typo?
3. The pretraining goal of this paper is different from the autoregression training in practice. Are there any potential approaches to study the autoregression pretraining and few-shot regression inference?

**Limitations:**

The authors have adequately addressed the limitations.

---

> ### Author Rebuttal · Authors · 2024-08-07
>
> Thank you for supporting our work! We will address your concerns as follows.
>
> Q1: In this paper, all non-linear parts in the transformer block (softmax and ReLU) are dropped. However, similar works on the non-linear transformer block have emerged in the ICL theory. [a] considers the ReLU MLP followed by a linear attention layer in the linear regression setting. [b] considers the standard transformer block in the in-context classification regime. Authors are suggested to discuss the potential impacts of these studies about non-linear transformers on the setting in this paper.
>
> A1: There are several differences between our work and [a,b]. Specifically, [a] considered an LSA layer after a network for ICL over the Barron class. In contrast, we consider a linear MLP after an LSA layer. In the former setting, the network plays the role of feature learning, while in the latter setting, the MLP plays the role of improving approximation power. [b] considered ICL of a stylized binary classification problem, while our focus is ICL of linear regression. The loss landscape of these two settings is significantly different. In the former setting, the authors use hinge loss for classification while we use square loss for regression. We will cite these two papers and discuss their relationship with our work in depth in the revision.
>
>
> Q2: Though the LTB class enjoys the same global minimizer as the GD-beta class and the training dynamics of the GD-beta class can converge to the global minima, whether the training dynamics of the LTB class can converge to the global minimizer is not discussed by the authors. The authors should clarify this point.
>
> A2: We agree that an optimization guarantee for GD-beta does not imply an optimization guarantee for LTB. This has been noted in Lines 362-366, but we will emphasize this more in the revision. However, the optimization of GD-beta offers insights for the optimization of LTB. First, as we have shown that the global minimizer of LTB is equivalent to that of GD-beta, so efficient optimization of GD-beta offers a way to find the global minimizer of LTB. Second, GD-beta can be viewed as LTB with restricted parameters. So optimization of GD-beta offers insights of optimization of LTB in certain subspaces. Finally, we emphasize that, although GD-beta is easier to optimize compared to LTB, its landscape is still non-convex, which captures part of the non-convexity challenges of the latter problem class.
>
>
> Q3: In the LTB class, the paper does not merge W_K^\top and W_P^\top W_V  into one matrix parameter, respectively. Are there any benefits? Does it will improve the approximation ability of the LTB?
>
> A3: We choose to present our results in terms of separate matrices just to be aligned with the practical Transformer setup. Our construction results also hold under merged matrices. Merging the matrices does not affect the approximation ability of LTB in our setting. We will clarify this in the revision.
>
> Q4: In line 272, I think the \Gamma is a precondition but not equivalent to the Newton step. Is it a typo?
>
> A4: In general $\Gamma$ is a precondition. When $M$ goes to infinity, the global optimal $\Gamma^*$ converges to $H^{-1}$, that is the inverse Hessian of the population MSE. Plugging this into equation (5.4), we get one step of Newton step under population MSE from $\beta^*$. We will clarify this in the revision.
>
> Q5: The pretraining goal of this paper is different from the autoregression training in practice. Are there any potential approaches to study the autoregression pretraining and few-shot regression inference?
>
> A5: Good question. We believe the few-shot regression inference results in Theorem 5.3 in [1] can be extended to GD-beta. However, we think autoregression training would not improve the few-shot regression inference for GD-beta, since the model capacity is limited. We believe a more comprehensive model is required to show the benefits of autoregression training.
>
> [1] Wu et al. ​​HOW MANY PRETRAINING TASKS ARE NEEDED FOR IN-CONTEXT LEARNING OF LINEAR REGRESSION? ICLR2024.

---

> > ### Comment · Reviewer_XdPR · 2024-08-08
> > **Official Comment by Reviewer XdPR**
> >
> > Thank you for providing the rebuttal. The authors adequately addressed my concerns, so my confidence level regarding my review of this paper has increased.

---

### Official Review · Reviewer_xFNi · 2024-07-09

**Soundness:** 3
**Presentation:** 3
**Contribution:** 2
**Rating:** 6
**Confidence:** 3

**Summary:**

his submission extends the earlier theoretical analyses of in-context learning with linear transformers by taking into account the data covariance and a non-zero mean on the task parameters. It is shown that a linear transformer with a linear MLP block (LTB) can implement the algorithm of one-step gradient descent with non-zero initialization. This additional flexibility of learned initialization allows LTB to match the Bayes optimal risk for linear regression, whereas the linear transformer studied in prior works cannot achieve such optimality at finite $M$. The authors also showed that the optimal one GD step solution can be obtained by training the parameters with gradient flow.

**Strengths:**

In the past years, there have been many works that examined the ICL ability of linear transformers in the linear regression setting, where it is typically shown that transformers implement one preconditioned GD step on the squared error in-context. This submission goes beyond the prior results by showing that linear transformers can adapt to the non-zero mean of the task distribution with the aid of a linear attention block. This yields a separation in the ICL efficiency between transformers with or without an attention block, which is an interesting message in my opinion.

**Weaknesses:**

I have the following concerns and questions:

1. Firstly, the obvious limitations of the analysis: the problem setting is still linear, and the authors only showed adaptivity of ICL to the task mean but not the full prior (see point below). Given the additional linear block, the approximation gap is not really surprising. The optimization analysis directly assumes the GD-$\beta$ parameterization, and it is not clear if gradient descent on the standard LTB reaches the same solution. Also, convergence is only shown at a population level with no finite-sample guarantees.

2. The Bayes optimal estimator in Lemma 6.1 is a generalized ridge regression estimator where the $\ell_2$ regularization is anisotropic and adapts to the prior. On the other hand, when $\beta^*=0$, the GD-$\beta$ solution in Theorem 5.2 reduces to the one-step GD estimator studied in prior works, which does not incorporate the prior covariance.
It is not clear to me why these two estimators always have the same statistical efficiency, as it is well known that anisotropic shrinkage provides a substantial advantage (Wu and Xu 2020) (Li et al. 2023). I guess the conclusion in Corollary 6.2 relies on $tr(H\Psi)$ being bounded. Can the authors comment on the optimality of GD-$\beta$ if this assumption is not satisfied?
(Wu and Xu 2020) *On the optimal weighted $\ell_2$ regularization in overparameterized linear regression*.
(Li et al. 2023) *Transformers as algorithms: generalization and stability in in-context learning*.

3. Some minor questions:
* On page 5, why does the "scratchpad" take the specific form of (3.1)?
* For the experiment reported in Table 1, what is the size of the trained model? Is the curriculum learning strategy in (Garg et al. 2022) also used?

---
**Post-rebuttal update:** the authors’ response addressed most of my concerns. I found that one of my criticisms is based on a misunderstanding of the optimal solution that the authors analyzed.
I have increased my score accordingly.

**Questions:**

See Weaknesses above.

**Limitations:**

See Weaknesses above.

---

> ### Author Rebuttal · Authors · 2024-08-07
>
> Thank you for your comments. We will address your concerns below.
>
> Q1: Firstly, the obvious limitations of the analysis: the problem setting is still linear, and the authors only showed adaptivity of ICL to the task mean but not the full prior (see point below). Given the additional linear block, the approximation gap is not really surprising. The optimization analysis directly assumes the GD-beta parameterization, and it is not clear if gradient descent on the standard LTB reaches the same solution. Also, convergence is only shown at a population level with no finite-sample guarantees.
>
> A1: Although an additional MLP component improves approximation power is to be expected, we think it is quite surprising to rigorously prove an approximation gap between LTB and LSA. Because such a result formally justifies the role of an MLP component in LTB for learning the mean signal.
>
> We agree that our results are in a simplified setup, where the task is linear and the optimization is under the GD-beta parameterization and at a population level. Yet even in the simplified setup, there is no such kind of result before our work. Without a full understanding of LTB in the simplest possible setup, it seems unlikely that one can theoretically understand LTB in more practical setups. As our work is the first work on studying LTB for ICL, we believe we have taken an important step in this direction and the contribution of our work is significant.
>
> Q2: The Bayes optimal estimator in Lemma 6.1 is a generalized ridge regression estimator where the $\ell_2$-regularization is anisotropic and adapts to the prior. On the other hand, when $\beta^*=0$, the GD-beta solution in Theorem 5.2 reduces to the one-step GD estimator studied in prior works, which does not incorporate the prior covariance. It is not clear to me why these two estimators always have the same statistical efficiency…. Can the authors comment on the optimality of GD-beta if this assumption is not satisfied?
>
>
> A2: We would like to point out a potential misunderstanding of our result. The GD-beta solution in Theorem 5.2 does incorporate the prior covariance. This is because $\Gamma^*$ is a function of $\Omega$ defined in Theorem 5.2, which is a function of the prior covariance. We hope this also clarifies the follow-up comments made by the reviewer.
>
> Q3: On page 5, why does the "scratchpad" take the specific form of (3.1)?
>
> A3: Initializing the “scratchpad” with a fixed constant is a natural choice.
> Here the constant is set to be $1$ for concreteness but this number is not special.
>
> Q4: For the experiment reported in Table 1, what is the size of the trained model? Is the curriculum learning strategy in (Garg et al. 2022) also used?
>
> A4: In our experiments, we use a small GPT2 model with 6 layers and 4 heads for each layer. The key dimension is 32 and the inner dimension in the FeedForward Network is 128. (Since we use a multi-head attention in the experiment, this does not directly correspond to the d_k and d_v in our theory part). We use the curriculum learning strategy in (Garg et al,2022). We will clarify this in the revision.

---

### Official Review · Reviewer_D6mY · 2024-07-13

**Soundness:** 3
**Presentation:** 3
**Contribution:** 3
**Rating:** 5
**Confidence:** 4

**Summary:**

In this paper, the authors investigate the in-context learning (ICL) of a linear transformer block (LTB), i.e., a single block comprised of a linear self-attention layer and a MLP layer. The task considered is ICL of linear regression with a Gaussian prior. Unlike earlier works that studied this problem, a Gaussian prior with a non-zero mean is used for generating the ICL tasks. For this setting, it is shown that the MLP layer is essential, as using only a self-attention layer results in a weaker approximation. Furthermore, the LTB at minimum is demonstrated to have a specific structure akin to one-step gradient descent with a learnable initialization, referred as GD-$\beta$. The optimisation and statistical properties of the GD-$\beta$ estimator is also widely studied.

**Strengths:**

a) In-context learning of statistical tasks particularly the linear regression has received particularly interest. The authors have identified an interesting and relevant modification to this problem setting when the ICL tasks are generated from a more general setup Gaussian prior with non-zero mean.

b) In this setting, an MLP layer and a skip connection are needed for approximation which is a novel finding. Even with these additional parameters it is shown that minimum of the task can be still be characterised and it corresponds to GD with learnable initialisation.

c) The statistical and non-convex optimization aspects of the GD-$\beta$ are interesting from a technical standpoint.

**Weaknesses:**

The paper attributes that property of learning the MLP layer, however the skip connection is also equally important. Hence it is a bit of an misleading claim to attribute this to the MLP layer. Overall, the model restricted to only linear attention and also a single layer.

**Questions:**

1)  What happens if we include positional encoding (P) are also included?  It is possible to get an additional $P^{\top} W_K^{\top} W_QP$ term can come from the attention and can this overcome the need of positional encoding?

2) Does the optimisation dynamics of GD-$\beta$ offer any insights on the optimisation of the LTB ?

3) Does the results qualitatively change if the input sequence is generated from uncentered Gaussian $\mathcal{N}( \mu, H )$ instead of zero mean Gaussian prior ?


Minor corrections:

1) In l.251, it should be $ \Tau^{\top} $ in the expression of $W_K^{\top} W_Q$.

2) In l.263, an additional bracket in the expression of $\Omega$.

---

> ### Author Rebuttal · Authors · 2024-08-07
>
> We appreciate your comments. We will make sure to fix the typos in the revision. We will address your concerns below.
>
> Q1: The paper attributes that property of learning the MLP layer, however the skip connection is also equally important. Hence it is a bit of an misleading claim to attribute this to the MLP layer. Overall, the model restricted to only linear attention and also a single layer.
>
> A1: We emphasize that the ability of LTB to learn non-zero mean is a joint effect of an MLP component and a skip connection. Note that LSA also has a skip connection (see also [1,2]) but its skip connection is inactive. In comparison, the MLP component in LTB activates the skip connection. Therefore, we attribute the ability to learn non-zero mean to the MLP component, which is the only difference between LTB and LSA.
>
> Nonetheless, we agree one can attribute the ability to learn non-zero mean to the skip connection — as you have mentioned, without a skip connection, LTB reduces to LSA with a potential rank constraint on the parameter, which cannot learn non-zero mean as we have proved. The above two explanations take different perspectives to interpret the same phenomenon. We believe the way of interpretation does not diminish the significance of our contribution. We have discussed this in Lines 198-206 in our submission, but we will clarify this further in the revision.
>
> Q2: What happens if we include positional encoding (P) are also included? It is possible to get an additional $P^\top W_K^\top W_Q P$ term can come from the attention and can this overcome the need of positional encoding?
>
> A2: This is an open question. Although we show evidence that a scathpad hardly helps, it may be possible that a positional encoding can achieve the effect of a MLP component. There are several ways of implementing positional encoding. For example, GPT 2 uses an additive trainable matrix $P$ as positional encoding. Note that this significantly increases the amount of trainable parameters in both LSA or LTB, and also makes optimization harder. A more popular method for positional encoding is to use sinusoidal functions, which are only partially trainable. A fully rigorous analysis of the effect of all kinds of positional encoding is an interesting question, which we will comment on as a future direction. With that being said, we believe our current results, which clarify the benefits of an MLP component, have already made significant contributions.
>
> Q3: Does the optimisation dynamics of GD-beta offer any insights on the optimisation of the LTB ?
>
> A3: Yes, we think the optimization of GD-beta offers insights for the optimization of LTB. First, as we have shown that the global minimizer of LTB is equivalent to that of GD-beta, so efficient optimization of GD-beta offers a way to find the global minimizer of LTB. Second, GD-beta can be viewed as LTB with restricted parameters. So optimization of GD-beta offers insights of optimization of LTB in certain subspaces. Finally, we emphasize that, although GD-beta is easier to optimize compared to LTB, its landscape is still non-convex, which captures part of the non-convexity challenges of the latter problem class. We will include these discussions in the revision.
>
> Q4: Does the results qualitatively change if the input sequence is generated from uncentered Gaussian $N(\mu, H)$ instead of zero mean Gaussian prior ?
>
> A4: Part of our results still hold when covariates are from uncentered Gaussian instead of centered Gaussian (see a detailed discussion below). Examining all of our results under uncentered Gaussian is left as a future work. In fact, we can derive an analog to Theorem 5.2 in our submission when the features $x$ are sampled from an uncentered Gaussian distribution $N(\mu,H)$. Without loss of generality, we assume $\Psi$ and $H$ are invertible. Under assumption 3.1 in our submission, we have the global minimizer of the GD-$\beta$ class is $\beta = \beta^*$ and
>
> $$\Gamma = \Psi (H + \mu \mu^\top) \cdot A^{-1},$$
>
> Where $A = \frac{1}{M^2} \mathbb{E}[X^\top X \Psi X^\top X] + \frac{\sigma^2}{M}(H+\mu \mu^\top)$ is a positive definite matrix and $X \in \mathbb{R}^{d \times M}$ is the feature matrix. The optimal ICL risk is $\sigma^2 + tr(\Psi (H+\mu \mu^\top) - (H+\mu \mu^\top) \Psi (H+\mu \mu^\top) A^{-1} (H+\mu \mu^\top) \Psi),$ where $A$ is defined above. Moreover, since the LTB class can always implement a GD-$\beta,$ we know there exists an LTB function that can achieve a constant level of ICL risk regardless of how large $\beta^*$ is.
>
> [1]. R Zhang et al.  Trained Transformers Learn Linear Models In-Context. JMLR 2024.
> [2]. K Ahn et al. Transformers learn to implement preconditioned gradient descent for in-context learning. NIPS 2023

---

> > ### Comment · Reviewer_D6mY · 2024-08-12
> > **Reply to Author Rebuttal**
> >
> > Thank you for the rebuttal and for addressing my questions. The rebuttal did not change my initial impression of the paper: it presents an interesting point but also has its share of limitations. Therefore, I maintain my rating.

---

### Official Review · Reviewer_mBt4 · 2024-07-29

**Soundness:** 3
**Presentation:** 3
**Contribution:** 3
**Rating:** 7
**Confidence:** 3

**Summary:**

This paper studies the in-context learning of linear regression with a Gaussian prior that has a finite, non-zero mean across tasks. Specifically, the authors show that a linear transformer block (linear attention with an MLP layer) can achieve near-optimal Bayes risk for this task, whereas a linear attention-only block suffers from an approximation error. They further demonstrate that an LTB hypothesis class can be well-represented by a subset class of one-step gradient descent estimators, GD-$\beta$, for the ICL task.

**Strengths:**

This work delves into theoretically understanding the very relevant and hot topic of ICL. The problem motivation is clear and concise, and I really like how the authors essentially redo the previous results of LSA and GD-$0$ estimators with LTB and GD-$\beta$, demonstrating the effective correspondence between them and showing that they achieve near Bayes optimal ICL risk. Overall, I think the authors do a great job of portraying the benefits of the MLP component in transformers, and it’s a good theoretical contribution, as most papers involve theory only from the LSA perspective.

**Weaknesses:**

I don’t see any major weaknesses in the paper as such, but I do have a few questions (see questions).

**Questions:**

Have the authors seen experimental evidence that trained LTB models converge to GD-$\beta$ models, similar to how LSA converges to GD-$0$? I believe the correspondence established between the two is excellent, and theoretically studying the convergence is probably another work in itself, but seeing some experimental evidence would have been really cool.

I believe the restrictions imposed on $d_k, d_v$ for the hypothesis classes are to ensure sufficient overparameterization, which is pretty common for theoretical works. However, I think in practice these dimensions are typically set to be less than $d$. Are these necessary conditions, or do the authors believe there are ways to override them for their analysis?

The constructions for Lemma $5.1$ and Theorem $5.3$ both operate on pairs of matrices composed together (if not more), which is also the common practice in the analysis that I’ve seen. Any reasons why the authors didn’t opt for analyzing with merged matrices, especially when the inner dimension of the product $W_p^TW_v​$, that is $d_v$​, is assumed to be larger than $d$? It doesn’t even enforce any additional rank constraints on the product matrix.

It was interesting to see that in your GPT-2 experiments, the full transformer model (with all the non-linearities) also relied heavily on the MLPs. How much of this do you think can be attributed to the presence of softmax only? Did you try experiments where you used LSA in GPT-2 training with MLPs?

**Limitations:**

Yes, the authors have addressed the limitations.

---

> ### Author Rebuttal · Authors · 2024-08-07
>
> Thank you for supporting our paper! We answer your questions below.
>
> Q1: Have the authors seen experimental evidence that trained LTB models converge to GD-beta models, similar to how LSA converges to GD-0?
>
> A1: Yes, empirically we also see that trained LTB models converge to GD-beta models. Similar results are also obtained in [2] for LSA converging to the GD-0 (see their figure 2). We will try to add the figure for LTB converging to GD-$\beta$ in the next version.
>
> Q2: I believe the restrictions imposed on $d_k$, $d_v$ for the hypothesis classes are to ensure sufficient overparameterization, which is pretty common for theoretical works. However, I think in practice these dimensions are typically set to be less than $d$. Are these necessary conditions, or do the authors believe there are ways to override them for their analysis?
>
> A2: In multi-head attention, $d_k$ and $d_v$ are set to be $(d+1)/m$, where $m$ is the number of heads and $d+1$ is the embedding size. Since we only consider single head attention, this reduces to $m=1$ and $d_k = d_v = d+1$, which satisfies our assumptions.
>
> The assumption that $d_k, d_v \ge d$ simplifies the discussion, but we believe our results can be generalized under relaxed assumptions using standard techniques. For instance, if $d_k < d$, then the global minimum of LSA is achieved when $W_K^\top W_Q$ is the best rank-$d_k$ approximation of $\Gamma^*$. Similar results are obtained in [1]. We expect a similar result can be proved for LTB, where the global minimum is achieved by certain low-rank approximators.
>
> Q3: The constructions for Lemma 5.1 and Theorem 5.3 both operate on pairs of matrices composed together (if not more)... Any reasons why the authors didn’t opt for analyzing with merged matrices, especially when the inner dimension of the product $W_p^\top W_v$​, that is $d_v$​, is assumed to be larger than d?...
>
> A3: We choose to present our results in terms of separate matrices just to be aligned with the practical Transformer setup. As you mentioned, our construction results also hold under merged matrices. We will clarify this in the revision.
>
> Q4: It was interesting to see that in your GPT-2 experiments, the full transformer model (with all the non-linearities) also relied heavily on the MLPs. How much of this do you think can be attributed to the presence of softmax only? Did you try experiments where you used LSA in GPT-2 training with MLPs?
>
> A4: We think the gap observed in our GPT-2 experiments (w/ MLP or w/o MLP) is caused by MLPs instead of softmax since softmax is applied in both cases. We thank the reviewer for the proposal to train deep LTB and LSA models. However, training deep linear models is extremely hard without normalization techniques like softmax, and we observe unstable training performance when we train deep LTB and LSA. In practice, the normalization is important for training a deep model.
>
> [1]. Y Li et al. Fine-grained Analysis of In-context Linear Estimation: Data, Architecture, and Beyond. 2024
> [2]. JV Oswald et al. Transformers Learn In-Context by Gradient Descent. ICML2023.

---

> ### Comment · Reviewer_mBt4 · 2024-08-12
>
> Thank you for answering my questions. Please add the figure regarding Q1, and the clarification regarding Q3 in the next version. I will keep my positive rating and recommend acceptance.

---

### Decision · Program_Chairs · 2024-09-25

**Decision:**

Accept (poster)

**Comment:**

This paper provides the benefit of an MLP layer on top of a linear transformer on an in-context learning task. This paper generalizes the problem setting to a non-zero mean task distribution while previous work mainly focused on mean 0 task distributions. The MLP layer is beneficial to deal with such a non-zero mean of the task distribution. Finally, it shows the convergence of gradient flow on the population risk .

- The paper is well written. The message of the paper is very clear and thus the readers can easily grasp the main contribution.
- Although there is not so much mathematical difficulty, this paper gives an interesting contribution to the line of research on this topic. Especially, it is nice that the paper provides not only statistical advantage but also optimization guarantee on the population risk.

Indeed, the reviewers are overall positive on this paper. I also agree with their opinion and would like to recommend acceptance of this paper.